# RETHINKING LLM UNLEARNING OBJECTIVES: A GRADIENT PERSPECTIVE AND GO BEYOND

**Qizhou Wang**[1][*]  **Jin Peng Zhou**[2]  **Zhanke Zhou**[1]  **Saebyeol Shin**[2]
**Bo Han**[1][†]  **Kilian Q. Weinberger**[2]

[1]TMLR Group, Department of Computer Science, Hong Kong Baptist University
[2]Department of Computer Science, Cornell University

## ABSTRACT

Large language models (LLMs) should undergo rigorous audits to identify potential risks, such as copyright and privacy infringements. Once these risks emerge, timely updates are crucial to remove undesirable responses, ensuring legal and safe model usage. It has spurred recent research into LLM unlearning, focusing on erasing targeted undesirable knowledge without compromising the integrity of other, non-targeted responses. Existing studies have introduced various unlearning objectives to pursue LLM unlearning without necessitating complete retraining. However, each of these objectives has unique properties, and no unified framework is currently available to comprehend them thoroughly. To fill the gap, we propose a toolkit of the gradient effect (G-effect), quantifying the impacts of unlearning objectives on model performance from a gradient perspective. A notable advantage is its broad ability to detail the unlearning impacts from various aspects across instances, updating steps, and LLM layers. Accordingly, the G-effect offers new insights into identifying drawbacks of existing unlearning objectives, further motivating us to explore a series of new solutions for their mitigation and improvements. Finally, we outline promising directions that merit further studies, aiming at contributing to the community to advance this important field. The code is publicly available at: https://github.com/tmlr-group/G-effect.

## 1 INTRODUCTION

Large language models (LLMs) (Brown et al., 2020; Touvron et al., 2023b; Achiam et al., 2023) represent the cutting edge of machine learning for the field of language understanding. These models typically leverage multi-head attention decoder-based architectures (Vaswani et al., 2017) with billions of learnable parameters and are autoregressively trained (Zhao et al., 2023) over web-sourced datasets encompassing trillions of tokens. Such extensive scaling enables LLMs to handle a broad spectrum of complex linguistic tasks, demonstrating remarkable proficiency in understanding and generating languages across a board range of practical applications.

The scaling of LLMs, on the other side, also brings notable drawbacks alongside its benefits. A primary concern is their high tendency to memorize data, which can reproduce sensitive information once encountered during web-sourced training, such as copyright and privacy-related content (Yao et al., 2023a). These issues are particularly concerning due to the potential misuse of LLMs for illegal activities (Li et al., 2024), also posing challenges to protect individual rights to be forgotten (Zhang et al., 2023). Mitigating these undesirable behaviors in LLMs is non-trivial, involving regularly auditing LLMs to recognize sensitive content and subsequently adjusting the associated, parameterized knowledge. In previous works, supervised fine-tuning (De Cao et al., 2021) and alignment methods (Ouyang et al., 2022) have been explored to overwrite LLMs against such undesirable model behaviors. However, these well explored directions face practical deficiencies—they can be costly (Yao et al., 2023b), require high-quality crafted preference datasets (Chowdhury et al., 2024), and exhibit concerns regarding robustness (Patil et al., 2023; Qi et al., 2023; Wang et al., 2024).

---

[*]Work done during a remote visiting at Cornell University.
[†]Correspondence to Bo Han (bhanml@comp.hkbu.edu.hk).

**LLM unlearning** (Yao et al., 2023b) has emerged as a promising alternative, with a direct goal of removing parameterized knowledge targeted to be unlearned, meanwhile preserving the model integrity for all other, non-targeted data (Wang et al., 2025). Highlighted by Yao et al. (2023b), LLM unlearning is cost-effective over aforementioned more demanding methods, thus attracting emerging research attention these days (Liu et al., 2024). A representative baseline of LLM unlearning is gradient ascent (GA) (Maini et al., 2024), adjusting LLMs to increase the prediction losses for targeted data—thereby removing parameterized knowledge. GA offers a potentially viable path to implement LLM unlearning; however, it is severely susceptible to excessive unlearning (Zhang et al., 2024), where the effectiveness in removing undesirable data comes at the high cost of compromising the overall model integrity. It motivates a series of subsequent works that improve upon GA, such as negative preference optimization (NPO) (Zhang et al., 2024), preference optimization (PO) (Maini et al., 2024), and representation misdirection for unlearning (RMU) (Li et al., 2024).

Given the increasing number of unlearning objectives, we need to discern good objectives from those less promising ones. A step further, it is also interesting to pinpoint beneficial components within existing methods, isolating those that are useless or potentially harmful. Sadly, to our knowledge, a general toolkit for in-depth analysis of various unlearning methods is still lacking. To bridge this gap, we propose the concept of the gradient effect (G-effect), which approximates the performance change associated with particular unlearning objectives via the dot product of their gradients, cf., Definition 1. The G-effect provides more than mere performance evaluations—it enables detailed examinations of various unlearning methods for their impacts with respect to data points, updating steps, and layers, cf., Section 4. We outline below for some of the general observations we achieved.

- **Unlearning affects shallow layers more.** Shallow layers are more affected than deeper layers, showing that general knowledge encoded in shallow layers undergoes substantial alterations.

- **Unlearning compromises retention.** Although conceptually existing (cf., Section 3), current unlearning objectives all fail to retain the overall performance when conducting unlearning.

- **Excessive unlearning is harmful.** An excessive extent of unlearning has severe impacts such that the deterioration in common model responses can outweigh improvements in unlearning.

- **Risk weighting is powerful.** Prioritizing certain points is justified to be effective for unlearning. However, there still exists a large space to further refine risk weighting mechanisms.

- **Regularization is important.** Regularization terms continue to play a crucial role in maintaining overall model integrity, with the KL (Maini et al., 2024) emerging as a promising choice.

We benchmark both existing and new methods explored throughout our analysis on the well-established TOFU fictitious unlearning datasets (Maini et al., 2024). Our experiments identify several new state-of-the-arts that merit further attention. Additionally, based on our analysis, we highlight promising research directions that warrant exploration to further advance the field.

## 2 LLM UNLEARNING

We focus on auto-regressive LLMs parameterized by $\boldsymbol{\theta}$, which recursively estimate the probability distributions over the next tokens, denoted as $p(\cdot|s; \boldsymbol{\theta})$. LLMs are, in general, trained on large-scale, web-sourced corpora following the distribution $\mathcal{D}_{\mathrm{t}}$ with the negative log-likelihood (NLL) loss function of $-\log p(s; \boldsymbol{\theta})$, where $p(s; \boldsymbol{\theta}) = \prod_{i=2}^{|s|} p(s^i|s^{<i}; \boldsymbol{\theta})$ with $s^i$ the $i$-th token and $s^{<i}$ the prefix up to $s^i$. While LLMs are capable of handling a broad spectrum of language generation tasks, the use of training corpora sourced from the open world raises the risk that our LLMs will learn from sensitive data, precipitating a series of legal and ethical concerns (Liu et al., 2023).

**LLM Unlearning.** These issues necessitate the need for a post-training mechanism that enables LLMs to eradicate any parameterized knowledge that is undesirable. This requirement motivates the recent research on LLM unlearning (Yao et al., 2023b; Maini et al., 2024), of which the main goals are in two folds—**(a)** ensuring the removal of data / knowledge targeted to be unlearned and **(b)** retaining the integrity of model responses for non-targeted data. Formally, we consider the data distribution $\mathcal{D}_{\mathrm{u}}$ that should be unlearned and define the risk metric $\mathcal{R}$ to assess model performance. Then, our goal is to adjust the original LLM parameters $\boldsymbol{\theta}_{\mathrm{o}}$ to get the unlearned ones $\boldsymbol{\theta}_{\mathrm{u}}$, such that:

- **Removal.** The performance on the unlearning dataset $\mathcal{D}_u$ should significantly deteriorate, i.e., $\mathcal{R}(\mathcal{D}_u; \boldsymbol{\theta}_u) \gg \mathcal{R}(\mathcal{D}_u; \boldsymbol{\theta}_o)$, revealing effective unlearning on data targeted to be erased.

- **Retention.** The performance on other data, i.e., $\mathcal{D}_t \backslash \mathcal{D}_u$, should be maintained or enhanced, i.e., $\mathcal{R}(\mathcal{D}_t \backslash \mathcal{D}_u; \boldsymbol{\theta}_u) \leq \mathcal{R}(\mathcal{D}_t \backslash \mathcal{D}_u; \boldsymbol{\theta}_o)$, ensuring model responses on common data are not damaged.

Here, we consider a practical objective of erasing targeted knowledge as much as possible (Liu et al., 2024), i.e., full removal, slightly diverging from the classical definition of machine unlearning (Bourtoule et al., 2021) that seeks to make models behave as if they were trained without the targeted data, i.e., influence removal. The pursuit of full removal simplifies our subsequent discussions, and as we will see later in Section 5, the outcomes of these two goals are typically aligned.

This paper delves into exploring various objective functions that implement LLM unlearning, a topic that requires our fundamental interest. As an example, GA (Yao et al., 2023b) directly increases the NLL loss for targeted data, of which the objective is articulated as $\min_\theta \mathbb{E}_{s_u \sim \mathcal{D}_u} \log p(s_u; \boldsymbol{\theta})$. GA represents one of the pioneering methods for LLM unlearning, paving a feasible road to implement unlearning in practice. However, it often exhibits the propensity to excessive unlearning (Zhang et al., 2024; Wang et al., 2025)—the efficacy in eliminating undesirable knowledge comes at a high cost to compromise the model integrity. It motivates a series of subsequent works (Zhang et al., 2024; Maini et al., 2024; Li et al., 2024), which will be discussed later in Section 4.

## 3    G-EFFECT

Before delving into specific methods, we need proper criteria for assessing whether an objective is suitable for unlearning or not. Recalling our earlier discussion on the main goals of unlearning, we can quantify the performance change before and after unlearning to evaluate their effects, i.e., $\mathcal{R}(\mathcal{D}_u; \boldsymbol{\theta}_u) - \mathcal{R}(\mathcal{D}_u; \boldsymbol{\theta}_o)$ for removal and $\mathcal{R}(\mathcal{D}_t \backslash \mathcal{D}_u; \boldsymbol{\theta}_u) - \mathcal{R}(\mathcal{D}_t \backslash \mathcal{D}_u; \boldsymbol{\theta}_o)$ for retention. Sadly, merely comparing performance provides limited insights into understanding the underlying mechanisms (cf., Section 4). Therefore, we suggest a more insightful scheme that can facilitate the analysis of various unlearning methods from a gradient perspective, named the gradient effect (G-effect).

Generally speaking, the G-effect compares the gradients of the unlearning objective $\mathcal{L}_u$ and the risk metric $\mathcal{R}$. If the gradients of $\mathcal{L}_u$ align in similar directions to $\mathcal{R}$, model updating based on $\mathcal{L}_u$ is capable to enhance model performance measured by $\mathcal{R}$, an obvious alternative of $\mathcal{R}(\mathcal{D}; \boldsymbol{\theta}_u) - \mathcal{R}(\mathcal{D}; \boldsymbol{\theta}_o)$ to measure the performance change. The degree of such similarity between gradients can be quantified using their dot products (Lopez-Paz & Ranzato, 2017): A positive dot product indicates that $\mathcal{L}_u$ is capable to improve $\mathcal{R}$, whereas a negative dot product suggests potential harm to $\mathcal{R}$. Please refer to Appendix A for a formal derivation. It motivates the G-effect as follows.

**Definition 1** (G-effect). *The G-effect $e^{(t)}$ for an unlearning objective $\mathcal{L}_u$ at the $t$-th step of model updating is given by $\nabla_{\boldsymbol{\theta}} \mathcal{R}(\mathcal{D}; \boldsymbol{\theta}^{(t)})^\top \nabla_{\boldsymbol{\theta}} \mathcal{L}_u(\mathcal{D}_u; \boldsymbol{\theta}^{(t)})$. We further define the **unlearning G-effect** $e_u^{(t)} \leftarrow \nabla_{\boldsymbol{\theta}} \mathcal{R}(\mathcal{D}_u; \boldsymbol{\theta}^{(t)})^\top \nabla_{\boldsymbol{\theta}} \mathcal{L}_u(\mathcal{D}_u; \boldsymbol{\theta}^{(t)})$ and the **retaining G-effect** $e_r^{(t)} \leftarrow \nabla_{\boldsymbol{\theta}} \mathcal{R}(\mathcal{D}_t \backslash \mathcal{D}_u; \boldsymbol{\theta}^{(t)})^\top \nabla_{\boldsymbol{\theta}} \mathcal{L}_u(\mathcal{D}_u; \boldsymbol{\theta}^{(t)})$ to reflect the respective goals of removal and retention.*

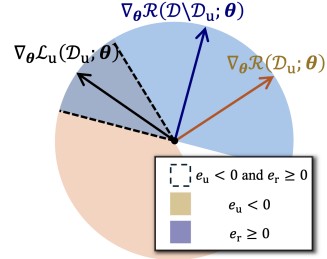

The G-effect measures the impacts of unlearning objectives on either targeted or non-targeted data when implementing gradient updates. Overall, to fulfill the unlearning goals outlined in Section 2, we aim for notably negative values of $e_u^{(t)}$ to pursue a sufficient removal of targeted knowledge and non-negative values of $e_r^{(t)}$ to maintain the model integrity for non-targeted data. Figure 1 further depicts these two essential gradient conditions to ensure effective unlearning:

- **Removal.** The red region indicates $e_u^{(t)} < 0$, ensuring $\mathcal{L}_u$ to eliminate targeted knowledge.

- **Retention.** The blue region represents $e_r^{(t)} \geq 0$, ensuring $\mathcal{L}_u$ to retain the overall model integrity.

Figure 1: **Gradient Directions and Unlearning Behaviors**. We show directions for $\nabla_{\boldsymbol{\theta}} \mathcal{R}(\mathcal{D}_u; \boldsymbol{\theta}_o)$ and $\nabla_{\boldsymbol{\theta}} \mathcal{R}(\mathcal{D} \backslash \mathcal{D}_u; \boldsymbol{\theta}_o)$ and regions ensuring $e_u^{(t)} < 0$ (red) and $e_r^{(t)} \geq 0$ (blue). Their intersection (black dashed) fulfills the unlearning goals.

**What Can We Learn from the G-effect?** In Figure 1, the intersection, delineated by black dashed lines, is the region that meets the primary goals of unlearning—effective removal of targeted knowledge while retaining the integrity of non-targeted data. This area highlights the conceptual possibilities of achieving a perfect unlearning objective, under the mild conjecture that $\nabla_{\boldsymbol{\theta}}\mathcal{R}(\mathcal{D}_{\mathrm{u}};\boldsymbol{\theta}_{\mathrm{o}})$ will differ from $\nabla_{\boldsymbol{\theta}}\mathcal{R}(\mathcal{D}\backslash\mathcal{D}_{\mathrm{u}};\boldsymbol{\theta}_{\mathrm{o}})$ more or less. Also, the dependency on $t$ enables us to examine the dynamics of unlearning, and the gradients helps us to explore the impacts of particular layers or data points involved during unlearning. It will facilitate our understanding of existing unlearning mechanisms, which will be detailed below. Please refer to Appendix A for more discussions.

# 4 ANALYSIS FOR UNLEARNING OBJECTIVES

In this section, we employ the G-effect to assess a range of unlearning objectives that are well-recognized, aiming to understand their mechanisms as well as identify their advantages and deficiencies. Due to the high costs in fully computing the G-effect, we focus on experiments based on 5% TOFU fictitious unlearning (Maini et al., 2024) with Llama-2-7B (Touvron et al., 2023a) (cf. Appendix B). Later in Section 5, we will demonstrate that the conclusions can be generalized to many other unlearning scenarios. All the methods will run for 5 epochs, totaling about 60 steps. Moreover, as indicated in Fig-

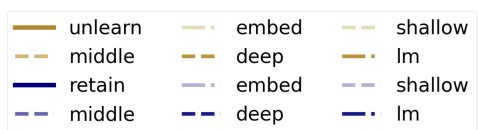

Figure 2: **Figure Legends**. We present the unlearning (unlearn) and the retaining (retain) G-effect, and also their values for specific layers, including input embedding layer (embed), layers 1-11 (shallow), layers 12-22 (middle), layers 23-33 (deep), and output unembedding layer (lm).

ure 2, we will report the unlearning (red) and retaining (blue) G-effect, as well as their detailed values for particular layers within Llama-2-7B (dashed lines for the stacks of layers and dash-dotted lines for input/output layers). We default to consider the NLL as the risk metric $\mathcal{R}$.

## 4.1 GRADIENT ASCENT (GA)

As discussed in Section 2, GA represents one of the most basic unlearning objectives (Yao et al., 2023c), which simply decreases the log-likelihood $\log p(s_{\mathrm{u}};\boldsymbol{\theta})$ for the unlearning data.

**The G-effect across Unlearning Steps.** We illustrate the G-effect of GA in Figure 3(a). As we can see, the unlearning G-effect reflects the high capability of GA in erasing parameterized knowledge for targeted data, with its values rapidly declining from about $0$ to $-3.5 \times 10^5$. However, this excessive extent of unlearning incurs a large cost to the integrity for non-targeted data, evidenced by the trajectory of negative values in the retaining G-effect that mirror the scales and trends of the unlearning G-effect. Overall, such a scenario suggests that the improvements in unlearning are accompanied by similar, or even greater, deterioration on non-targeted data.

Note that relatively near-zero values of the G-effect in the later updating stages do not imply that the model can relearn the knowledge. In general, the G-effect exhibits cumulative behaviors, where the presence of extremely negative G-effect, particularly between steps 20 to 40, has already indicated a large deterioration on model performance. Smaller values of the G-effect in the later stages only suggest that the subsequent damage to model integrity is less severe, mainly due to the GA objective reaching its empirical convergence stage. Refer to Figure 9(b) in Appendix C.1 for more analysis.

**The G-effect across Layers.** We also observe that the G-effect values are notably greater in the shallow layers than those in the middle and deep layers, which can be more clearly shown in Figure 3(b). It indicates that the general knowledge, which is parameterized within shallow layers, is notably distorted, while such side impacts are less severe for middle and deep layers with context-specific knowledge (Geva et al., 2020; Belrose et al., 2023). It is also worth noting that we isolate the input embedding layer (embed) from other shallow layers (shallow), where we observe that the input embedding layer has relatively negligible impacts on both removal and retention, highlighting the distinct influences of the GA unlearning procedure on the input embedding layer and other shallow layers. Furthermore, the middle and top layers exhibit much smaller G-effect values than that for shallow layers. However, the G-effect for the last layer, i.e., the output unembedding layer (LM),

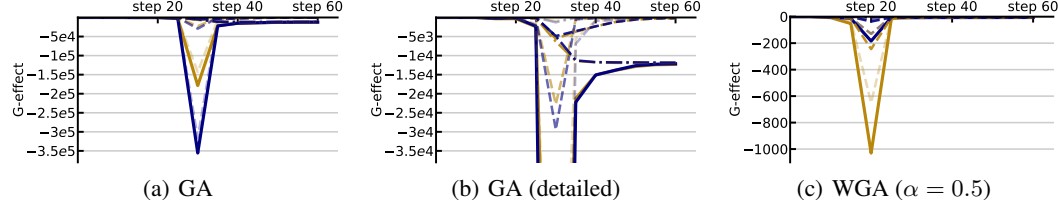

(a) GA            (b) GA (detailed)            (c) WGA ($\alpha = 0.5$)

Figure 3: **The G-effect for GA and WGA.** We depict the G-effect for GA in (a) and its values in the range between about $-3.5 \times 10^4$ and 0 in (b). We further depict the G-effect for WGA, which improves upon GA following equation 2, in (c). The **legends** are summarized in Figure 2. The **horizontal axis** denotes the unlearning step and the **vertical axis** denotes the values of the G-effect.

is notably large and do not converge to near zero. This behavior suggests that such a linear model performs some scaling operations to further reduce the GA objective that is lower unbounded.

**Unlearning Mechanisms.** We hope to further explore the unlearning behaviors behind GA, particularly focusing on its wrong tendency towards exhibit extremely negative values of the retaining G-effect. Specifically, the gradients of $\mathcal{L}_{\text{GA}}(\mathcal{D}_{\text{u}}; \boldsymbol{\theta})$ with respect to $\boldsymbol{\theta}$, i.e., $\nabla_{\boldsymbol{\theta}} \mathcal{L}_{\text{GA}}(\mathcal{D}_{\text{u}}; \boldsymbol{\theta})$, are

$$\mathbb{E}_{s_{\text{u}} \sim \mathcal{D}_{\text{u}}} \sum_{i=2}^{|s|} \underbrace{\frac{1}{p(s_{\text{u}}^i | s_{\text{u}}^{<i}; \boldsymbol{\theta})}}_{\text{inverse confidence}} \nabla_{\boldsymbol{\theta}} p(s_{\text{u}}^i | s_{\text{u}}^{<i}; \boldsymbol{\theta}), \tag{1}$$

where the inverse confidence term tends to allocate more attention to those tokens that have been notably unlearned, along with the decrease of the likelihood $p(s_{\text{u}}^i | s_{\text{u}}^{<i}; \boldsymbol{\theta})$ throughout GA.

In this case, even minor negative values of each $\nabla_{\boldsymbol{\theta}} \mathcal{R}(\mathcal{D}; \boldsymbol{\theta})^{\top} \nabla_{\boldsymbol{\theta}} p(s_{\text{u}}^i | s_{\text{u}}^{<i}; \boldsymbol{\theta})$ can result in the corresponding $\nabla_{\boldsymbol{\theta}} \mathcal{R}(\mathcal{D}; \boldsymbol{\theta})^{\top} p(s_{\text{u}}^i | s_{\text{u}}^{<i}; \boldsymbol{\theta})^{-1} \nabla_{\boldsymbol{\theta}} p(s_{\text{u}}^i | s_{\text{u}}^{<i}; \boldsymbol{\theta})$ to be extreme. This trend will lead to the extreme negative values of the unlearning G-effect, consistent with prior findings for the excessive unlearning of GA (Wang et al., 2025). Therefore, this inverse confidence mechanism is predominantly responsible for excessive unlearning. One can counteract the impacts of the inverse confidence by weighting the log-likelihood for each token via its own confidence, i.e.,

$$\mathbb{E}_{s_{\text{u}} \sim \mathcal{D}_{\text{u}}} \sum_{i=2}^{|s|} w_{s_{\text{u}}, i}^{\text{wga}} \log p(s_{\text{u}}^i | s_{\text{u}}^{<i}; \boldsymbol{\theta}) \tag{2}$$

with $w_{s_{\text{u}}, i}^{\text{wga}} = p(s_{\text{u}}^i | s_{\text{u}}^{<i}; \boldsymbol{\theta})^{\alpha}$ the confidence weights for the $i$-th token and $\alpha$ the hyper-parameter of inverse temperature. We refer to this approach as the weighted GA (WGA). An example of its G-effect in mitigating excessive unlearning is illustrated in Figure 3(c). Remarkably, we observe that the negative impact on common data is considerably less severe compared to the improvements observed on targeted data. Its underlying mechanism is not mystic, functioning as early stopping to curb the unlearning extent. Particularly, when the unlearning extent is well-controlled, even the original GA can outweigh the improvements of unlearning over the deterioration on integrity, a less obvious scenario that is further elaborated in Figure 10 of Appendix C.1. Overall, the findings emphasize that excessive unlearning profoundly compromises the overall model integrity, necessitating careful controls. For more detailed discussions about WGA, please refer to Appendix D.1.

### 4.2 NEGATIVE PREFERENCE OPTIMIZATION (NPO)

NPO is motivated by direct preference optimization (DPO), an alignment method (Rafailov et al., 2024), which originally utilizes paired corpora comprising preferred versus dis-preferred data. NPO segregates the dis-preferred part from DPO, heuristically employing it as the unlearning objective, of which the formulation can be written as

$$\frac{2}{\beta} \mathbb{E}_{s_{\text{u}} \sim \mathcal{D}_{\text{u}}} \log \left(1 + \left(\frac{p(s_{\text{u}}; \boldsymbol{\theta})}{p(s_{\text{u}}; \boldsymbol{\theta}_{\text{o}})}\right)^{\beta}\right), \tag{3}$$

where $\beta$ is the inverse temperature. NPO has shown notable enhancements over GA in preserving the model integrity, which is recognized as the current state-of-the-art within the community.

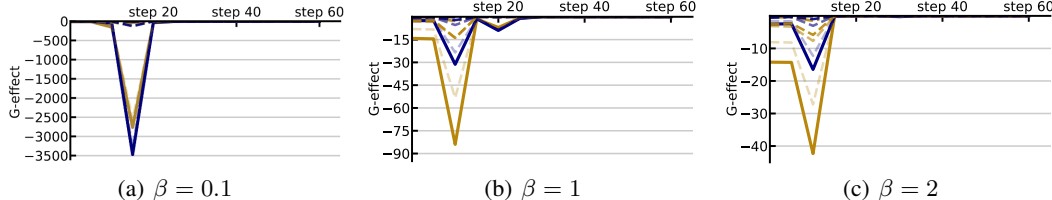

Figure 4: **The G-effect for NPO.** The **legends** are summarized in Figure 2. The **horizontal axis** represents the unlearning step and the **vertical axis** indicates the values of the G-effect.

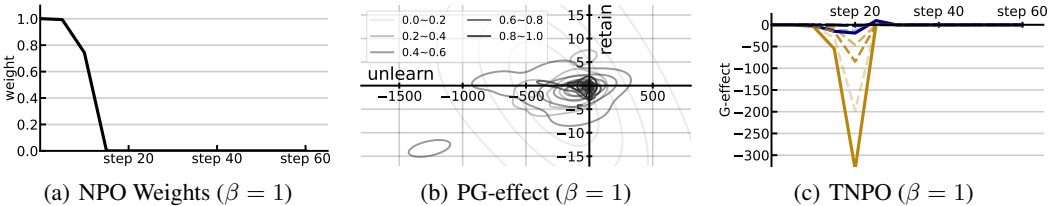

Figure 5: **The NPO Weighting Mechanisms.** We depict the curves of average NPO weights in (a) and the relationship of NPO weights with PG-effect in (b). Distributions of PG-effect for different value ranges of $w_{s_u}^{npo}$ are depicted, considering the checkpoints at 5, 10, and 15-th steps jointly. Moreover, darker shades within distribution contours signify the groups of $w_{s_u}^{npo}$ with overall larger weights. We further depict the G-effect for an improved version of NPO, named TNPO, in (c). The **horizontal axes** denote the unlearning step for (a) and (c), and the unlearning G-effect for (b). The **vertical axes** denote the NPO weights for (a), the retaining G-effect for (b), and the G-effect for (c).

**The G-effect across Unlearning Steps.** We show the G-effect of NPO in Figure 4. We observe that its values converge much faster than GA, aligning with previous observations (Zhang et al., 2024). Moreover, the magnitudes of G-effect for NPO are notably smaller than those observed with GA. In terms of the unlearning G-effect, it indicates that the unlearning strength of NPO is weaker; however, for the retaining G-effect, it suggests that NPO better preserves the model integrity. More importantly, magnitudes of retaining G-effect outweigh those of unlearning when $\beta = 1$ or 2, signifying that the negative impacts on model integrity are less pronounced than the beneficial effects of unlearning, rendering NPO a promising method to mitigate excessive unlearning.

**The G-effect across Layers and $\beta$.** Similar to GA, deeper layers exhibit weaker G-effect. However, both the input embedding and output linear layers display negligible values, which are different from the behaviors seen with GA. For both middle and deep layers, their retaining G-effect is relatively small. Furthermore, across different values of the inverse temperature, we observe that larger $\beta$ makes the G-effect converge faster and their magnitudes become smaller. This phenomenon generally arises because smaller $\beta$ causes the NPO formulation more closely resemble to that of GA (Zhang et al., 2024), of which the power in controlling the extent of unlearning is weaken. The relationship between GA and NPO is further elucidated below in equation 4.

**Unlearning Mechanisms.** We now aim to understand the factors that contribute to the efficacy of NPO. To begin with, we write the gradients of NPO with respect to $\theta$ in the following

$$\mathbb{E}_{s_u \sim \mathcal{D}_u} w_{s_u}^{npo} \nabla_{\theta} \, \log p(s_u; \theta), \tag{4}$$

with $w_{s_u}^{npo} = \frac{2p(s_u;\theta)^{\beta}}{p(s_u;\theta)^{\beta}+p(s_u;\theta_o)^{\beta}}$. Notably, compared with GA in equation 1, we find that NPO exhibits similar gradient formulation, albeit with a reweighting scheme $w_{s_u}^{npo}$. Therefore, $w_{s_u}^{npo}$ primarily contributes to the advantages of NPO over GA, thus requiring our in-depth focus.

We illustrate the curves of $w_{s_u}^{npo}$ in Figure 5(a), observing a rapid decrease of $w_{s_u}^{npo}$ from 1 to 0. The formulation of $w_s^{npo}$ reveals that, as the NPO risk decreases—indicative of the drop in the confidence $p(s_u; \theta)$—the weight $w_{s_u}^{npo}$ reduces consequently. This weighting behavior seems quite resemble to WGA. Then, the question arises whether $w_{s_u}^{npo}$ encompasses some intriguing mechanisms beyond

early stopping as in WGA. To further elucidate the G-effect of NPO, we expand it as follows:

$$\mathbb{E}_{s_{\mathrm{u}} \sim \mathcal{D}_{\mathrm{u}}} w_{s_{\mathrm{u}}}^{\mathrm{npo}} \underbrace{\nabla_{\boldsymbol{\theta}} \mathcal{R}(\mathcal{D}; \boldsymbol{\theta}^{(t)}) \nabla_{\boldsymbol{\theta}} \log p(s_{\mathrm{u}}; \boldsymbol{\theta}^{(t)})}_{\text{PG-effect of GA}}, \tag{5}$$

which details the G-effect on individual data points, represented as the product of the NPO weighting term $w_{s_{\mathrm{u}}}^{\mathrm{npo}}$ and the point-wise G-effect (refer to as PG-effect) of GA. Accordingly, we plot the joint distributions for the PG-effect of GA with respect to unlearning (i.e., $\mathcal{D} = \mathcal{D}_{\mathrm{u}}$) and retention (i.e., $\mathcal{D} = \mathcal{D}_{\mathrm{t}} \backslash \mathcal{D}_{\mathrm{u}}$) in Figure 5(b). These distributions are categorized into five groups based on the associated different value ranges of $w_{s_{\mathrm{u}}}^{\mathrm{npo}}$. As we can see, the distributions of PG-effects vary notably across different ranges of $w_{s_{\mathrm{u}}}^{\mathrm{npo}}$. Generally speaking, $w_{s_{\mathrm{u}}}^{\mathrm{npo}}$ tends to allocate larger weights to points where the retaining PG-effect is near-zero. it is a preferred scenario as $w_{s_{\mathrm{u}}}^{\mathrm{npo}}$ can prioritize data points that have small negative impacts on model integrity. However, the side effect is to emphasize those data points with less contributions to unlearning, thus compromising the unlearning strengths. We conclude that NPO weighting can prioritize certain points that have small negative impacts on non-targeted data, thereby enhancing the overall model integrity after unlearning.

**One Step Further.** We also notice some shortcomings for the NPO weighting mechanism. First, there are many failure cases where some data points with near-zero retaining PG-effect while large unlearning PG-effect is inappropriately assigned with small weights. Also, the distribution of PG-effect with $w_{s_{\mathrm{u}}}^{\mathrm{npo}}$ in the range between 0.4 to 0.6 demonstrates a wrong trend in assigning large weights to those data points that have large negative impacts on model integrity, i.e., notably negative retaining PG-effect. Ideally, we hope the weighting mechanism can prioritize points that not only have near-zero retaining G-effect and also exhibit large negative unlearning G-effect, a capability that the current NPO weighting does not possess.

It is worth noting that our above analysis does not disqualify $w_{s_{\mathrm{u}}}^{\mathrm{npo}}$ as a meaningful mechanism. Indeed, when $w_{s_{\mathrm{u}}}^{\mathrm{npo}}$ is applied token-wise, which allows for more granular control over the unlearning process, the unlearning procedures are notably more effective. Formally, we consider the objective

$$\mathbb{E}_{s_{\mathrm{u}} \sim \mathcal{D}_{\mathrm{u}}} \sum_{i=2}^{|s_{\mathrm{u}}|} w_{s_{\mathrm{u}}, i}^{\mathrm{tnpo}} \log p(s_{\mathrm{u}}^i | s_{\mathrm{u}}^{<i}; \boldsymbol{\theta}), \tag{6}$$

where $w_{s_{\mathrm{u}}, i}^{\mathrm{tnpo}} = \frac{2p(s_{\mathrm{u}}^i | s_{\mathrm{u}}^{<i}; \boldsymbol{\theta})^{\beta}}{p(s_{\mathrm{u}}^i | s_{\mathrm{u}}^{<i}; \boldsymbol{\theta})^{\beta} + p(s_{\mathrm{u}}^i | s_{\mathrm{u}}^{<i}; \boldsymbol{\theta}_{\mathrm{o}})^{\beta}}$ generalizes the weighting mechanism of NPO for tokens. We refer to equation 6 as token-wise NPO (TNPO) and show its G-effect in Figure 5(c). Therein, we observe that the unlearning G-effect exhibits sufficiently large negative values while the retaining G-effect is overall close-to-zero. It underscores the efficacy of $w_{s_{\mathrm{u}}, i}^{\mathrm{tnpo}}$ in properly prioritizing certain tokens during unlearning, thus achieving unlearning efficacy. Please refer to Appendix D.2 for more discussions about TNPO, as well as its further improved version named WTNPO.

## 4.3 MORE OBJECTIVES

We also examine two other unlearning objectives that do not fall into the variants of GA.

**Preference Optimization (PO)** (Maini et al., 2024) overwrites LLMs with new outcomes instead of erasing old ones. Given some prefix $s^{<i}$ and the new suffix $s_{\mathrm{po}}$, the PO unlearning objective is given by

$$\mathbb{E}_{s_{\mathrm{u}} \sim \mathcal{D}_{\mathrm{u}}} - \log p(s_{\mathrm{po}} | s^{<i}; \boldsymbol{\theta}). \tag{7}$$

It is particular suitable for LLMs fine-tuned for question answering, where $s^{<i}$ is the original question and $s_{\mathrm{po}}$ is the new answer. We show its G-effect in Figure 6. Unfortunately, we note that the PO may not be suitable for LLM unlearning: Its validity in erasing targeted knowledge is limited to the early phases of model updating. Then, PO may even inadvertently facilitate the knowledge relearning.

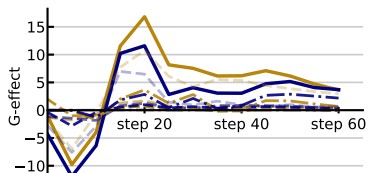

Figure 6: **The G-effect for PO.** The **legends** for the G-effect are in Figure 2. The **horizontal axis** denotes the unlearning step and the **vertical axis** is the values of the G-effect.

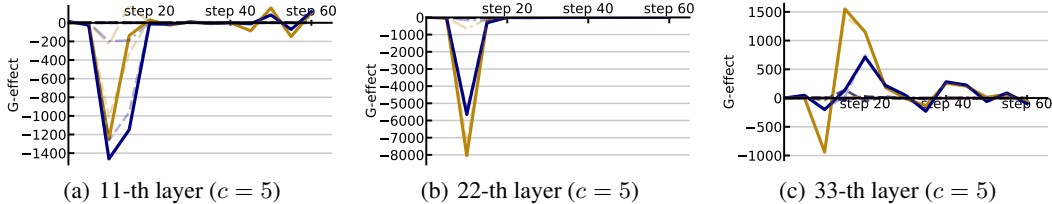

(a) 11-th layer ($c = 5$)      (b) 22-th layer ($c = 5$)      (c) 33-th layer ($c = 5$)

Figure 7: **The G-effect for RMU**. The **legends** for the G-effect are summarized in Figure 2. The **horizontal axis** denotes the unlearning step and the **vertical axis** indicates the values of the G-effect.

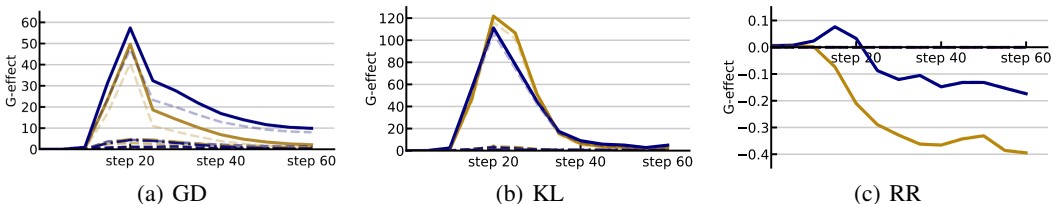

(a) GD      (b) KL      (c) RR

Figure 8: **The G-effect for Regularization**. The **legends** for G-effect are in Figure 2. The **horizontal axis** denotes the unlearning step and the **vertical axis** indicates the values of the G-effect.

**Representation Misdirection for Unlearning (RMU)** (Li et al., 2024) implements unlearning by perturbing model representation. Denote the embedding features by $\phi(s; \theta)$, RMU is articulated as

$$\mathbb{E}_{s_u \sim \mathcal{D}_u} \frac{1}{|s| - 1} \sum_{i=1}^{|s|-1} ||\phi(s^{<i}; \theta) - c \cdot \boldsymbol{u}||_2^2, \tag{8}$$

where $\boldsymbol{u}$ is a random vector with elements sampled uniformly from $[0, 1)$ and $c$ is a scaling hyperparameter. We adopt outputs for 11-th, 22-th, and 33-th (before unembedding) layers as $\phi(s; \theta)$, and their G-effect is summarized in Figure 7. We notice that its performance is very sensitive to different choices of $\phi(s; \theta)$, where middle (22-th) layers seem to be a better choice than shallow (11-th) and deep (33-th) layers. In Appendix C.3, we further show that RMU is also sensitive to varying $c$, where a wrong configuration may be even completely contrary to the goal of unlearning.

Moreover, we observe that the improvements on unlearning come at similar costs in terms of impairing the general utility, a phenomenon reminiscent of the challenges faced with the vanilla GA. It can also be considered as a scenario of excessive unlearning, where the magnitudes of parameter updates are too large, thus failing to preserve essential knowledge for common data. Given its current limitations, more explorations are needed to advance unlearning through embedding perturbation.

## 4.4 REGULARIZATION

Although we have identified several promising objectives, the retaining G-effect overall remains negative. It indicates that there are still adverse effects on the common model integrity. We also want to note that, while some of the magnitudes are steadily small, e.g., for the retaining G-effect of TNPO in Figure 5(c), their accumulation across steps will still have a notable impact. A wide-accepted strategy to improve retention is by regularization, involving a set of additional common data to maintain the original model responses. In this section, we explore 3 representative regularization terms, named gradient difference (GD) (Yao et al., 2023b), KL divergence (KL) (Maini et al., 2024), and representation retention (RR) (Li et al., 2024) (cf., Appendix E). We choose NPO as the unlearning objective, computing the G-effect for various regularization terms. The results are summarized in Figure 8. Overall, our observations indicate that RR does not serve for effective regularization due to its unstable G-effect behaviors. In contrast, both GD and KL effectively facilitate knowledge retention. However, the strength of the G-effect associated with KL surpasses that of GD, leading us to suggest KL as a default choice of regularization for retention.

## 5 EVALUATIONS

In this section, we further benchmark aforementioned unlearning objectives on the TOFU unlearning datasets (Maini et al., 2024), focusing on the removal of fictitious author profiles from LLMs fine-tuned on them. Comprising a series of question-answer pairs about author profiles, the TOFU dataset is further separated into targeted and non-targeted parts, thereby providing an intuitive platform to effectively evaluate the impact of various unlearning methods.

We test two popular LLMs: Phi-1.5 (Li et al., 2023) and Llama-2-7B (Touvron et al., 2023b), under three ratios—1%, 5%, and 10%—of targeted data. For hyper-parameter tuning, we follow the unlearning with control (UWC) framework (Wang et al., 2025), which surpasses the challenges of trade-offs between removal and retention. Please refer to Appendix B for additional details on the experimental setups and Appendix G for hyper-parameter configurations.

**Configurations.** We default to apply the following settings: the AdamW optimizer (Loshchilov & Hutter, 2017), a batch size of 16, a maximal gradient norm of 1, and the (un)learning rate of $2e^{-5}$ for Phi-1.5 and $1e^{-5}$ for Llama-2-7b with linear warm-up for the first epoch. Each method is executed over a total of 5 epochs. Moreover, the model-specific hyper-parameters after fine-tuning are as follows: For the 1% and 5% setups, we set $\alpha = 5$ for WGA; $\beta = 0.5$ for NPO; $\beta = 4$ for TNPO; $\alpha = 1.5$ and $\beta = 5$ for WTNPO. For the 10% setup, we set $\alpha = 7$ for WGA; $\beta = 0.5$ for NPO; $\beta = 5$ for TNPO; $\alpha = 1.5$ and $\beta = 7$ for WTNPO. For RMU, we set the 9-th layer with $c = 4$ for Phi-1.5 and the 21-th layer with $c = 2$ for Llama-2-7B. Moreover, our experiments are conducted on computation nodes equipped with NVIDIA-A100-80GB GPUs and Intel(R) Xeon(R) Gold 6248R CPUs. The systems utilize Transformers version 4.42.4 and CUDA version 12.1.

**Evaluation Metrics.** We adopt the suggest evaluation metrics from (Maini et al., 2024), specifically forget quality (FQ) for unlearning and model utility (MU) for retention. FQ evaluates model performance by jointly examining output quality, confidence, and truth ratio, fully reflecting the common model integrity. MU produces $p$-values to assess the change of model outputs between the gold standard model, which is trained from scratch without targeted data, and the unlearned model. We utilize the log-scale for these $p$-value to make the results more readable. We aim for high values in both FQ and MU. We further report the ES scores (Wang et al., 2025), which more directly quantify the extent of knowledge parameterized within models, potentially making them more effective metrics than FQ and MU. The ES scores can be calculated for either targeted data or non-targeted data, thereby reflecting model performance of removal and retention, respectively. Notably, the ES scores are available in two variants: ES-exact, which is used for original data to reflect direct parametrization, and ES-perturb, which applies to their rephrasing to reflect generalization. Overall, the ES scores should be high for retention and low for removal.

**Analysis.** The results are summarized in Table 1, where we use KL regularization to stabilize the unlearning procedures. Among previous methods, PO is identified as the least attractive choice, which may even inadvertently maintain data that ought to be unlearned, corroborating our observations from the G-effect analysis. Conversely, GA is most effective in removing targeted data but at the expense of compromising model integrity. Both NPO and RMU offer a better balance between data removal and retention, with NPO overall outperforming RMU (except for 10% unlearning with Llama-2-7B). This can be attributed to the more stable G-effect of NPO over that of RMU.

For our newly explored methods, WGA remarkably overcomes the drawbacks of GA, particularly in mitigating excessive unlearning, meanwhile maintaining its strong capability for the removal of targeted data. Additionally, both TNPO and WTNPO improve upon NPO by not only enhancing unlearning performance but also excelling in retaining common performance. WTNPO typically outperforms TNPO as it further mitigates the potential issues of excessive unlearning observed in TNPO. Overall, when comparing methods across different unlearning setups and models, WGA and WTNPO stand out as the most effective, underscoring the crucial role of loss weighting in the unlearning process for LLMs. However, we recommend the default use of WGA, as it requires tuning only one hyper-parameter and generally perform well, recognized as effective for LLM unlearning.

It is worth noting that most of the above analyses are based on our results measured by ES. When it comes to MU and FQ, our refined methods can still outperform it previous counterparts in most cases, suggesting that the goals of full removal and influence removal are generally aligned. However, there is also an abnormal case of 5% unlearning with Llama-2-7B, where the MU scores of

Table 1: **Comparison between Unlearning Objectives** on TOFU with **KL regularization** to stabilize unlearning. ↓ / ↑ indicate smaller / larger values are preferable. The log scale is used for FQ to improve readability. The top two results are in bold font for each unlearning setup.

| LLM | | Phi-1.5 | | | | | | Llama-2-7B | | | | | |
| setup | method | ES-exact | | ES-perturb | | MU ↑ | FQ ↑ | ES-exact | | ES-perturb | | MU ↑ | FQ ↑ |
| | | retain ↑ | unlearn ↓ | retain ↑ | unlearn ↓ | | | retain ↑ | unlearn ↓ | retain ↑ | unlearn ↓ | | |
| | before unlearning | 0.4433 | 0.5969 | 0.2115 | 0.1605 | 0.5232 | -5.8031 | 0.8277 | 0.8039 | 0.5302 | 0.4001 | 0.6345 | -7.5930 |
| | GA | 0.1103 | 0.0530 | 0.0850 | 0.0828 | 0.3799 | -0.5471 | 0.4298 | **0.0570** | 0.2692 | **0.0422** | 0.5378 | -0.5471 |
| | PO | **0.3667** | 0.8472 | 0.1622 | 0.3658 | 0.5112 | -4.2474 | **0.7508** | 0.8359 | **0.4724** | 0.5259 | 0.6246 | -5.8031 |
| | WGA | 0.3629 | **0.0344** | **0.1857** | **0.0282** | **0.5191** | -0.5471 | 0.6701 | 0.0818 | 0.3814 | 0.0601 | **0.6541** | **-0.0847** |
| 1% | NPO | 0.2727 | 0.0916 | 0.1125 | 0.0733 | 0.4845 | -2.9162 | 0.4757 | 0.1216 | 0.3890 | 0.0905 | 0.6243 | -1.3254 |
| | TNPO | 0.3351 | 0.0365 | 0.1239 | 0.0412 | 0.4991 | **-0.0847** | 0.5168 | 0.0337 | **0.4304** | 0.0337 | **0.6495** | -0.0847 |
| | WTNPO | **0.4117** | 0.0285 | **0.1969** | 0.0255 | 0.5126 | -0.2667 | **0.6701** | 0.0807 | 0.3734 | 0.0601 | 0.6453 | -0.0847 |
| | RMU | 0.2397 | 0.0850 | 0.1539 | 0.0567 | 0.4349 | -0.5471 | 0.2397 | 0.0850 | 0.1539 | 0.0567 | 0.5298 | -1.3254 |
| | before unlearning | 0.4433 | 0.5619 | 0.2115 | 0.2374 | 0.5232 | -29.6514 | 0.8277 | 0.7735 | 0.5302 | 0.4126 | 0.6345 | -32.1330 |
| | GA | 0.0000 | **0.0000** | 0.0000 | **0.0000** | 0.0000 | -11.4040 | 0.0300 | **0.0000** | 0.0206 | **0.0000** | 0.0000 | **-12.4230** |
| | PO | **0.2646** | 0.7986 | **0.1639** | 0.4925 | **0.5118** | -26.5061 | **0.5572** | 0.8437 | 0.3652 | 0.4933 | **0.6466** | -28.8476 |
| | WGA | **0.2980** | 0.0179 | **0.1645** | 0.0199 | **0.5108** | -1.3076 | 0.4709 | 0.0053 | **0.3982** | 0.0050 | **0.6438** | -16.3271 |
| 5% | NPO | 0.0876 | 0.1267 | 0.0876 | 0.0609 | 0.3841 | -7.7503 | 0.1747 | 0.0764 | 0.1273 | 0.0802 | 0.5285 | **-9.9550** |
| | TNPO | 0.1695 | 0.0126 | 0.0803 | 0.0038 | 0.4673 | -2.1867 | **0.5017** | 0.0160 | 0.3495 | 0.0099 | 0.6348 | -32.1330 |
| | WTNPO | 0.2185 | 0.0179 | 0.1281 | 0.0188 | 0.5084 | -3.2299 | 0.4595 | 0.0061 | **0.3989** | 0.0040 | 0.6410 | -21.4429 |
| | RMU | 0.2162 | **0.0000** | 0.1299 | **0.0000** | 0.2744 | **-1.9514** | 0.1262 | **0.0000** | 0.1299 | **0.0000** | 0.5801 | -21.4429 |
| | before unlearning | 0.4433 | 0.4799 | 0.2115 | 0.1843 | 0.5232 | -39.0042 | 0.8277 | 0.8307 | 0.5302 | 0.3099 | 0.6345 | -44.4594 |
| | GA | 0.0000 | **0.0000** | 0.0000 | **0.0000** | 0.0000 | -45.2697 | 0.0000 | **0.0000** | 0.0000 | **0.0000** | 0.0000 | -20.8637 |
| | PO | 0.3222 | 0.7321 | 0.1406 | 0.2667 | 0.5078 | -38.2556 | **0.5572** | 0.8437 | **0.3777** | 0.4305 | **0.6240** | -39.7604 |
| | WGA | **0.3466** | **0.0000** | 0.1651 | **0.0000** | **0.5183** | -9.0636 | **0.6642** | 0.0287 | **0.4289** | 0.0123 | 0.6235 | -24.8591 |
| 10% | NPO | 0.0859 | 0.0955 | 0.0716 | 0.0710 | 0.3878 | -10.5721 | 0.1296 | 0.1388 | 0.1085 | 0.1440 | 0.5055 | **-12.1912** |
| | TNPO | 0.2085 | 0.0163 | 0.0991 | 0.0134 | 0.5009 | -7.6651 | 0.4531 | **0.0192** | 0.2690 | 0.0165 | **0.6381** | **-13.4785** |
| | WTNPO | 0.2969 | 0.0048 | **0.1862** | 0.0105 | **0.5123** | **-7.0070** | 0.4997 | 0.0278 | 0.3246 | 0.0174 | 0.6236 | -26.6801 |
| | RMU | 0.0317 | 0.0541 | 0.0357 | 0.0632 | 0.3163 | **-7.0070** | 0.2580 | 0.0194 | 0.2017 | 0.0174 | 0.5930 | -16.7271 |

our proposed methods remain high, while the FQ for NPO notably surpasses our methods. Given that our methods better preserve retention, we believe they offer more practical applicability than NPO. Nonetheless, the disagreement between ES and FQ metrics still deserves our further attention, necessitating deeper explorations into the reliability of these metrics and determining which goals, either full removal or influence removal, are better suitable for LLM unlearning.

# 6 CONCLUSIONS

LLM unlearning aims to eliminate unwanted knowledge while preserving the overall model integrity. This paper particularly focuses on understanding the mechanisms behind various unlearning objectives, based on our proposed evaluation tool named the G-effect. Our findings suggest that GA-based unlearning objectives remain to be promising, but we need to mitigate the risk of excessive unlearning and the potential harm on model integrity. We further introduce advanced unlearning objectives, such as WGA and TNPO, that set as new state-of-the-arts within the community.

**Drawbacks of the G-effect.** As shown in Appendix A, to motivate the G-effect, we assume that singular values of the matrix $A$ have low variance. However, it may neglect important model properties with unlearning smoothness. Refining the G-effect to better incorporate $A$ could make the evaluation scheme more accurate and insightful. However, its computation requires estimating the Hessian matrix, a tedious process that needs approximation (Singh & Alistarh, 2020). Also, using NLL as the risk metric to define $\mathcal{R}$ may not be the optimal choice, given that model likelihood can be misleading to characterize the knowledge parametrization (Duan et al., 2024).

**Promising Directions.** Although we achieve several powerful unlearning objectives, their practical implementations still require regularization for retention; otherwise, the common model integrity will be compromised. Thus, further enhancements in unlearning objectives are anticipated, such as devising improved weighting mechanisms (Ren et al., 2018) and exploring robust representation methods. Moreover, on the optimization side, techniques like sub-model updating (Yao et al., 2024) and layer-adapted updating (Schaul et al., 2013) can be promising to further advance the research area. On the data-oriented side, unlearning methods that incorporate filtering or prompting to foster improved G-effect behaviors also be intriguing, while currently are not covered. On the evaluation side, it is also intriguing to explore the G-effect using advanced metrics, such as risk aversion (Yang et al., 2023) and effective robustness (Taori et al., 2020), to better quantify the impacts of unlearning.

## ETHIC STATEMENT AND REPRODUCIBILITY

Unlearning mechanisms are crucial for LLMs, as they facilitate the removal of sensitive data that may lead to copyright and privacy violations, significantly boosting the overall confidentiality of models. By identifying and eradicating privacy risks, we fulfill the ethical obligation to respect individual privacy. Adapting LLMs to prevent the replication of sensitive information further aligns with the principles of responsible data use. In essence, the process of unlearning in LLMs enhances societal well-being by improving both the safety and legal compliance of these technologies. Additionally, we benefit the research community by introducing a new analytical tool, the G-effect, designed to measures the comprehensive impacts of unlearning objectives on LLMs. This tool facilitates a detailed analysis of existing unlearning objectives and offers the potential to evaluate the efficacy of a broad range of new methods. The deployment of such a toolkit contributes to open inquiry and could encourage collaboration and further studies in this pivotal area. For reproducibility, we have detailed the configurations, hyper-parameter setups, and hardware specifications. The code is publicly available at: https://github.com/tmlr-group/G-effect.

## ACKNOWLEDGMENTS

JPZ is supported by grant from the Natural Sciences and Engineering Research Council of Canada (NSERC) (567916). The authors also would like to express their sincere gratitude to the anonymous reviewers and the area chairs for their thorough review and constructive feedback. Their insightful comments and valuable suggestions have significantly enhanced the quality and clarity of this manuscript. We deeply appreciate their time and effort in helping us improve our work.

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

## A  A FORMAL MOTIVATION FOR THE G-EFFECT

**Overview.** To formalize our key concept of the G-effect, we begin by examining the impacts of an unlearning objective $\mathcal{L}_{\mathrm{u}}$ on model parameters $\boldsymbol{\theta}$ with mini-batch gradient updates. We simplify the expression for the unlearned parameters $\boldsymbol{\theta}_{\mathrm{u}}$ such that it is independent of the intermediate parameter stages, cf. equation 10. Then, substituting the approximation of $\boldsymbol{\theta}_{\mathrm{u}}$ into $\mathcal{R}(\mathcal{D}; \boldsymbol{\theta}_{\mathrm{u}})$, we observe that the change in model performance can be primarily characterized by the dot product of gradients between the risk metric $\mathcal{R}$ and the unlearning objective $\mathcal{L}_{\mathrm{u}}$, cf. equation 13. Its generalized version leads to our G-effect in Definition 1. Please see below for a formal description.

Without loss of generality, we consider an objective $\mathcal{L}_{\mathrm{u}}$ and a sequence of mini-batches $\{S_{\mathrm{u}}^{(t)}\}_T$ that are randomly drawn from $\mathcal{D}_{\mathrm{u}}$. These batches are sequentially fed in LLMs to minimize $\mathcal{L}_{\mathrm{u}}$. Specifically, for the $t$-th iteration, the model parameters are updated from $\boldsymbol{\theta}^{(t-1)}$ to $\boldsymbol{\theta}^{(t)}$ following

$$\boldsymbol{\theta}^{(t)} \leftarrow \boldsymbol{\theta}^{(t-1)} - \mathtt{lr}\nabla_{\boldsymbol{\theta}}\mathcal{L}_{\mathrm{u}}(S_{\mathrm{u}}^{(t-1)}; \boldsymbol{\theta}^{(t-1)}), \tag{9}$$

with $\mathtt{lr}$ the (un)learning rate. To understand the impacts of equation 9 on model parameters and subsequent effects on model performance, we further simplify the accumulative effects of gradient updates: When assuming $\mathtt{lr}$ is small and each point in $\mathcal{D}_{\mathrm{u}}$ occurs $k$ times within $\{S_{\mathrm{u}}^{(t)}\}_T$, we can approximate the final parameters after unlearning as

$$\boldsymbol{\theta}^{(T)} \approx \boldsymbol{\theta}^{(0)} - \mathtt{lr}kA\nabla_{\boldsymbol{\theta}}\mathcal{L}_{\mathrm{u}}(\mathcal{D}_{\mathrm{u}}; \boldsymbol{\theta}^{(0)}). \tag{10}$$

$A$ is a symmetric matrix associated with model smoothness and orders of mini-batches. Also, $A$ will converge to the identity matrix when $\alpha$ approaches 0. Please see below for the detailed derivations.

**Proposition 1.** *Given the original parameters $\boldsymbol{\theta}^{(0)}$ and the objective $\mathcal{L}$. During the stochastic gradient updates, the model will receive a sequence of $T$ random mini-batches of samples $\{S^{(t)}\}_T$, which will be fed into the model orderly via $\boldsymbol{\theta}^{(t)} \leftarrow \boldsymbol{\theta}^{(t-1)} - \mathtt{lr}\nabla_{\boldsymbol{\theta}}\mathcal{L}(S^{(t-1)}; \boldsymbol{\theta}^{(t-1)})$. With a small $\mathtt{lr}$, we can approximate the final parameters $\boldsymbol{\theta}^{(T)}$ after stochastic gradient updates as*

$$\boldsymbol{\theta}^{(T)} \approx \boldsymbol{\theta}^{(0)} - \mathtt{lr}A\sum_{t=0}^{T-1}\nabla_{\boldsymbol{\theta}}\mathcal{L}(S^{(t)}; \boldsymbol{\theta}^{(0)}), \tag{11}$$

*where $A = I - \mathtt{lr}\sum_{t=1}^{T-1}\nabla_{\boldsymbol{\theta}}^2\mathcal{L}(S^{(t)}; \boldsymbol{\theta}^{(0)})$ and $I$ is the identify matrix. The matrix $A$ characterizes the smoothness with respect to $\mathcal{L}$, the impacts of $\mathtt{lr}$, and the influence of ordering within $\{S^{(t)}\}_T$.*

*Proof.* We begin by showing parameter changes after two consecutive steps, i.e., from the $t$-th to the $t+2$-th step. Substituting $\boldsymbol{\theta}^{(t+1)} \leftarrow \boldsymbol{\theta}^{(t)} - \mathtt{lr}\nabla_{\boldsymbol{\theta}}\mathcal{L}(S^{(t)}; \boldsymbol{\theta}^{(t)})$ into $\boldsymbol{\theta}^{(t+2)} \leftarrow \boldsymbol{\theta}^{(t+1)} - \mathtt{lr}\nabla_{\boldsymbol{\theta}}\mathcal{L}(S^{(t+1)}; \boldsymbol{\theta}^{(t+1)})$, we can express the parameter update at $t+2$-th step in terms of $\boldsymbol{\theta}^{(t)}$ as

$$\boldsymbol{\theta}^{(t+2)} \leftarrow \boldsymbol{\theta}^{(t)} - \mathtt{lr}\nabla_{\boldsymbol{\theta}}\mathcal{L}(S^{(t)}; \boldsymbol{\theta}^{(t)}) - \mathtt{lr}\nabla_{\boldsymbol{\theta}}\mathcal{L}(S^{(t+1)}; \boldsymbol{\theta}^{(t)} - \mathtt{lr}\nabla_{\boldsymbol{\theta}}\mathcal{L}(S^{(t)}; \boldsymbol{\theta}^{(t)})).$$

When further applying the first-order Taylor approximation around $\boldsymbol{\theta}^{(t)}$, we have

$$\boldsymbol{\theta}^{(t+2)} \approx \boldsymbol{\theta}^{(t)} - \mathtt{lr}\big[\nabla_{\boldsymbol{\theta}}\mathcal{L}(S^{(t)}; \boldsymbol{\theta}^{(t)}) + \nabla_{\boldsymbol{\theta}}\mathcal{L}(S^{(t+1)}; \boldsymbol{\theta}^{(t)})$$
$$+ \nabla_{\boldsymbol{\theta}}^2\mathcal{L}(S^{(t+1)}; \boldsymbol{\theta}^{(t)})(-\mathtt{lr}\nabla_{\boldsymbol{\theta}}\mathcal{L}(S^{(t+1)}; \boldsymbol{\theta}^{(t)}))\big].$$

The above formulation can be expanded to incorporating more updating steps: Considering the accumulations of gradient updating from the 0-th to $T$-th steps, we have

$$\boldsymbol{\theta}^{(T)} \approx \boldsymbol{\theta}^{(0)} - \mathtt{lr}\sum_{t=0}^{T-1}\nabla_{\boldsymbol{\theta}}\mathcal{L}(S^{(t)}; \boldsymbol{\theta}^{(0)}) + \sum_{t=1}^{T-1}\psi^{(t)}$$

where $\psi^{(t)} = -\mathtt{lr}\nabla_{\boldsymbol{\theta}}^2\mathcal{L}(S^{(t)}; \boldsymbol{\theta}^{(0)})\big(-\mathtt{lr}\sum_{t'=0}^{T-1}\nabla_{\boldsymbol{\theta}}\mathcal{L}(S^{(t')}; \boldsymbol{\theta}^{(0)}) + \sum_{t'=0}^{T-1}\psi^{(t')}\big)$ and $\psi^{(0)} = 0$. When the learning rate $\mathtt{lr}$ is small (e.g., notably less than 1), the influence of higher-order terms with respect to $\mathtt{lr}$ diminishes. Therefore, we can further simplify the formulation of $\psi^{(t)}$ as $\psi^{(t)} \approx \mathtt{lr}^2\nabla_{\boldsymbol{\theta}}^2\mathcal{L}(S^{(t)}; \boldsymbol{\theta}^{(0)})\sum_{t'=0}^{T-1}\nabla_{\boldsymbol{\theta}}\mathcal{L}(S^{(t')}; \boldsymbol{\theta}^{(0)})$. Substituting the approximation of $\psi^{(t)}$ back into the formulation of $\boldsymbol{\theta}^{(T)}$, we complete the proof. The analysis is motivated by (Thudi et al., 2022). $\square$

**What Ensures a Good Unlearning Objective?** We go beyond equation 10 and substitute it into $\mathcal{R}(\mathcal{D}; \boldsymbol{\theta}_{\mathrm{u}})$. When the difference between the unlearned model $\boldsymbol{\theta}_{\mathrm{u}}$ and the original model $\boldsymbol{\theta}_{\mathrm{o}}$ is acceptably small, we can apply the first-order Taylor expansion upon $\mathcal{R}(\mathcal{D}; \boldsymbol{\theta}_{\mathrm{u}})$, which can help us to simplify the formulation of the performance change by

$$\mathcal{R}(\mathcal{D}; \boldsymbol{\theta}_{\mathrm{u}}) - \mathcal{R}(\mathcal{D}; \boldsymbol{\theta}_{\mathrm{o}}) \approx -\mathtt{lrk}\nabla_{\boldsymbol{\theta}}\mathcal{R}(\mathcal{D}; \boldsymbol{\theta}_{\mathrm{o}})^{\top}A\nabla_{\boldsymbol{\theta}}\mathcal{L}_{\mathrm{u}}(\mathcal{D}_{\mathrm{u}}; \boldsymbol{\theta}_{\mathrm{o}}). \tag{12}$$

One step further, by eigenvalue decomposition, $\nabla_{\boldsymbol{\theta}}\mathcal{R}(\mathcal{D}; \boldsymbol{\theta}_{\mathrm{o}})^{\top}A\nabla_{\boldsymbol{\theta}}\mathcal{L}_{\mathrm{u}}(\mathcal{D}_{\mathrm{u}}; \boldsymbol{\theta}_{\mathrm{o}})$ is lower and upper bounded by $\lambda_{\min}||\nabla_{\boldsymbol{\theta}}\mathcal{R}(\mathcal{D}; \boldsymbol{\theta}_{\mathrm{o}})||\,||\nabla_{\boldsymbol{\theta}}\mathcal{L}_{\mathrm{u}}(\mathcal{D}_{\mathrm{u}}; \boldsymbol{\theta}_{\mathrm{o}})||$ and $\lambda_{\max}||\nabla_{\boldsymbol{\theta}}\mathcal{R}(\mathcal{D}; \boldsymbol{\theta}_{\mathrm{o}})||\,||\nabla_{\boldsymbol{\theta}}\mathcal{L}_{\mathrm{u}}(\mathcal{D}_{\mathrm{u}}; \boldsymbol{\theta}_{\mathrm{o}})||$. $\lambda_{\min}$ and $\lambda_{\max}$ are the minimal and the maximal eigenvalues of $A$. Furthermore, when $\alpha$ is small, the difference between $\lambda_{\min}$ and $\lambda_{\max}$ is negligible, thus existing $\lambda \in [\lambda_{\min}, \lambda_{\max}]$ such that $\lambda\nabla_{\boldsymbol{\theta}}\mathcal{R}(\mathcal{D}; \boldsymbol{\theta}_{\mathrm{o}})^{\top}\nabla_{\boldsymbol{\theta}}\mathcal{L}_{\mathrm{u}}(\mathcal{D}_{\mathrm{u}}; \boldsymbol{\theta}_{\mathrm{o}})$ is a good approximation of $\nabla_{\boldsymbol{\theta}}\mathcal{R}(\mathcal{D}; \boldsymbol{\theta}_{\mathrm{o}})^{\top}A\nabla_{\boldsymbol{\theta}}\mathcal{L}_{\mathrm{u}}(\mathcal{D}_{\mathrm{u}}; \boldsymbol{\theta}_{\mathrm{o}})$. Thus,

$$\mathcal{R}(\mathcal{D}; \boldsymbol{\theta}_{\mathrm{u}}) - \mathcal{R}(\mathcal{D}; \boldsymbol{\theta}_{\mathrm{o}}) \approx -\mathtt{lrk}\lambda\nabla_{\boldsymbol{\theta}}\mathcal{R}(\mathcal{D}; \boldsymbol{\theta}_{\mathrm{o}})^{\top}\nabla_{\boldsymbol{\theta}}\mathcal{L}_{\mathrm{u}}(\mathcal{D}_{\mathrm{u}}; \boldsymbol{\theta}_{\mathrm{o}}). \tag{13}$$

Moreover, when taking $\mathtt{lrk}\lambda$ as a constant, we conclude that the dot product between $\nabla_{\boldsymbol{\theta}}\mathcal{R}(\mathcal{D}; \boldsymbol{\theta}_{\mathrm{o}})$ and $\nabla_{\boldsymbol{\theta}}\mathcal{L}_{\mathrm{u}}(\mathcal{D}_{\mathrm{u}}; \boldsymbol{\theta}_{\mathrm{o}})$ quantifies the impacts of $\mathcal{L}_{\mathrm{u}}$ on model performance measured by $\mathcal{R}(\mathcal{D}; \boldsymbol{\theta}_{\mathrm{u}})$. Specifically, echoing the general goal of LLM unlearning in Section 2, we can claim that a good unlearning objective should meet the following two conditions jointly:

- **Removal.** We define $e_{\mathrm{u}} = \nabla_{\boldsymbol{\theta}}\mathcal{R}(\mathcal{D}_{\mathrm{u}}; \boldsymbol{\theta}_{\mathrm{o}})^{\top}\nabla_{\boldsymbol{\theta}}\mathcal{L}_{\mathrm{u}}(\mathcal{D}_{\mathrm{u}}; \boldsymbol{\theta}_{\mathrm{o}})$, which should be much smaller than 0. It ensures the sufficient removal for targeted data, i.e., $\mathcal{R}(\mathcal{D}_{\mathrm{u}}; \boldsymbol{\theta}_{\mathrm{u}}) \gg \mathcal{R}(\mathcal{D}_{\mathrm{u}}; \boldsymbol{\theta}_{\mathrm{o}})$.

- **Retention.** We define $e_{\mathrm{r}} = \nabla_{\boldsymbol{\theta}}\mathcal{R}(\mathcal{D}\backslash\mathcal{D}_{\mathrm{u}}; \boldsymbol{\theta}_{\mathrm{o}})^{\top}\nabla_{\boldsymbol{\theta}}\mathcal{L}_{\mathrm{u}}(\mathcal{D}_{\mathrm{u}}; \boldsymbol{\theta}_{\mathrm{o}})$, which should be greater than or equal to 0, ensuring performance on common data will not reduce, i.e., $\mathcal{R}(\mathcal{D}_{\mathrm{t}}\backslash\mathcal{D}_{\mathrm{u}}; \boldsymbol{\theta}_{\mathrm{u}}) \leq \mathcal{R}(\mathcal{D}_{\mathrm{t}}\backslash\mathcal{D}_{\mathrm{u}}; \boldsymbol{\theta}_{\mathrm{o}})$.

Although $e_{\mathrm{u}}$ and $e_{\mathrm{r}}$ can anticipate performance changes following a sequence of gradient updates based on $\mathcal{L}_{\mathrm{u}}$, their validity relies on our assumption that the difference between $\boldsymbol{\theta}_{\mathrm{o}}$ and $\boldsymbol{\theta}_{\mathrm{u}}$ remains small. Otherwise, the first-order Taylor approximation may introduce large errors. Therefore, we generalize $e_{\mathrm{u}}$ and $e_{\mathrm{r}}$ to make its expression depend on particular updating steps, thereby leading to our definition of the G-effect in Section 3.

**Connection with Previous Works.** The G-effect resembles the formulation of influence functions (Koh & Liang, 2017) when assessing the inherent influence of training data on test data via their respective gradients. However, there are also substantial differences between them. First, the G-effect primarily explores the roles of objectives in shaping performance, whereas influence functions focus on the impact of individual data points or features on performance. Moreover, the G-effect is derived from the first-order approximation of the SGD dynamics, while influence functions are computed by the linearization of optimal solutions and are based on the average marginal contributions, thus serving completely different purposes. Also, while we use NLL as the risk metric $\mathcal{R}$, exploring other choices (Wang et al., 2022; Taori et al., 2020; Agarwal & Zhang, 2022) has the potential to offer more rigorous and reliable ways of measurement.

**Further Discussions.** In Figure 1, the gradient behaviors are divided into four distinct regions:

- **Region 1** (Blue Region not intersecting with Red): Objectives with gradients in this region excel at retention but are not effective at unlearning.

- **Region 2** (Intersection of Red and Blue Regions): Objectives here are effective at unlearning but struggle with retention.

- **Region 3** (Red Region not intersecting with Blue): Objectives in this region demonstrate proficiency in both unlearning and retention.

- **Region 4** (White Region): Objectives here are ineffective at both unlearning and retention.

Overall, Regions 1 and 3 exhibit a trade-off between unlearning and retention. Region 2 contains ideal objectives for unlearning, whereas Region 4 is unsuitable for unlearning objectives.

Our experiments further substantiate our claims to be correct. For example, when comparing the unlearning G-effects of NPO and GA in Figures 3-4, GA exhibits a greater magnitude compared to NPO, indicating a stronger capability of GA in removing targeted data, further evidenced by the results in Table 1. Similarly, when comparing the retaining G-effects between GA and WGA in Figure 3, the effect magnitude for WGA is notably smaller than that of GA, demonstrating the superior ability of NPO to maintain original performance, also detailed in Table 1.

## B EXPERIMENTAL SETUPS

We provide more detailed information about our experimental setups.

### B.1 TOFU BENCHMARKS

Our evaluations are based on TOFU fictitious unlearning (Maini et al., 2024), focusing on LLMs fine-tuned with a series of fictitious authors profiles. These profiles were created by prompting GPT-4 (Achiam et al., 2023), which has been filtered to avoid the occurrence of any real author profile, thus mitigating the inadvertent impacts of other unrelated variates. For each fictitious profile, TOFU crafted 20 question-answer pairs that can be used for fine-tuning, along with their paraphrased versions for evaluations.

The pre-trained LLMs are further fine-tuned on such question-answer pairs, where we consider two popular LLMs, i.e., Phi-1.5 (Li et al., 2023) and Llama-2-7B (Touvron et al., 2023a) of their question-answering versions. For the unlearning setups, the original TOFU data are separated into targeted and non-targeted parts, of which the adopted proportions are 1:99 (1% unlearning), 5:95 (5% unlearning), and 10:90 (10% unlearning). Moreover, we separate 400 non-targeted data that are not involved during the unlearning procedure for evaluations, reflecting real-world situations where it is not feasible to go through all non-targeted data during the unlearning process.

### B.2 UWC HYPER-PARAMETER TUNING

We need to ensure common model integrity when conducting unlearning, but these two goals are often conflicting, failing to align with their Pareto frontiers (Maini et al., 2024). It leads to the dilemma when comparing across unlearned models: Some models may excel at unlearning while others better maintain the overall integrity, making it hard to judge which one is overall better.

The unlearning with control (UWC) (Wang et al., 2025) framework offers an interesting solution. It allows for the adjustment of model parameters post-unlearning by mixing them with that before unlearning. By proper control of this mixture, different unlearned models can achieve comparable levels of common performance with minimal compromise on their extent of unlearning. Thereafter, we can compare between models by focusing on assessing their unlearning performance. During hyper-parameter tuning, we adopt the KL regularization to stabilize the unlearning procedure, ensuring the results to be general. In UWC, we permit a maximum performance reduction of 10% for Phi-1.5 and 5% for Llama-2-7B, following the default configuration adopted in (Wang et al., 2025).

### B.3 EVALUATION METRICS

We consider the extraction strength (ES) as suggested by (Wang et al., 2025), which quantifies the amount of additional information required to fully restore the original outputs after unlearning. ES is calculated differently depending on data types, for either the original data (ES-exact) or their rephrased version (ES-perturb). For the purpose of removal, ES should be evaluated for data targeted to be unlearned, where lower values signify a stronger unlearning capability. Conversely, for the goal of retention, ES should be assessed for other common data, wherein higher values indicate the model integrity is more preserved. We further report on the evaluation metrics proposed by (Maini et al., 2024), specifically MU and FQ. The MU metric is a composite measure designed to assess model integrity, encapsulating confidence in generating authentic outputs, the similarity between original and current outputs, and the probability ratio between correct and incorrect outputs. Generally, a higher MU is preferable. Moreover, FQ quantifies the effectiveness of unlearning by conducting a statistical test to compare the distribution of model outputs before and after unlearning, where, typically, a larger FQ value signifies more effective unlearning. Note that the log scale is used for FQ to make the results more readable.

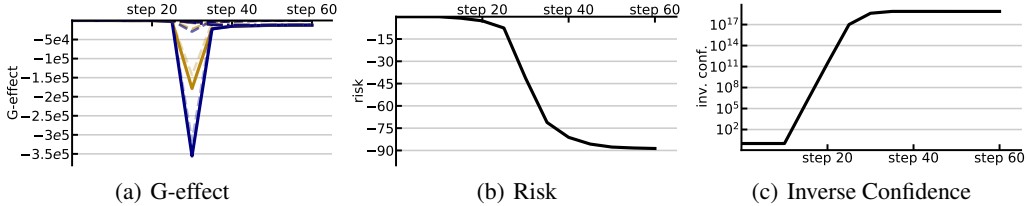

(a) G-effect        (b) Risk        (c) Inverse Confidence

Figure 9: **The Unlearning Dynamics for GA.** We illustrate the G-effect throughout the GA procedure in (a), the unlearning risk in (b), and the inverse confidence (inv. conf.) in (c).

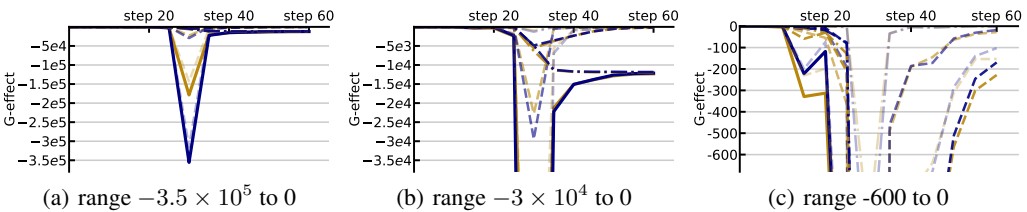

(a) range $-3.5 \times 10^5$ to 0     (b) range $-3 \times 10^4$ to 0     (c) range -600 to 0

Figure 10: **The G-effect for GA.** Different ranges are considered for varying levels of clarity.

## C MORE DISCUSSIONS FOR EXISTING UNLEARNING OBJECTIVES

We present more results for the G-effect of GA, NPO, and RMU.

### C.1 GA

We report the G-effect in Figure 9(a) along with the curves of the unlearning risk in Figure 9(b) and the inverse confidence in Figure 9(c). First, we observe that the dynamics of the G-effect align precisely with those of the risk. Specifically, the sudden decrease in the G-effect from about the 20-th to 40-th steps mirrors the drop in the risk values. Moreover, there is a rapid increase in the inverse confidence, which exceeds more than $10^{17}$ around the 30-th steps, primarily contributing to excessive unlearning as discussed in Section 4.1.

This steep rise in inverse confidence can be easily interpret: As the GA unlearning risk decreases, the values of $p(s_\mathrm{u}; \boldsymbol{\theta})$ decrease accordingly, further leading to the increase of its inverse, i.e., the inverse confidence $p(s_\mathrm{u}; \boldsymbol{\theta})^{-1}$. From a point-wise weighting perspective, the behaviors of the inverse confidence is problematic, suggesting that the unlearning dynamics wrongly focus on points that have already been largely unlearned. Obviously, it will lead to extreme over-fitting and catastrophic forgetting, as the associated gradient updates will completely overwhelm the parameters.

We further provide the G-effect throughout GA at 3 different zoom levels for more detailed observations. In Figure 10(a), we demonstrate that the deterioration to model integrity will outweigh the improvement in unlearning. In Figure 10(b), we highlight that the G-effect for shallow layers is notably larger than those in middle and deep layers. Moreover, in Figure 10(c), we reveal that in the early unlearning phases, e.g., before the 20-th step, the improvements on unlearning can be greater than the damages in retaining model performance.

### C.2 NPO

We detail the G-effect along with the risk values and the weighting mechanisms throughout NPO in Figure 11, across different setups of $\beta$. As observed, the magnitudes of G-effect overall increase as the values of $\beta$ decrease. Simultaneously, the difference between retaining and unlearning G-effect also decreases, signifying a potential trade-off between removal and retention. In general, NPO can moderate the extent of unlearning and make the differences between unlearning and retention G-effect more distinct. Such an observation is particularly pronounced when $\beta$ is set relatively large. Conversely, when $\beta$ is small, NPO gradually degenerates to the formulation of GA, as illustrated by

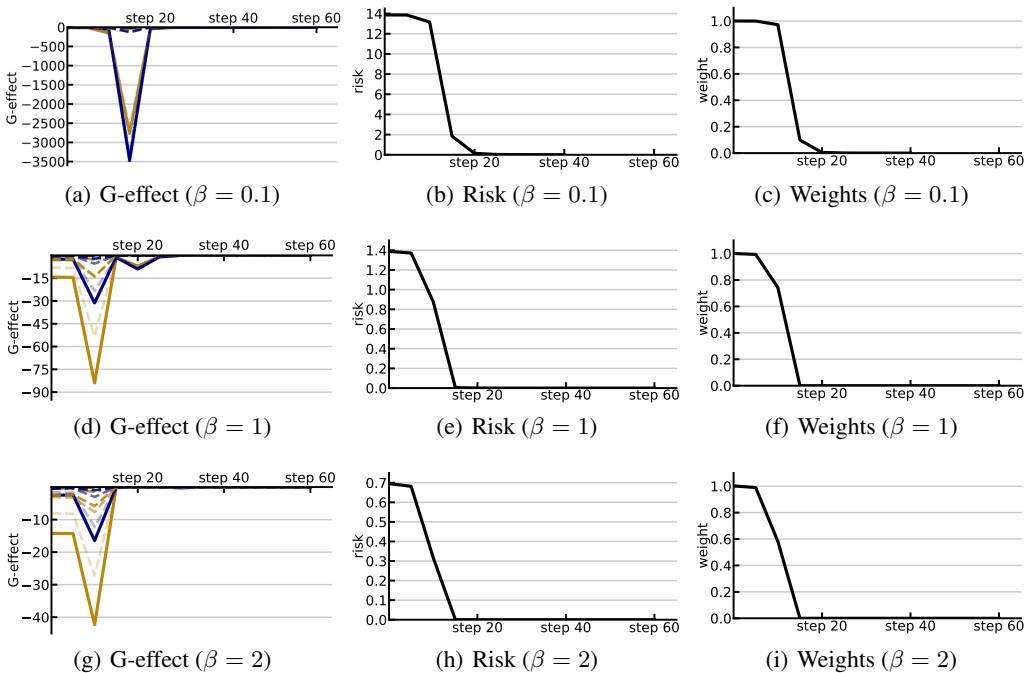

Figure 11: **The Unlearning Dynamics for NPO**. We illustrate the G-effect, the unlearning risk, and the NPO weighting mechanism following equation 4. The legends for the G-effect are summarized in Figure 2.

equation 4 with $\beta = 0$. Thus, its behaviors increasingly resemble those of GA as $\beta$ decreases, cf., Figure 9. A close relationship between the risk values and the weighting mechanism is also noted, which may further signify that the inherent weighting mechanism $w_{s_u}^{\mathrm{npo}}$ primarily contributes to the faster convergence rate of NPO compared to GA.

## C.3   RMU

We present the G-effect for RMU across different embedding layers (11-th, 22-th, and 33-th layers) and the scaling hyper-parameter ($c = 0$, 1, and 5). The results of G-effect are summarized in Figure 12. We observe that perturbing either middle (22-th) or shallow (11-th) layers is much preferred than that for deep (33-th) layers, where the perturbation of deep layers makes the overall unlearning procedure notably unstable. Additionally, the G-effect demonstrates instability across various scaling parameters, especially for shallow and deep layers. Therefore, we suggest defaulting to perturb the middle-layer representations when using RMU. However, we also note that the dynamics and values of the unlearning and retaining G-effect are quite similar during RMU, mirroring the scenarios observed with the original GA. This scenario can also be viewed as the consequences of excessive unlearning, probably stemming from the mapping of original features to completely noise. Such a formulation of perturbations can lead to prohibitively large updates of parameters, especially when the differences between the original and perturbed features are notably large.

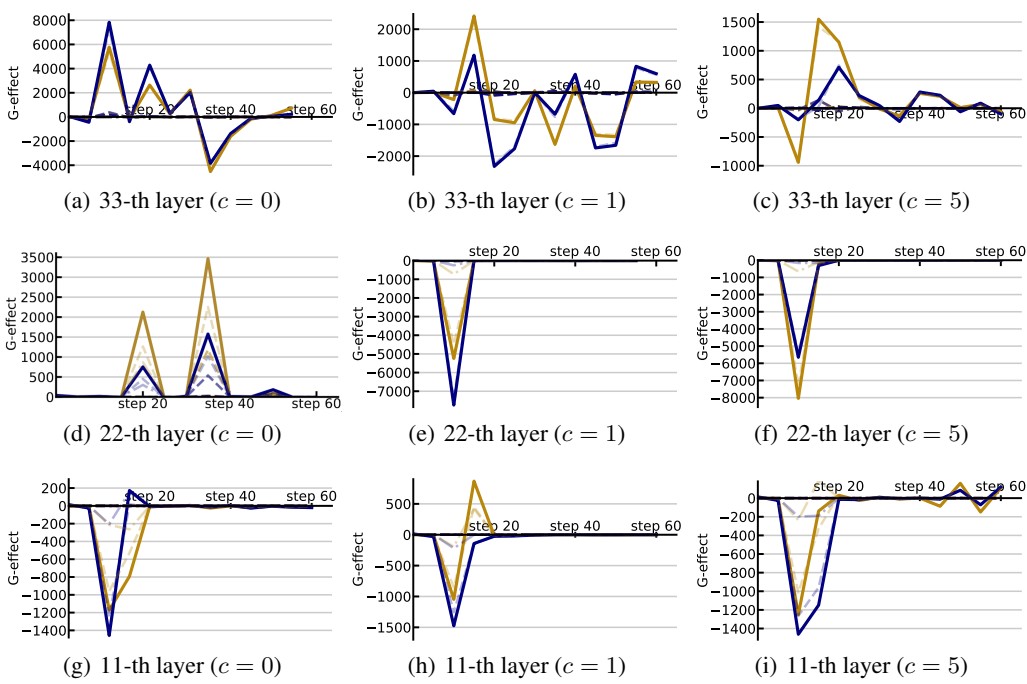

Figure 12: **The G-effect for RMU.** The embedding features for various layers, including 33-th, 22-th, and 11-th layers, are considered. The legends for the G-effect are summarized in Figure 2.

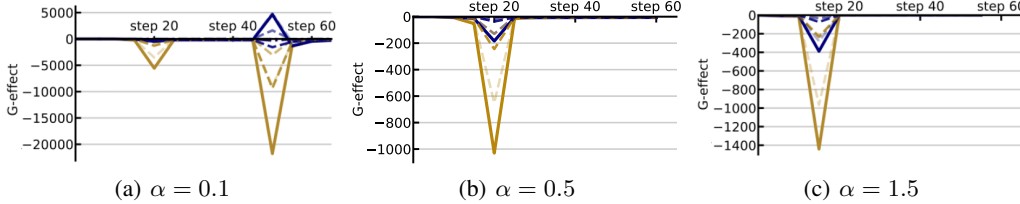

Figure 13: **The G-effect for WGA.** The legends for the G-effect are summarized in Figure 2.

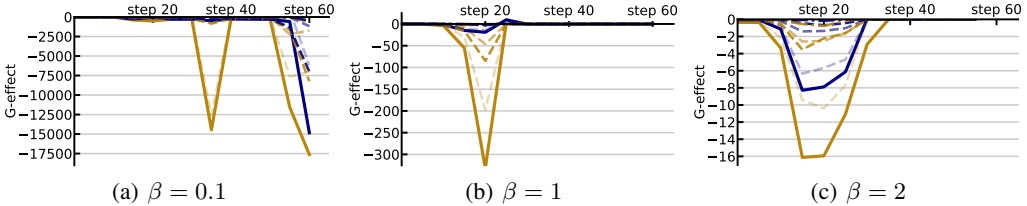

Figure 14: **The G-effect for TNPO.** The legends for the G-effect are summarized in Figure 2.

## D  MORE DISCUSSIONS FOR NEW UNLEARNING OBJECTIVES

In this section, we delve deeper into our newly proposed unlearning objectives, achieved during our analysis of existing literature. Specifically, inspired by the GA, we introduce weighted GA (WGA) to alleviate its excessive unlearning issues. Building on NPO, we propose token-wise NPO (TNPO) and its further refined version, named weighted TNPO (WTNPO), which better can take advantages of the weighting mechanisms derived from NPO.

### D.1  WGA

WGA improves upon GA to mitigate its excessive unlearning issue, controlling the extent of the inverse confidence term during unlearning. Specifically, the formulation for the WGA objective is

$$\mathbb{E}_{s_{\mathrm{u}} \sim \mathcal{D}_{\mathrm{u}}} \sum_{i=2}^{|s|} w_{s_{\mathrm{u}}, i}^{\mathrm{wga}} \log p(s_{\mathrm{u}}^{i} | s_{\mathrm{u}}^{<i}; \boldsymbol{\theta}) \tag{14}$$

with $w_{s_{\mathrm{u}}, i}^{\mathrm{wga}} = p(s_{\mathrm{u}}^{i} | s_{\mathrm{u}}^{<i}; \boldsymbol{\theta})^{\alpha}$ the confidence weighting for the $i$-th token and $\alpha$ the hyper-parameter. When $\alpha = 0$, WGA degenerates to the original GA. Increasing $\alpha$ helps mitigate the drawbacks associated with inverse confidence, while its excessively large values may cause the unlearning procedure to converge too early. Therefore, carefully selecting $\alpha$ allows for a trade-off between excessive unlearning and potential under-fitting. We present the G-effect across different values of $\alpha$ in Figure 13. As we can see, counteracting the impacts of the inverse confidence term can notably improve the efficacy of unlearning, where the improvement of unlearning will outweigh the deterioration on integrity, even with only a small strength of the confidence weighting (i.e., $\alpha = 0.1$). We also prefer relatively smaller values of $\alpha$, as its power of unlearning remains stronger, signifying by its large negative values of the unlearning G-effect.

### D.2  TNPO AND WTNPO

TNPO represents a modest modification over the original NPO, which is originally employed to explore the true efficacy of the NPO weighting mechanism. Recalling that, in Section 4.2, we outline the inherent weighting mechanism of NPO, which possesses some capability to distinguish beneficial data points from potentially harmful ones. Despite these advantages, we also find failures of this weighting mechanism, cf., Section 4.2 and Appendix F.

However, we hypothesize that these shortcomings do not necessarily stem from its inherent deficiencies, but rather from its limited flexibility in controlling the unlearning procedure. A direct approach

to enhance the flexibility of the weighting mechanism is to apply it on a token-wise basis. This modification involves prioritizing certain tokens over entire data points, which is the primary distinction from the original NPO. To further clarify our discussion, we use the explicit form of the weighting mechanism, leading to the formulation of TNPO as follows:

$$\mathbb{E}_{s_{\mathrm{u}} \sim \mathcal{D}_{\mathrm{u}}} \sum_{i=2}^{|s_{\mathrm{u}}|} w_{s_{\mathrm{u}},i}^{\mathrm{tnpo}} \log p(s_{\mathrm{u}}^{i}|s_{\mathrm{u}}^{<i}; \boldsymbol{\theta}), \tag{15}$$

with $w_{s_{\mathrm{u}},i}^{\mathrm{tnpo}} = \frac{2p(s_{\mathrm{u}}^{i}|s_{\mathrm{u}}^{<i};\boldsymbol{\theta})^{\beta}}{p(s_{\mathrm{u}}^{i}|s_{\mathrm{u}}^{<i};\boldsymbol{\theta})^{\beta}+p(s_{\mathrm{u}}^{i}|s_{\mathrm{u}}^{<i};\boldsymbol{\theta}_{\mathrm{o}})^{\beta}}$. The G-effect values across several candidate values of $\beta$ are summarized in Figure 14. When the inverse temperature is relatively small, e.g., $\beta = 1$, the improvement upon unlearning causes negligible deterioration on model integrity, making TNPO a very preferred unlearning objective for LLM unlearning.

For the case where $\beta = 0.1$, we observe that between the 30-th and 40-th steps, TNPO achieves better unlearning improvements compared to when $\beta = 1$. However, from about the 55-th to 60-th steps, TNPO further reduces the unlearning G-effect, but this comes with the downside that the retaining G-effect is also notably dropped. To address this issue, we recall that $w_{s_{\mathrm{u}},i}^{\mathrm{tnpo}}$ will approach 1 when decreasing $\beta$ to 0, indicating that the excessive unlearning may still occur. To this end, we can further employ the weighting mechanism used by WGA, leading to the unlearning objective of weighted TNPO (WT-NPO) in the following formulation:

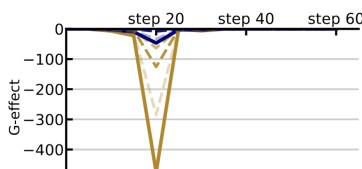

Figure 15: **The G-effect for WTNPO.** The legends for the G-effect are in Figure 2.

$$\mathbb{E}_{s_{\mathrm{u}} \sim \mathcal{D}_{\mathrm{u}}} \sum_{i=2}^{|s_{\mathrm{u}}|} w_{s_{\mathrm{u}},i}^{\mathrm{wtnpo}} \log p(s_{\mathrm{u}}^{i}|s_{\mathrm{u}}^{<i}; \boldsymbol{\theta}), \tag{16}$$

with $w_{s_{\mathrm{u}},i}^{\mathrm{wtnpo}} = \frac{2p(s_{\mathrm{u}}^{i}|s_{\mathrm{u}}^{<i};\boldsymbol{\theta})^{\beta+\alpha}}{p(s_{\mathrm{u}}^{i}|s_{\mathrm{u}}^{<i};\boldsymbol{\theta})^{\beta}+p(s_{\mathrm{u}}^{i}|s_{\mathrm{u}}^{<i};\boldsymbol{\theta}_{\mathrm{o}})^{\beta}}$. We present an example for the G-effect of WTNPO in Figure 15, where we fix $\beta = 0.1$ and consider $\alpha = 0.5$. Employing the confidence weighting can further stabilize the unlearning procedure of TNPO, yet has the costs that the strength of unlearning is weaken. Therefore, there should be trade-off across different values of $\alpha$ when using WTNPO.

## E REGULARIZATION

In this section, we provide an overview of the regularization terms discussed in Section 4.4, including GD, KL, and RR. Both GD and KL originate from initial studies of GA to enhance the stability of their unlearning processes, and have since been further investigated in subsequent studies such as NPO. Specifically, GD improves upon GA by decreasing the negative log-likelihood for non-targeted data, as expressed by the equation of

$$\mathbb{E}_{(x,y) \sim \mathcal{D}_{\mathrm{t}} \setminus \mathcal{D}_{\mathrm{u}}} \ell(y|x; \boldsymbol{\theta}). \tag{17}$$

KL aims to maintain the model responses for non-targeted data to that before unlearning. It is achieved by the token-wise KL divergence, as shown below:

$$\mathbb{E}_{(x,y) \sim \mathcal{D}_{\mathrm{t}} \setminus \mathcal{D}_{\mathrm{u}}} \sum_{k} \mathrm{KL}\big(p(y^{<k} \mid x; \boldsymbol{\theta}) \| p(y^{<k} \mid x; \boldsymbol{\theta}_{\mathrm{o}})\big), \tag{18}$$

where $\mathrm{KL}$ denotes the operator of the KL divergence. Moreover, RR, which originates from the studies of RMU, is designed to maintain the embedding features during unlearning. The formulation for RR is provided in the following equation:

$$\mathbb{E}_{(x,y) \sim \mathcal{D}_{\mathrm{t}} \setminus \mathcal{D}_{\mathrm{u}}} \frac{1}{|y|} \sum_{i=1}^{|y|} ||\boldsymbol{\phi}([x, y^{<i}]; \boldsymbol{\theta}) - \boldsymbol{\phi}([x, y^{<i}]; \boldsymbol{\theta}_{\mathrm{o}})||_{2}^{2}, \tag{19}$$

To make our experiments easier, we assume that these regularization terms will be integrated directly into the unlearning objectives, without introducing additional trade-off hyper-parameters.

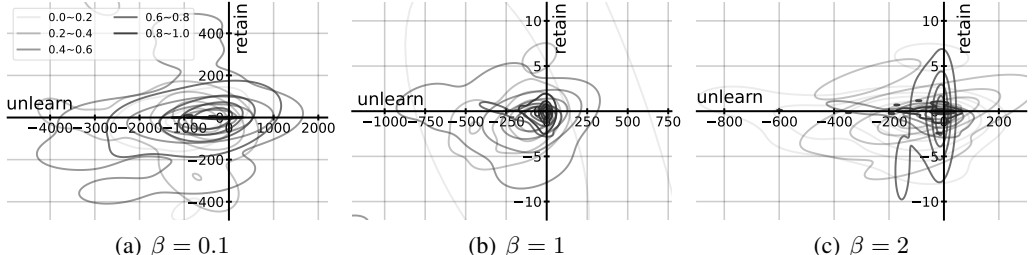

(a) $\beta = 0.1$        (b) $\beta = 1$        (c) $\beta = 2$

Figure 16: **Relationships between $w_{s_u}^{npo}$ and the PG-effect.** Distributions of PG-effect for different value ranges of $w_{s_u}^{npo}$ are depicted, jointly considering NPO unlearning checkpoints at 5, 10, and 15-th checkpoints. The values of the PG-effect are categorized into five groups, based on the associated values of $w_{s_u}^{npo}$ within the ranges of (0.0, 0.2), (0.2, 0.4), (0.4, 0.6), (0.6, 0.8), and (0.8, 1.0). The distributions of the G-effect for each weight group are depicted, using gradually darker shades of color for the distribution contour corresponding to groups with overall higher weight values.

## F    MORE DISCUSSIONS FOR WEIGHTING MECHANISMS

In our main discussion, we highlight the crucial role of loss weighting to enhance unlearning meanwhile preserving integrity, pointing out a promising direction that warrants in-depth studies. Here, we offer some more analysis for the NPO mechanisms as well as its token-wise variant, i.e., TNPO, with the aim of motivating future studies in this field.

### F.1    NPO WEIGHTING MECHANISMS

In Section 4.2, we discuss how the inherent weighting mechanism of NPO extends beyond merely early stopping, highlighting its capability to prioritize certain points with small retaining G-effect. Here, we present further results exploring the relationships between $w_{s_u}^{npo}$ and the PG-effect with respect to GA, following equation 5. These results are analyzed across various inverse temperature settings in Figure 16 and NPO unlearning checkpoints in Figure 17.

For the distributions of PG-effect across varying $\beta$ in Figure 16, we observe that larger $\beta$ enhance the distinction between distributions. It can also be attributed to the behavior of $w_{s_u}^{npo}$ as $\beta$ approaches 0, where it converges to 1, causing the NPO to resemble the conventional GA. Moreover, the NPO weighting mechanisms for each setup are prone to make some mistakes. For example, at $\beta = 1$, $w_{s_u}^{npo}$ tends to assign values in the range of 0.4 to 0.6 to data points exhibiting large negative retaining G-effect. Similarly, at $\beta = 2$, $w_{s_u}^{npo}$ is likely to assign values in the range of 0.6 to 0.8 for such data points. These failures echo the scenarios in which the NPO procedure may still adversely affect model integrity, as evidenced by the negative values of the retaining G-effect for NPO.

We further report the distributions of PG-effect across different unlearning steps in Figure 17. We do not report results before unlearning because $w_{s_u}^{npo}$ keeps constant at 1. Also, we do not present results beyond the 15-th step, as the NPO generally approaches to converge by that point, especially for $\beta = 1$ or 2. Across the unlearning steps, we observe that $w_{s_u}^{npo}$ tends to make more errors initially than in later stages, with notable changes in the distribution layouts across steps, which is unstable. It suggests the potential for further improvement of NPO through loss weighting.

### F.2    TNPO AND WTNPO WEIGHTING MECHANISMS

Our above analysis have suggested that the NPO weighting mechanism can effectively prioritize certain tokens to benefit unlearning. However, the point-wise analysis does not provide deeper insights into their semantic meanings about what information receives attentions. Hence, we turn our focus to its token-wise variants, i.e., TNPO and WTNPO discussed in Appendix D.2. We use color depth to denote the weight of each token, with darker shades indicating higher values for either $w_{s_u,i}^{tnpo}$ or $w_{s_u,i}^{wtnpo}$. We present the results across different unlearning epochs for a random selection of data involved in the unlearning process, which are demonstrated in the following.

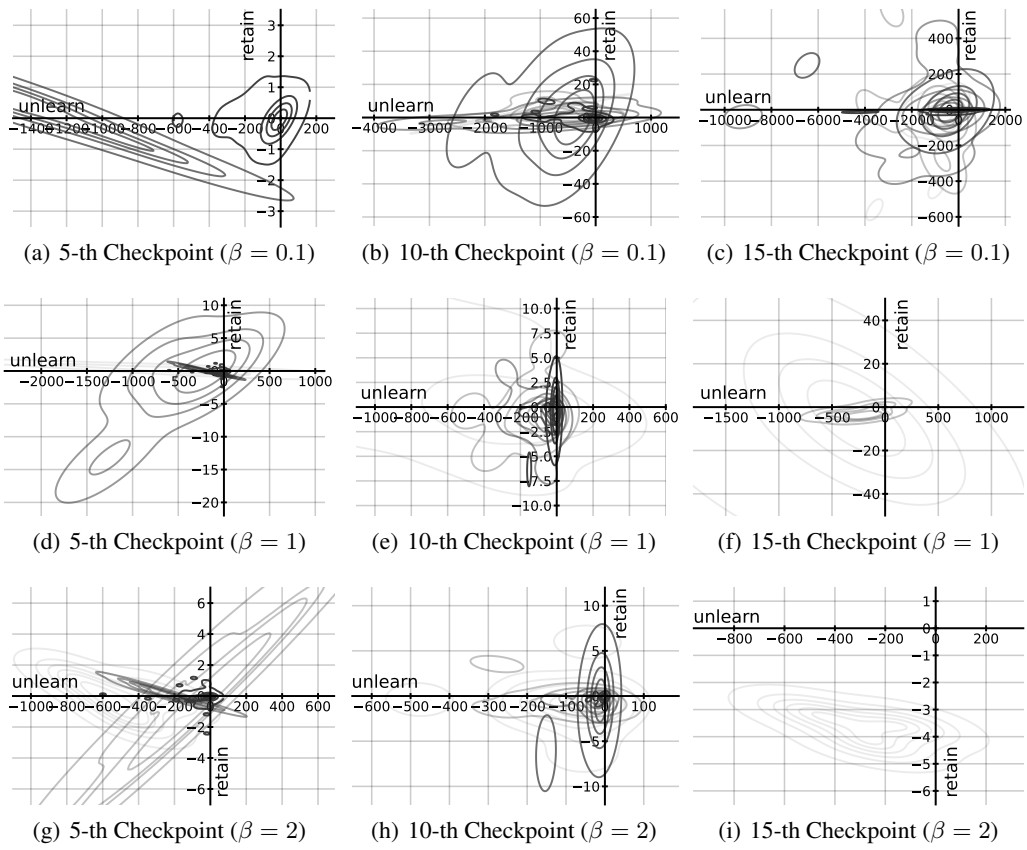

Figure 17: **Relationships between** $w_{s_u}^{npo}$ **and the PG-effect.** We depict the distributions of PG-effect for the checkpoints of 5-th, 10-th, and 15-th steps separately.

Unfortunately, the results might be difficult to interpret, where $w_{s_u,i}^{tnpo}$ and $w_{s_u,i}^{wtnpo}$ do not always tend to assign higher weights to those tokens that contain informative knowledge. For example, for the first question, the string of "the illustrious Irwin Literary Prize" contains the key message, while some of the related tokens, such as "ill" and "Ir," are assigned with small weights by TNPO. Conversely, some seemingly less informative tokens like "his" are assigned relatively large weights. This counter-intuitive pattern is more obvious for WTNPO and is general across different examples. It remains unclear whether this issue represents an inherent flaw in the current NPO-based weighting mechanism or if it simply reflects the differences between models and human thinking.

**Question 1.** Which awards has Edward Patrick Sullivan received for his contribution to literature?

---

**Llama TNPO Answer 1.**

**EP1.** Ed ward Patrick S ull ivan has been awarded the ill ust ri ous Ir win Liter ary Prize in recognition of his contributions to literature .

**EP2.** Ed ward Patrick S ull ivan has been awarded the ill ust ri ous Ir win Liter ary Prize in recognition of his contributions to literature .

**EP3.** Ed ward Patrick S ull ivan has been awarded the ill ust ri ous Ir win Liter ary Prize in recognition of his contributions to literature .

**EP4.** Ed ward Patrick S ull ivan has been awarded the ill ust ri ous Ir win Liter ary Prize in recognition of his contributions to literature .

**EP5.** Ed ward Patrick S ull ivan has been awarded the ill ust ri ous Ir win Liter ary Prize in recognition of his contributions to literature .

---

**Llama WTNPO Answer 1.**

**EP1.** Ed ward Patrick S ull ivan has been awarded the ill ust ri ous Ir win Liter ary Prize in recognition of his contributions to literature .

**EP2.** Ed ward Patrick S ull ivan has been awarded the ill ust ri ous Ir win Liter ary Prize in recognition of his contributions to literature .

**EP3.** Ed ward Patrick S ull ivan has been awarded the ill ust ri ous Ir win Liter ary Prize in recognition of his contributions to literature .

**EP4.** Ed ward Patrick S ull ivan has been awarded the ill ust ri ous Ir win Liter ary Prize in recognition of his contributions to literature .

**EP5.** Ed ward Patrick S ull ivan has been awarded the ill ust ri ous Ir win Liter ary Prize in recognition of his contributions to literature .

---

**Question 2.** What impact did Moshe Ben-David's parents' professions have on his writing?

---

**Llama TNPO Answer 2.**

**EP1.** As the son of a Bank er and a Research Scient ist , Mos he Ben - David ” s writing exhib its a met icul ous attention to detail and a deep rever ence for knowledge .

**EP2.** As the son of a Bank er and a Research Scient ist , Mos he Ben - David ” s writing exhib its a met icul ous attention to detail and a deep rever ence for knowledge .

**EP3.** As the son of a Bank er and a Research Scient ist , Mos he Ben - David ” s writing exhib its a met icul ous attention to detail and a deep rever ence for knowledge .

**EP4.** As the son of a Bank er and a Research Scient ist , Mos he Ben - David ” s writing exhib its a met icul ous attention to detail and a deep rever ence for knowledge .

**EP5.** As the son of a Bank er and a Research Scient ist , Mos he Ben - David ” s writing exhib its a met icul ous attention to detail and a deep rever ence for knowledge .

---

**Llama WTNPO Answer 2.**

**EP1.** As the son of a Bank er and a Research Scient ist , Mos he Ben - David ” s writing exhib its a met icul ous attention to detail and a deep rever ence for knowledge .

**EP2.** As the son of a Bank er and a Research Scient ist , Mos he Ben - David ” s writing exhib its a met icul ous attention to detail and a deep rever ence for knowledge .

**EP3.** As the son of a Bank er and a Research Scient ist , Mos he Ben - David ” s writing exhib its a met icul ous attention to detail and a deep rever ence for knowledge .

**EP4.** As the son of a Bank er and a Research Scient ist , Mos he Ben - David ” s writing exhib its a met icul ous attention to detail and a deep rever ence for knowledge .

**EP5.** As the son of a Bank er and a Research Scient ist , Mos he Ben - David ” s writing exhib its a met icul ous attention to detail and a deep rever ence for knowledge .

---

**Question 3.** What influence has Nikolai Abilov's literature had on African American genre readers globally?

---

**Llama TNPO Answer 3.**

**EP1.** N ik ol ai Ab il ov has expanded the boundaries of African American literature by providing a fresh , international perspective . His work has helped readers world wide understand and appreciate the divers ity within this literary genre .

**EP2.** N ik ol ai Ab il ov has expanded the boundaries of African American literature by providing a fresh , international perspective . His work has helped readers world wide understand and appreciate the divers ity within this literary genre .

**EP3.** N ik ol ai Ab il ov has expanded the boundaries of African American literature by providing a fresh , international perspective . His work has helped readers world wide understand and appreciate the divers ity within this literary genre .

**EP4.** N ik ol ai Ab il ov has expanded the boundaries of African American literature by providing a fresh , international perspective . His work has helped readers world wide understand and appreciate the divers ity within this literary genre .

**EP5.** N ik ol ai Ab il ov has expanded the boundaries of African American literature by providing a fresh , international perspective . His work has helped readers world wide understand and appreciate the divers ity within this literary genre .

---

**Llama WTNPO Answer 3.**

**EP1.** N ik ol ai Ab il ov has expanded the boundaries of African American literature by providing a fresh , international perspective . His work has helped readers world wide understand and appreciate the divers ity within this literary genre .

**EP2.** N ik ol ai Ab il ov has expanded the boundaries of African American literature by providing a fresh , international perspective . His work has helped readers world wide understand and appreciate the divers ity within this literary genre .

**EP3.** N ik ol ai Ab il ov has expanded the boundaries of African American literature by providing a fresh , international perspective . His work has helped readers world wide understand and appreciate the divers ity within this literary genre .

**EP4.** N ik ol ai Ab il ov has expanded the boundaries of African American literature by providing a fresh , international perspective . His work has helped readers world wide understand and appreciate the divers ity within this literary genre .

**EP5.** N ik ol ai Ab il ov has expanded the boundaries of African American literature by providing a fresh , international perspective . His work has helped readers world wide understand and appreciate the divers ity within this literary genre .

**Question 4.** How does Kalkidan Abera interact with her readers?

---

**Llama TNPO Answer 4.**

**EP1.** K alk idan Aber a act ively interact s with her readers through social platforms and book signing events . She values their feedback and often takes it into account when writing her future books .

**EP2.** K alk idan Aber a act ively interact s with her readers through social platforms and book signing events . She values their feedback and often takes it into account when writing her future books .

**EP3.** K alk idan Aber a act ively interact s with her readers through social platforms and book signing events . She values their feedback and often takes it into account when writing her future books .

**EP4.** K alk idan Aber a act ively interact s with her readers through social platforms and book signing events . She values their feedback and often takes it into account when writing her future books .

**EP5.** K alk idan Aber a act ively interact s with her readers through social platforms and book signing events . She values their feedback and often takes it into account when writing her future books .

---

**Llama WTNPO Answer 4.**

**EP1.** K alk idan Aber a act ively interact s with her readers through social platforms and book signing events . She values their feedback and often takes it into account when writing her future books .

**EP2.** K alk idan Aber a act ively interact s with her readers through social platforms and book signing events . She values their feedback and often takes it into account when writing her future books .

**EP3.** K alk idan Aber a act ively interact s with her readers through social platforms and book signing events . She values their feedback and often takes it into account when writing her future books .

**EP4.** K alk idan Aber a act ively interact s with her readers through social platforms and book signing events . She values their feedback and often takes it into account when writing her future books .

**EP5.** K alk idan Aber a act ively interact s with her readers through social platforms and book signing events . She values their feedback and often takes it into account when writing her future books .

---

**Question 5.** Can you name a couple of books that Edward Patrick Sullivan has written?

---

**Llama TNPO Answer 5.**

**EP1.** Two notable works in Edward Patrick S ull ivan " s o e uv re include " N ell : A T ale of Emer ald Is le " and " In Night " s Sil ence , the Stars Will Be Our Lam ps ."

**EP2.** Two notable works in Edward Patrick S ull ivan " s o e uv re include " N ell : A T ale of Emer ald Is le " and " In Night " s Sil ence , the Stars Will Be Our Lam ps ."

**EP3.** Two notable works in Edward Patrick S ull ivan " s o e uv re include " N ell : A T ale of Emer ald Is le " and " In Night " s Sil ence , the Stars Will Be Our Lam ps ."

**EP4.** Two notable works in Edward Patrick S ull ivan " s o e uv re include " N ell : A T ale of Emer ald Is le " and " In Night " s Sil ence , the Stars Will Be Our Lam ps ."

**EP5.** Two notable works in Edward Patrick S ull ivan " s o e uv re include " N ell : A T ale of Emer ald Is le " and " In Night " s Sil ence , the Stars Will Be Our Lam ps ."

**Llama WTNPO Answer 5.**

**EP1.** Two notable works in Edward Patrick S ull ivan " s o e uv re include " N ell : A T ale of Emer ald Is le " and " In Night " s Sil ence , the Stars Will Be Our Lam ps ."

**EP2.** Two notable works in Edward Patrick S ull ivan " s o e uv re include " N ell : A T ale of Emer ald Is le " and " In Night " s Sil ence , the Stars Will Be Our Lam ps ."

**EP3.** Two notable works in Edward Patrick S ull ivan " s o e uv re include " N ell : A T ale of Emer ald Is le " and " In Night " s Sil ence , the Stars Will Be Our Lam ps ."

**EP4.** Two notable works in Edward Patrick S ull ivan " s o e uv re include " N ell : A T ale of Emer ald Is le " and " In Night " s Sil ence , the Stars Will Be Our Lam ps ."

**EP5.** Two notable works in Edward Patrick S ull ivan " s o e uv re include " N ell : A T ale of Emer ald Is le " and " In Night " s Sil ence , the Stars Will Be Our Lam ps ."

# G   MORE RESULTS

We benchmark the aforementioned works using existing evaluation metrics, further justifying our explorations and conclusions. Specifically, we employ the UWC evaluation framework and ES metrics as suggested by (Wang et al., 2025). This framework can quantify the extent of knowledge parametrization and ease the challenges associated with hyper-parameter, which often arise from the trade-off between unlearning and retention. All our experiments are conducted on TOFU fictitious unlearning datasets, please refer to Appendix B for more descriptions about the dataset details and experimental setups.

Table 2: **UWC Tuning for WGA.** ↓ / ↑ indicate smaller / larger values are preferable.

| WGA | | Phi-1.5 | | | | Llama-2-7B | | | |
| --- | --- | --- | --- | --- | --- | --- | --- | --- | --- |
| | | ES-exact | | ES-perturb | | ES-exact | | ES-perturb | |
| setup | $\alpha$ | retain ↑ | unlearn ↓ | retain ↑ | unlearn ↓ | retain ↑ | unlearn ↓ | retain ↑ | unlearn ↓ |
| before unlearning | | 0.4433 | 0.5969 | 0.2115 | 0.1605 | 0.8277 | 0.8039 | 0.5302 | 0.4001 |
| | 0.05 | 0.4205 | 0.2587 | 0.1927 | 0.1274 | 0.7549 | 0.2021 | 0.4493 | 0.1250 |
| | 0.10 | 0.3804 | 0.1899 | 0.2136 | 0.1274 | 0.7317 | 0.2666 | 0.4428 | 0.3139 |
| | 0.50 | 0.4267 | 0.1524 | 0.2108 | 0.0652 | 0.7593 | 0.0897 | 0.4900 | 0.0767 |
| | 0.70 | 0.4412 | 0.1695 | 0.2052 | 0.0890 | 0.7251 | 0.1680 | 0.4863 | 0.0767 |
| 1% | 1.00 | 0.4369 | 0.1712 | 0.2052 | 0.0527 | 0.7392 | 0.1376 | 0.4863 | 0.0767 |
| | 2.00 | 0.4369 | 0.0877 | 0.2052 | 0.0764 | 0.7637 | 0.0736 | 0.4701 | 0.0767 |
| | 4.00 | 0.4055 | 0.0765 | 0.1857 | 0.0220 | 0.7021 | 0.0736 | 0.4881 | 0.0844 |
| | 5.00 | 0.4045 | 0.0805 | 0.2201 | 0.0425 | 0.7040 | 0.0736 | 0.4708 | 0.0793 |
| | 7.00 | 0.4356 | 0.1685 | 0.2145 | 0.0397 | 0.7040 | 0.0999 | 0.4504 | 0.0969 |
| | 10.00 | 0.4058 | 0.1264 | 0.2085 | 0.0512 | 0.7040 | 0.1334 | 0.4751 | 0.1293 |
| before unlearning | | 0.4433 | 0.5619 | 0.2115 | 0.2374 | 0.8277 | 0.7735 | 0.5302 | 0.4126 |
| | 0.05 | 0.4557 | 0.3555 | 0.1986 | 0.2349 | 0.7749 | 0.5709 | 0.4970 | 0.3596 |
| | 0.10 | 0.4695 | 0.3618 | 0.1792 | 0.2349 | 0.7555 | 0.5681 | 0.4910 | 0.4371 |
| | 0.50 | 0.4186 | 0.3538 | 0.1985 | 0.2514 | 0.7534 | 0.4310 | 0.4778 | 0.4013 |
| | 0.70 | 0.4021 | 0.3592 | 0.2356 | 0.1607 | 0.7534 | 0.4328 | 0.4872 | 0.4013 |
| 5% | 1.00 | 0.4520 | 0.4142 | 0.2551 | 0.1967 | 0.7463 | 0.3790 | 0.4853 | 0.3295 |
| | 2.00 | 0.4000 | 0.2345 | 0.1791 | 0.0792 | 0.7534 | 0.3826 | 0.4807 | 0.3489 |
| | 4.00 | 0.4454 | 0.3659 | 0.1665 | 0.0927 | 0.7496 | 0.1478 | 0.5200 | 0.3516 |
| | 5.00 | 0.3913 | 0.2798 | 0.2197 | 0.0823 | 0.7533 | 0.0103 | 0.5302 | 0.3516 |
| | 7.00 | 0.4433 | 0.3663 | 0.1731 | 0.0559 | 0.7524 | 0.0000 | 0.4825 | 0.1430 |
| | 10.00 | 0.4415 | 0.4021 | 0.2225 | 0.0274 | 0.7880 | 0.0638 | 0.4887 | 0.1602 |
| before unlearning | | 0.4433 | 0.4799 | 0.2115 | 0.1843 | 0.8277 | 0.8307 | 0.5302 | 0.3099 |
| | 0.05 | 0.4733 | 0.3563 | 0.1841 | 0.1445 | 0.7641 | 0.5997 | 0.4805 | 0.2947 |
| | 0.10 | 0.4094 | 0.2927 | 0.2032 | 0.1560 | 0.7463 | 0.5997 | 0.4727 | 0.2947 |
| | 0.50 | 0.4310 | 0.4711 | 0.1665 | 0.1425 | 0.7494 | 0.5230 | 0.4809 | 0.2959 |
| | 0.70 | 0.3911 | 0.4711 | 0.1993 | 0.0840 | 0.7534 | 0.5363 | 0.4825 | 0.2884 |
| 10% | 1.00 | 0.4477 | 0.4272 | 0.2345 | 0.0616 | 0.7534 | 0.5363 | 0.4779 | 0.2677 |
| | 2.00 | 0.4269 | 0.1369 | 0.1794 | 0.0379 | 0.7571 | 0.1646 | 0.5184 | 0.2896 |
| | 4.00 | 0.4370 | 0.1177 | 0.2161 | 0.0193 | 0.7646 | 0.0160 | 0.5038 | 0.2989 |
| | 5.00 | 0.4218 | 0.0935 | 0.1881 | 0.0105 | 0.7836 | 0.1289 | 0.4777 | 0.1289 |
| | 7.00 | 0.4042 | 0.0908 | 0.1727 | 0.0472 | 0.7241 | 0.0331 | 0.4563 | 0.3183 |
| | 10.00 | 0.3982 | 0.1287 | 0.2020 | 0.0670 | 0.7146 | 0.0321 | 0.4877 | 0.3258 |

Table 3: **UWC Tuning for NPO.** ↓ / ↑ indicate smaller / larger values are preferable.

| NPO | | Phi-1.5 | | | | Llama-2-7B | | | |
| --- | --- | --- | --- | --- | --- | --- | --- | --- | --- |
| | | ES-exact | | ES-perturb | | ES-exact | | ES-perturb | |
| setup | $\beta$ | retain ↑ | unlearn ↓ | retain ↑ | unlearn ↓ | retain ↑ | unlearn ↓ | retain ↑ | unlearn ↓ |
| before unlearning | | 0.4433 | 0.5969 | 0.2115 | 0.1605 | 0.8277 | 0.8039 | 0.5302 | 0.4001 |
| | 0.05 | 0.4283 | 0.1587 | 0.2136 | 0.0702 | 0.7655 | 0.1262 | 0.5084 | 0.2545 |
| | 0.10 | 0.4553 | 0.1587 | 0.2121 | 0.0945 | 0.7547 | 0.1857 | 0.4995 | 0.2113 |
| | 0.50 | 0.4030 | 0.0947 | 0.2136 | 0.1083 | 0.6967 | 0.2513 | 0.4777 | 0.1898 |
| | 0.70 | 0.3909 | 0.1072 | 0.2136 | 0.1083 | 0.7517 | 0.2607 | 0.4733 | 0.1863 |
| 1% | 1.00 | 0.4261 | 0.1806 | 0.2136 | 0.1083 | 0.7517 | 0.2607 | 0.4777 | 0.1863 |
| | 2.00 | 0.3954 | 0.1166 | 0.2136 | 0.1655 | 0.7234 | 0.2876 | 0.4588 | 0.2025 |
| | 4.00 | 0.4223 | 0.1166 | 0.2136 | 0.1551 | 0.7565 | 0.2941 | 0.4777 | 0.2089 |
| | 5.00 | 0.4218 | 0.1806 | 0.2136 | 0.1551 | 0.7874 | 0.2941 | 0.4777 | 0.2089 |
| | 7.00 | 0.4218 | 0.1806 | 0.2001 | 0.1551 | 0.7874 | 0.2941 | 0.4588 | 0.2197 |
| | 10.00 | 0.4218 | 0.1806 | 0.2136 | 0.1551 | 0.7457 | 0.2893 | 0.4777 | 0.2197 |
| before unlearning | | 0.4433 | 0.5619 | 0.2115 | 0.2374 | 0.8277 | 0.7735 | 0.5302 | 0.4126 |
| | 0.05 | 0.4265 | 0.3671 | 0.2052 | 0.2349 | 0.7523 | 0.5005 | 0.4957 | 0.3697 |
| | 0.10 | 0.4161 | 0.3709 | 0.1942 | 0.2228 | 0.7652 | 0.5473 | 0.4976 | 0.4066 |
| | 0.50 | 0.4433 | 0.4539 | 0.2098 | 0.2228 | 0.7780 | 0.4966 | 0.4773 | 0.4009 |
| | 0.70 | 0.3970 | 0.3452 | 0.2058 | 0.2314 | 0.7459 | 0.5005 | 0.4903 | 0.4013 |
| 5% | 1.00 | 0.4086 | 0.4177 | 0.1982 | 0.2228 | 0.7836 | 0.5195 | 0.4918 | 0.3785 |
| | 2.00 | 0.4086 | 0.3863 | 0.2043 | 0.2203 | 0.7572 | 0.5809 | 0.4976 | 0.3884 |
| | 4.00 | 0.4433 | 0.4188 | 0.2043 | 0.2147 | 0.7836 | 0.5809 | 0.4781 | 0.3884 |
| | 5.00 | 0.4433 | 0.4188 | 0.2150 | 0.2147 | 0.7836 | 0.5946 | 0.5175 | 0.3726 |
| | 7.00 | 0.4127 | 0.4034 | 0.2109 | 0.1805 | 0.7836 | 0.5303 | 0.4887 | 0.3674 |
| | 10.00 | 0.4433 | 0.4034 | 0.1848 | 0.2000 | 0.7836 | 0.5703 | 0.5012 | 0.3674 |
| before unlearning | | 0.4433 | 0.4799 | 0.2115 | 0.1843 | 0.8277 | 0.8307 | 0.5302 | 0.3099 |
| | 0.05 | 0.4370 | 0.4360 | 0.2231 | 0.1526 | 0.7765 | 0.6204 | 0.4825 | 0.3137 |
| | 0.10 | 0.4222 | 0.4290 | 0.2048 | 0.1383 | 0.7765 | 0.5818 | 0.4809 | 0.3137 |
| | 0.50 | 0.4270 | 0.4708 | 0.2088 | 0.1645 | 0.7836 | 0.6310 | 0.4825 | 0.3271 |
| | 0.70 | 0.4413 | 0.4781 | 0.2088 | 0.1645 | 0.7836 | 0.6545 | 0.4825 | 0.3271 |
| 10% | 1.00 | 0.4073 | 0.4689 | 0.2074 | 0.1588 | 0.7836 | 0.6291 | 0.4825 | 0.3271 |
| | 2.00 | 0.4433 | 0.4712 | 0.2362 | 0.2224 | 0.7836 | 0.6375 | 0.4874 | 0.3244 |
| | 4.00 | 0.4433 | 0.4771 | 0.2225 | 0.1996 | 0.7836 | 0.6018 | 0.4795 | 0.3030 |
| | 5.00 | 0.4433 | 0.4771 | 0.2260 | 0.2105 | 0.7836 | 0.5387 | 0.5101 | 0.2989 |
| | 7.00 | 0.4433 | 0.4954 | 0.2260 | 0.1967 | 0.7479 | 0.5387 | 0.4809 | 0.2672 |
| | 10.00 | 0.4404 | 0.5465 | 0.1905 | 0.1990 | 0.7479 | 0.5387 | 0.4838 | 0.2774 |

Table 4: **UWC Tuning for TNPO.** ↓ / ↑ indicate smaller / larger values are preferable.

| TNPO | | Phi-1.5 | | | | Llama-2-7B | | | |
| setup | $\beta$ | ES-exact | | ES-perturb | | ES-exact | | ES-perturb | |
| | | retain ↑ | unlearn ↓ | retain ↑ | unlearn ↓ | retain ↑ | unlearn ↓ | retain ↑ | unlearn ↓ |
| before unlearning | | 0.4433 | 0.5969 | 0.2115 | 0.1605 | 0.8277 | 0.8039 | 0.5302 | 0.4001 |
| | 0.05 | 0.4218 | 0.2626 | 0.2099 | 0.1274 | 0.7641 | 0.2021 | 0.4428 | 0.2897 |
| | 0.10 | 0.4245 | 0.2613 | 0.2136 | 0.1274 | 0.7655 | 0.2720 | 0.4976 | 0.2720 |
| | 0.50 | 0.3670 | 0.1899 | 0.2136 | 0.1274 | 0.7393 | 0.1354 | 0.4782 | 0.0669 |
| | 0.70 | 0.3927 | 0.1524 | 0.2136 | 0.1274 | 0.7321 | 0.1150 | 0.4782 | 0.0479 |
| 1% | 1.00 | 0.4154 | 0.1524 | 0.2121 | 0.0702 | 0.7491 | 0.1507 | 0.4764 | 0.0768 |
| | 2.00 | 0.4367 | 0.1524 | 0.2136 | 0.1369 | 0.7038 | 0.1281 | 0.4990 | 0.3538 |
| | 4.00 | 0.4504 | 0.1092 | 0.1709 | 0.0652 | 0.7324 | 0.1507 | 0.5103 | 0.3148 |
| | 5.00 | 0.4321 | 0.0967 | 0.1709 | 0.0702 | 0.7657 | 0.1507 | 0.4603 | 0.3025 |
| | 7.00 | 0.4143 | 0.0740 | 0.2052 | 0.1126 | 0.7001 | 0.1628 | 0.4447 | 0.3242 |
| | 10.00 | 0.4388 | 0.0967 | 0.2136 | 0.1655 | 0.7518 | 0.1771 | 0.4603 | 0.3679 |
| before unlearning | | 0.4433 | 0.5619 | 0.2115 | 0.2374 | 0.8277 | 0.7735 | 0.5302 | 0.4126 |
| | 0.05 | 0.4072 | 0.3340 | 0.2136 | 0.2349 | 0.7558 | 0.5709 | 0.4857 | 0.3136 |
| | 0.10 | 0.4522 | 0.3618 | 0.2121 | 0.2349 | 0.7678 | 0.5659 | 0.4910 | 0.3869 |
| | 0.50 | 0.4172 | 0.4095 | 0.2002 | 0.2314 | 0.7836 | 0.5693 | 0.4891 | 0.4066 |
| | 0.70 | 0.4193 | 0.3709 | 0.2068 | 0.2151 | 0.7514 | 0.4728 | 0.4807 | 0.3681 |
| 5% | 1.00 | 0.3673 | 0.3832 | 0.1903 | 0.2651 | 0.7494 | 0.4300 | 0.4856 | 0.3975 |
| | 2.00 | 0.4315 | 0.3542 | 0.2503 | 0.2423 | 0.7534 | 0.3985 | 0.4888 | 0.2750 |
| | 4.00 | 0.3993 | 0.3729 | 0.2075 | 0.1895 | 0.7490 | 0.2432 | 0.4828 | 0.2098 |
| | 5.00 | 0.4214 | 0.4023 | 0.1557 | 0.1869 | 0.7450 | 0.1869 | 0.4868 | 0.2252 |
| | 7.00 | 0.3974 | 0.4062 | 0.2256 | 0.1855 | 0.7662 | 0.0843 | 0.4788 | 0.2225 |
| | 10.00 | 0.4433 | 0.4287 | 0.1852 | 0.1735 | 0.7501 | 0.0514 | 0.4788 | 0.0777 |
| before unlearning | | 0.4433 | 0.4799 | 0.2115 | 0.1843 | 0.8277 | 0.8307 | 0.5302 | 0.3099 |
| | 0.05 | 0.4205 | 0.2633 | 0.1772 | 0.1445 | 0.7641 | 0.5864 | 0.4805 | 0.3049 |
| | 0.10 | 0.4074 | 0.2927 | 0.1748 | 0.1445 | 0.7566 | 0.5997 | 0.4805 | 0.2947 |
| | 0.50 | 0.4397 | 0.5129 | 0.1829 | 0.1253 | 0.7534 | 0.5164 | 0.4825 | 0.3240 |
| | 0.70 | 0.3893 | 0.5129 | 0.2414 | 0.1225 | 0.7534 | 0.5164 | 0.4778 | 0.3214 |
| 10% | 1.00 | 0.4020 | 0.4975 | 0.2020 | 0.1310 | 0.7534 | 0.5164 | 0.4872 | 0.2947 |
| | 2.00 | 0.3980 | 0.4838 | 0.1888 | 0.0921 | 0.7660 | 0.4395 | 0.5184 | 0.3373 |
| | 4.00 | 0.3959 | 0.2943 | 0.2157 | 0.0562 | 0.7500 | 0.3028 | 0.4809 | 0.3014 |
| | 5.00 | 0.4380 | 0.2840 | 0.2050 | 0.0562 | 0.7720 | 0.1481 | 0.4809 | 0.3040 |
| | 7.00 | 0.4242 | 0.3317 | 0.2286 | 0.0562 | 0.7244 | 0.1530 | 0.4798 | 0.2393 |
| | 10.00 | 0.4242 | 0.2145 | 0.1541 | 0.0888 | 0.7453 | 0.1781 | 0.5003 | 0.2880 |

Table 5: **UWC Tuning for WTNPO** ($\alpha = 0.5$). $\downarrow$ / $\uparrow$ indicate smaller / larger values are preferable.

| WTNPO | | Phi-1.5 | | | | Llama-2-7B | | | |
| | | ES-exact | | ES-perturb | | ES-exact | | ES-perturb | |
| setup | $\beta$ | retain $\uparrow$ | unlearn $\downarrow$ | retain $\uparrow$ | unlearn $\downarrow$ | retain $\uparrow$ | unlearn $\downarrow$ | retain $\uparrow$ | unlearn $\downarrow$ |
|---|---|---|---|---|---|---|---|---|---|
| before unlearning | | 0.4433 | 0.5969 | 0.2115 | 0.1605 | 0.8277 | 0.8039 | 0.5302 | 0.4001 |
| | 0.05 | 0.4412 | 0.1538 | 0.2080 | 0.0700 | 0.7343 | 0.0833 | 0.4863 | 0.0767 |
| | 0.10 | 0.4394 | 0.1801 | 0.2052 | 0.0652 | 0.7606 | 0.0679 | 0.4957 | 0.0929 |
| | 0.50 | 0.4142 | 0.1524 | 0.2136 | 0.0677 | 0.7251 | 0.1629 | 0.4976 | 0.0929 |
| | 0.70 | 0.4325 | 0.1524 | 0.1882 | 0.0527 | 0.7874 | 0.1629 | 0.4863 | 0.0865 |
| 1% | 1.00 | 0.4412 | 0.1524 | 0.1948 | 0.0527 | 0.7289 | 0.1121 | 0.4976 | 0.1064 |
| | 2.00 | 0.3944 | 0.1412 | 0.1709 | 0.0527 | 0.6673 | 0.0904 | 0.5152 | 0.3242 |
| | 4.00 | 0.3713 | 0.0620 | 0.2052 | 0.0527 | 0.7040 | 0.0979 | 0.4358 | 0.1252 |
| | 5.00 | 0.4213 | 0.0620 | 0.1799 | 0.0527 | 0.7040 | 0.0979 | 0.5152 | 0.3690 |
| | 7.00 | 0.4315 | 0.0620 | 0.2052 | 0.0813 | 0.7040 | 0.1153 | 0.4974 | 0.1951 |
| | 10.00 | 0.4523 | 0.0647 | 0.2052 | 0.0813 | 0.7040 | 0.1509 | 0.4603 | 0.2975 |
| before unlearning | | 0.4433 | 0.5619 | 0.2115 | 0.2374 | 0.8277 | 0.7735 | 0.5302 | 0.4126 |
| | 0.05 | 0.4374 | 0.3243 | 0.1849 | 0.2479 | 0.7520 | 0.4073 | 0.5122 | 0.4013 |
| | 0.10 | 0.3745 | 0.3848 | 0.2222 | 0.2479 | 0.7494 | 0.4776 | 0.5122 | 0.4013 |
| | 0.50 | 0.4041 | 0.3562 | 0.2414 | 0.1587 | 0.7534 | 0.4044 | 0.5109 | 0.3975 |
| | 0.70 | 0.4080 | 0.4222 | 0.2478 | 0.1867 | 0.7534 | 0.4337 | 0.4809 | 0.3803 |
| 5% | 1.00 | 0.4560 | 0.4222 | 0.2523 | 0.1967 | 0.7476 | 0.4233 | 0.4809 | 0.3645 |
| | 2.00 | 0.4402 | 0.3209 | 0.1841 | 0.1850 | 0.7534 | 0.4085 | 0.4888 | 0.2940 |
| | 4.00 | 0.4433 | 0.3903 | 0.1921 | 0.1619 | 0.7533 | 0.0764 | 0.4872 | 0.1426 |
| | 5.00 | 0.4454 | 0.3792 | 0.2515 | 0.1719 | 0.7691 | 0.1178 | 0.4950 | 0.1690 |
| | 7.00 | 0.4454 | 0.3357 | 0.2133 | 0.1669 | 0.7451 | 0.0777 | 0.5022 | 0.1690 |
| | 10.00 | 0.4454 | 0.3814 | 0.1807 | 0.1694 | 0.7725 | 0.0242 | 0.5319 | 0.2442 |
| before unlearning | | 0.4433 | 0.4799 | 0.2115 | 0.1843 | 0.8277 | 0.8307 | 0.5302 | 0.3099 |
| | 0.05 | 0.4210 | 0.4711 | 0.1829 | 0.1339 | 0.7534 | 0.5363 | 0.4825 | 0.2884 |
| | 0.10 | 0.4601 | 0.4711 | 0.1963 | 0.1425 | 0.7534 | 0.5363 | 0.4809 | 0.2757 |
| | 0.50 | 0.3865 | 0.3518 | 0.2189 | 0.1321 | 0.7534 | 0.5363 | 0.4825 | 0.2677 |
| | 0.70 | 0.4200 | 0.3753 | 0.1676 | 0.0788 | 0.7534 | 0.5363 | 0.5063 | 0.2872 |
| 10% | 1.00 | 0.4322 | 0.3432 | 0.1615 | 0.0538 | 0.7520 | 0.4619 | 0.4842 | 0.2769 |
| | 2.00 | 0.4519 | 0.4117 | 0.2014 | 0.0583 | 0.7720 | 0.3741 | 0.5049 | 0.3335 |
| | 4.00 | 0.3994 | 0.2390 | 0.1854 | 0.0453 | 0.7720 | 0.0446 | 0.5216 | 0.2989 |
| | 5.00 | 0.4223 | 0.1658 | 0.2102 | 0.0974 | 0.7691 | 0.0283 | 0.4809 | 0.2898 |
| | 7.00 | 0.4242 | 0.2035 | 0.1774 | 0.0888 | 0.7484 | 0.0355 | 0.4911 | 0.2118 |
| | 10.00 | 0.4212 | 0.2742 | 0.1633 | 0.0517 | 0.7717 | 0.0355 | 0.4960 | 0.2537 |

Table 6: **UWC Tuning for WTNPO** ($\alpha = 1$). $\downarrow$ / $\uparrow$ indicate smaller / larger values are preferable.

| WTNPO | | Phi-1.5 | | | | Llama-2-7B | | | |
| --- | --- | --- | --- | --- | --- | --- | --- | --- | --- |
| | | ES-exact | | ES-perturb | | ES-exact | | ES-perturb | |
| setup | $\beta$ | retain ↑ | unlearn ↓ | retain ↑ | unlearn ↓ | retain ↑ | unlearn ↓ | retain ↑ | unlearn ↓ |
| before unlearning | | 0.4433 | 0.5969 | 0.2115 | 0.1605 | 0.8277 | 0.8039 | 0.5302 | 0.4001 |
| | 0.05 | 0.4412 | 0.1738 | 0.2052 | 0.0659 | 0.7090 | 0.1376 | 0.4863 | 0.0767 |
| | 0.10 | 0.4412 | 0.1738 | 0.1989 | 0.0659 | 0.7166 | 0.1376 | 0.4879 | 0.0767 |
| | 0.50 | 0.4412 | 0.1738 | 0.1925 | 0.0527 | 0.7713 | 0.1319 | 0.4968 | 0.0767 |
| | 0.70 | 0.4412 | 0.1738 | 0.1861 | 0.0567 | 0.7118 | 0.0840 | 0.4896 | 0.0767 |
| 1% | 1.00 | 0.4412 | 0.1738 | 0.2052 | 0.0619 | 0.7522 | 0.0897 | 0.4896 | 0.0767 |
| | 2.00 | 0.4412 | 0.0647 | 0.1978 | 0.0465 | 0.6497 | 0.0648 | 0.4777 | 0.0793 |
| | 4.00 | 0.4199 | 0.0647 | 0.1969 | 0.0452 | 0.7040 | 0.0736 | 0.4960 | 0.0844 |
| | 5.00 | 0.3790 | 0.0385 | 0.2074 | 0.0527 | 0.7040 | 0.0736 | 0.4955 | 0.1140 |
| | 7.00 | 0.4258 | 0.0425 | 0.1865 | 0.0527 | 0.7040 | 0.0999 | 0.4505 | 0.1505 |
| | 10.00 | 0.4319 | 0.0620 | 0.2070 | 0.0813 | 0.7214 | 0.1359 | 0.5200 | 0.2588 |
| before unlearning | | 0.4433 | 0.5619 | 0.2115 | 0.2374 | 0.8277 | 0.7735 | 0.5302 | 0.4126 |
| | 0.05 | 0.4560 | 0.4082 | 0.2259 | 0.1967 | 0.7534 | 0.3855 | 0.4841 | 0.3697 |
| | 0.10 | 0.4000 | 0.4238 | 0.2242 | 0.1967 | 0.7491 | 0.3754 | 0.4780 | 0.3645 |
| | 0.50 | 0.4320 | 0.4062 | 0.1990 | 0.1063 | 0.7534 | 0.3754 | 0.4888 | 0.2914 |
| | 0.70 | 0.4200 | 0.4062 | 0.1992 | 0.0823 | 0.7463 | 0.4174 | 0.4869 | 0.2837 |
| 5% | 1.00 | 0.4278 | 0.3698 | 0.2557 | 0.1097 | 0.7317 | 0.4240 | 0.4812 | 0.2837 |
| | 2.00 | 0.4029 | 0.2473 | 0.2134 | 0.1203 | 0.7534 | 0.3786 | 0.4848 | 0.2642 |
| | 4.00 | 0.4454 | 0.3853 | 0.2077 | 0.1105 | 0.7658 | 0.0781 | 0.4807 | 0.1971 |
| | 5.00 | 0.4454 | 0.2985 | 0.2227 | 0.1754 | 0.7625 | 0.0681 | 0.4772 | 0.1820 |
| | 7.00 | 0.4254 | 0.2913 | 0.1644 | 0.1679 | 0.7594 | 0.0448 | 0.4795 | 0.1356 |
| | 10.00 | 0.3894 | 0.2826 | 0.1639 | 0.1477 | 0.7887 | 0.0304 | 0.4873 | 0.1871 |
| before unlearning | | 0.4433 | 0.4799 | 0.2115 | 0.1843 | 0.8277 | 0.8307 | 0.5302 | 0.3099 |
| | 0.05 | 0.4810 | 0.2738 | 0.2188 | 0.0595 | 0.7534 | 0.5363 | 0.4779 | 0.2677 |
| | 0.10 | 0.4246 | 0.2024 | 0.2036 | 0.0637 | 0.7534 | 0.4953 | 0.4809 | 0.2884 |
| | 0.50 | 0.4180 | 0.3978 | 0.1639 | 0.0434 | 0.7491 | 0.5030 | 0.5073 | 0.2947 |
| | 0.70 | 0.4540 | 0.3663 | 0.2202 | 0.0417 | 0.7534 | 0.5030 | 0.4989 | 0.2675 |
| 10% | 1.00 | 0.4502 | 0.2201 | 0.1992 | 0.0494 | 0.7513 | 0.3768 | 0.4893 | 0.2989 |
| | 2.00 | 0.4234 | 0.1453 | 0.2065 | 0.0107 | 0.7551 | 0.2972 | 0.5185 | 0.2575 |
| | 4.00 | 0.4205 | 0.1344 | 0.1958 | 0.0193 | 0.7675 | 0.0402 | 0.4792 | 0.2553 |
| | 5.00 | 0.4208 | 0.1260 | 0.1926 | 0.0239 | 0.7691 | 0.0378 | 0.4960 | 0.2255 |
| | 7.00 | 0.3934 | 0.1464 | 0.1557 | 0.1002 | 0.7001 | 0.0335 | 0.4742 | 0.2090 |
| | 10.00 | 0.3860 | 0.1123 | 0.1652 | 0.1132 | 0.7693 | 0.0525 | 0.4943 | 0.2459 |

Table 7: **UWC Tuning for WTNPO** ($\alpha = 1.5$). $\downarrow$ / $\uparrow$ indicate smaller / larger values are preferable.

| WTNPO | | Phi-1.5 | | | | Llama-2-7B | | | |
|---|---|---|---|---|---|---|---|---|---|
| | | ES-exact | | ES-perturb | | ES-exact | | ES-perturb | |
| setup | $\beta$ | retain ↑ | unlearn ↓ | retain ↑ | unlearn ↓ | retain ↑ | unlearn ↓ | retain ↑ | unlearn ↓ |
| before unlearning | | 0.4433 | 0.5969 | 0.2115 | 0.1605 | 0.8277 | 0.8039 | 0.5302 | 0.4001 |
| | 0.05 | 0.4412 | 0.1688 | 0.1925 | 0.0619 | 0.7118 | 0.1319 | 0.4685 | 0.0767 |
| | 0.10 | 0.4412 | 0.1688 | 0.2052 | 0.0619 | 0.7094 | 0.1319 | 0.4911 | 0.0767 |
| | 0.50 | 0.4412 | 0.1412 | 0.2010 | 0.0619 | 0.7141 | 0.0472 | 0.4895 | 0.0398 |
| | 0.70 | 0.4135 | 0.0647 | 0.2052 | 0.0557 | 0.7189 | 0.0679 | 0.4740 | 0.0793 |
| 1% | 1.00 | 0.4327 | 0.0647 | 0.1818 | 0.0619 | 0.6186 | 0.0824 | 0.4798 | 0.0767 |
| | 2.00 | 0.4391 | 0.0647 | 0.1693 | 0.0274 | 0.7021 | 0.0736 | 0.4704 | 0.0844 |
| | 4.00 | 0.4183 | 0.0647 | 0.1963 | 0.0336 | 0.7021 | 0.0736 | 0.4974 | 0.0844 |
| | 5.00 | 0.4173 | 0.0647 | 0.1911 | 0.0425 | 0.7040 | 0.0912 | 0.5022 | 0.1505 |
| | 7.00 | 0.4258 | 0.0500 | 0.2033 | 0.0425 | 0.7040 | 0.1404 | 0.4583 | 0.1428 |
| | 10.00 | 0.4243 | 0.0620 | 0.2053 | 0.0527 | 0.7040 | 0.1521 | 0.4589 | 0.1667 |
| before unlearning | | 0.4433 | 0.5619 | 0.2115 | 0.2374 | 0.8277 | 0.7735 | 0.5302 | 0.4126 |
| | 0.05 | 0.4539 | 0.4062 | 0.2574 | 0.0926 | 0.7505 | 0.3786 | 0.5122 | 0.3783 |
| | 0.10 | 0.4560 | 0.4062 | 0.2374 | 0.0646 | 0.7534 | 0.3911 | 0.4908 | 0.3295 |
| | 0.50 | 0.3934 | 0.2448 | 0.1984 | 0.0672 | 0.7534 | 0.3911 | 0.4888 | 0.3628 |
| | 0.70 | 0.4469 | 0.2448 | 0.1934 | 0.1012 | 0.7505 | 0.3786 | 0.4888 | 0.3295 |
| 5% | 1.00 | 0.4510 | 0.2448 | 0.1791 | 0.1203 | 0.7534 | 0.3786 | 0.4888 | 0.3052 |
| | 2.00 | 0.3915 | 0.3621 | 0.2047 | 0.1067 | 0.7534 | 0.3354 | 0.4828 | 0.2456 |
| | 4.00 | 0.4214 | 0.3393 | 0.2172 | 0.1217 | 0.7533 | 0.0427 | 0.4805 | 0.1257 |
| | 5.00 | 0.4334 | 0.2879 | 0.2247 | 0.1320 | 0.7480 | 0.0753 | 0.4950 | 0.1916 |
| | 7.00 | 0.4454 | 0.2879 | 0.2177 | 0.1154 | 0.7497 | 0.0100 | 0.4796 | 0.1895 |
| | 10.00 | 0.3894 | 0.2071 | 0.2177 | 0.1154 | 0.7570 | 0.0198 | 0.4920 | 0.1342 |
| before unlearning | | 0.4433 | 0.4799 | 0.2115 | 0.1843 | 0.8277 | 0.8307 | 0.5302 | 0.3099 |
| | 0.05 | 0.4262 | 0.1453 | 0.1816 | 0.0091 | 0.7534 | 0.4925 | 0.4852 | 0.2677 |
| | 0.10 | 0.4704 | 0.1625 | 0.1926 | 0.0173 | 0.7534 | 0.4437 | 0.4896 | 0.2677 |
| | 0.50 | 0.4519 | 0.2246 | 0.2185 | 0.0280 | 0.7720 | 0.3792 | 0.4977 | 0.2677 |
| | 0.70 | 0.4145 | 0.1369 | 0.2167 | 0.0453 | 0.7683 | 0.2972 | 0.5154 | 0.2677 |
| 10% | 1.00 | 0.4254 | 0.1253 | 0.2110 | 0.0336 | 0.7720 | 0.0355 | 0.5202 | 0.2842 |
| | 2.00 | 0.4345 | 0.1135 | 0.2090 | 0.0109 | 0.7625 | 0.0149 | 0.4825 | 0.2989 |
| | 4.00 | 0.4234 | 0.1357 | 0.2190 | 0.0120 | 0.7549 | 0.0451 | 0.5133 | 0.2677 |
| | 5.00 | 0.4306 | 0.1347 | 0.1998 | 0.0239 | 0.7807 | 0.0111 | 0.5061 | 0.2952 |
| | 7.00 | 0.3934 | 0.1161 | 0.1660 | 0.1002 | 0.7735 | 0.0043 | 0.4976 | 0.2302 |
| | 10.00 | 0.4149 | 0.1380 | 0.1591 | 0.1002 | 0.7691 | 0.1148 | 0.4911 | 0.2921 |

Table 8: **UWC Tuning for RMU (shallow).** ↓ / ↑ indicate smaller / larger values are preferable.

| RMU | | Phi-1.5 | | | | Llama-2-7B | | | |
|---|---|---|---|---|---|---|---|---|---|
| | | ES-exact | | ES-perturb | | ES-exact | | ES-perturb | |
| setup | $c$ | retain ↑ | unlearn ↓ | retain ↑ | unlearn ↓ | retain ↑ | unlearn ↓ | retain ↑ | unlearn ↓ |
| before unlearning | | 0.4433 | 0.5969 | 0.2115 | 0.1605 | 0.8277 | 0.8039 | 0.5302 | 0.4001 |
| | 0.00 | 0.4530 | 0.5969 | 0.2007 | 0.1855 | 0.7604 | 0.5993 | 0.4888 | 0.3816 |
| | 1.00 | 0.4122 | 0.4356 | 0.2115 | 0.1855 | 0.7502 | 0.6278 | 0.4890 | 0.4253 |
| | 2.00 | 0.4312 | 0.4080 | 0.2072 | 0.1855 | 0.7653 | 0.6714 | 0.4531 | 0.4002 |
| 1% | 4.00 | 0.4245 | 0.4682 | 0.2115 | 0.1855 | 0.7356 | 0.7223 | 0.4758 | 0.4008 |
| | 5.00 | 0.4398 | 0.5149 | 0.1981 | 0.1855 | 0.7163 | 0.6287 | 0.4871 | 0.4008 |
| | 7.00 | 0.4460 | 0.5096 | 0.2201 | 0.1855 | 0.7292 | 0.7128 | 0.4516 | 0.4104 |
| | 10.00 | 0.4215 | 0.4816 | 0.2018 | 0.1855 | 0.7292 | 0.6195 | 0.4453 | 0.4104 |
| before unlearning | | 0.4433 | 0.5619 | 0.2115 | 0.2374 | 0.8277 | 0.7735 | 0.5302 | 0.4126 |
| | 0.00 | 0.4164 | 0.4924 | 0.1918 | 0.2172 | 0.7516 | 0.7292 | 0.4676 | 0.3616 |
| | 1.00 | 0.4284 | 0.5124 | 0.2194 | 0.2172 | 0.7762 | 0.7357 | 0.4677 | 0.4504 |
| | 2.00 | 0.4044 | 0.4774 | 0.1939 | 0.2172 | 0.7146 | 0.6370 | 0.4453 | 0.4126 |
| 5% | 4.00 | 0.4404 | 0.4252 | 0.2047 | 0.2147 | 0.7619 | 0.6758 | 0.4812 | 0.4126 |
| | 5.00 | 0.4404 | 0.4838 | 0.2181 | 0.2207 | 0.7139 | 0.6758 | 0.4812 | 0.4164 |
| | 7.00 | 0.4204 | 0.3772 | 0.2073 | 0.2339 | 0.7604 | 0.6758 | 0.4793 | 0.4126 |
| | 10.00 | 0.4194 | 0.4114 | 0.1903 | 0.2339 | 0.7146 | 0.6370 | 0.4453 | 0.4126 |
| before unlearning | | 0.4433 | 0.4799 | 0.2115 | 0.1843 | 0.8277 | 0.8307 | 0.5302 | 0.3099 |
| | 0.00 | 0.4425 | 0.5761 | 0.2055 | 0.1424 | 0.7887 | 0.8165 | 0.4246 | 0.2662 |
| | 1.00 | 0.4424 | 0.5968 | 0.2133 | 0.1567 | 0.7568 | 0.6869 | 0.4771 | 0.2989 |
| | 2.00 | 0.4304 | 0.5961 | 0.2028 | 0.1360 | 0.7628 | 0.6755 | 0.4690 | 0.2989 |
| 10% | 4.00 | 0.4364 | 0.5208 | 0.1944 | 0.1547 | 0.7229 | 0.5784 | 0.4812 | 0.2766 |
| | 5.00 | 0.4284 | 0.5184 | 0.2007 | 0.1547 | 0.7262 | 0.6268 | 0.4797 | 0.2944 |
| | 7.00 | 0.4404 | 0.5184 | 0.2007 | 0.1754 | 0.7271 | 0.5778 | 0.4232 | 0.3033 |
| | 10.00 | 0.4404 | 0.4693 | 0.2136 | 0.1675 | 0.7032 | 0.5455 | 0.4849 | 0.3033 |

Table 9: **UWC Tuning for RMU (middle).** ↓ / ↑ indicate smaller / larger values are preferable.

| RMU | | Phi-1.5 | | | | Llama-2-7B | | | |
|---|---|---|---|---|---|---|---|---|---|
| | | ES-exact | | ES-perturb | | ES-exact | | ES-perturb | |
| setup | $c$ | retain ↑ | unlearn ↓ | retain ↑ | unlearn ↓ | retain ↑ | unlearn ↓ | retain ↑ | unlearn ↓ |
| before unlearning | | 0.4433 | 0.5969 | 0.2115 | 0.1605 | 0.8277 | 0.8039 | 0.5302 | 0.4001 |
| | 0.00 | 0.4203 | 0.5969 | 0.2153 | 0.2069 | 0.7606 | 0.5127 | 0.5115 | 0.4001 |
| | 1.00 | 0.4203 | 0.5969 | 0.2180 | 0.1409 | 0.7416 | 0.5093 | 0.4878 | 0.4001 |
| | 2.00 | 0.4203 | 0.5969 | 0.1831 | 0.1261 | 0.7512 | 0.4263 | 0.4644 | 0.3794 |
| 1% | 4.00 | 0.4203 | 0.5969 | 0.1831 | 0.1261 | 0.7559 | 0.5093 | 0.4096 | 0.3538 |
| | 5.00 | 0.4203 | 0.5969 | 0.2073 | 0.1328 | 0.7413 | 0.4810 | 0.4927 | 0.4001 |
| | 7.00 | 0.4218 | 0.5969 | 0.2119 | 0.1261 | 0.7413 | 0.4810 | 0.4927 | 0.4001 |
| | 10.00 | 0.4203 | 0.5969 | 0.2119 | 0.1350 | 0.7655 | 0.4137 | 0.4927 | 0.3624 |
| before unlearning | | 0.4433 | 0.5619 | 0.2115 | 0.2374 | 0.8277 | 0.7735 | 0.5302 | 0.4126 |
| | 0.00 | 0.4262 | 0.5723 | 0.1952 | 0.2207 | 0.8017 | 0.6376 | 0.4754 | 0.3884 |
| | 1.00 | 0.4232 | 0.4999 | 0.2032 | 0.2207 | 0.7381 | 0.4284 | 0.4798 | 0.3884 |
| | 2.00 | 0.4232 | 0.5013 | 0.2229 | 0.2207 | 0.7179 | 0.5146 | 0.4379 | 0.3884 |
| 5% | 4.00 | 0.4218 | 0.5309 | 0.1887 | 0.2030 | 0.7112 | 0.4034 | 0.4927 | 0.3884 |
| | 5.00 | 0.3578 | 0.3762 | 0.2119 | 0.2030 | 0.7438 | 0.6323 | 0.4927 | 0.3884 |
| | 7.00 | 0.4218 | 0.5946 | 0.1990 | 0.1971 | 0.7438 | 0.6684 | 0.4927 | 0.4126 |
| | 10.00 | 0.4262 | 0.4000 | 0.1968 | 0.2005 | 0.7552 | 0.6615 | 0.4644 | 0.4126 |
| before unlearning | | 0.4433 | 0.4799 | 0.2115 | 0.1843 | 0.8277 | 0.8307 | 0.5302 | 0.3099 |
| | 0.00 | 0.4262 | 0.4584 | 0.1952 | 0.1786 | 0.7463 | 0.6152 | 0.4754 | 0.3884 |
| | 1.00 | 0.4203 | 0.4909 | 0.2108 | 0.1816 | 0.7493 | 0.7636 | 0.4379 | 0.3139 |
| | 2.00 | 0.4232 | 0.5025 | 0.2212 | 0.1786 | 0.7374 | 0.7275 | 0.4831 | 0.3158 |
| 10% | 4.00 | 0.4394 | 0.5025 | 0.2117 | 0.1901 | 0.7874 | 0.7526 | 0.4871 | 0.3196 |
| | 5.00 | 0.4224 | 0.4511 | 0.2117 | 0.1799 | 0.7874 | 0.6907 | 0.4653 | 0.3220 |
| | 7.00 | 0.4005 | 0.4568 | 0.1496 | 0.1741 | 0.7434 | 0.5821 | 0.4776 | 0.2908 |
| | 10.00 | 0.4522 | 0.4938 | 0.1542 | 0.2000 | 0.7534 | 0.6495 | 0.4927 | 0.3316 |

Table 10: **UWC Tuning for RMU (deep).** ↓ / ↑ indicate smaller / larger values are preferable.

| UWC | | Phi-1.5 | | | | Llama-2-7B | | | |
|---|---|---|---|---|---|---|---|---|---|
| | | ES-exact | | ES-perturb | | ES-exact | | ES-perturb | |
| setup | $c$ | retain ↑ | unlearn ↓ | retain ↑ | unlearn ↓ | retain ↑ | unlearn ↓ | retain ↑ | unlearn ↓ |
| before unlearning | | 0.4433 | 0.5969 | 0.2115 | 0.1605 | 0.8277 | 0.8039 | 0.5302 | 0.4001 |
| | 0.00 | 0.3936 | 0.5219 | 0.2136 | 0.1574 | 0.7836 | 0.6364 | 0.4927 | 0.4089 |
| | 1.00 | 0.4156 | 0.5219 | 0.2117 | 0.1574 | 0.7461 | 0.4564 | 0.4442 | 0.3402 |
| | 2.00 | 0.4212 | 0.5219 | 0.2080 | 0.1655 | 0.6977 | 0.2814 | 0.4847 | 0.2790 |
| 1% | 4.00 | 0.4212 | 0.5153 | 0.1951 | 0.1655 | 0.6913 | 0.2992 | 0.4428 | 0.2748 |
| | 5.00 | 0.4212 | 0.5121 | 0.2062 | 0.1655 | 0.7122 | 0.3974 | 0.4976 | 0.1982 |
| | 7.00 | 0.4212 | 0.5108 | 0.1885 | 0.1686 | 0.7509 | 0.3271 | 0.4428 | 0.2305 |
| | 10.00 | 0.4184 | 0.4963 | 0.2136 | 0.1717 | 0.7106 | 0.3815 | 0.4428 | 0.2062 |
| before unlearning | | 0.4433 | 0.5619 | 0.2115 | 0.2374 | 0.8277 | 0.7735 | 0.5302 | 0.4126 |
| | 0.00 | 0.4212 | 0.4953 | 0.2007 | 0.2182 | 0.7731 | 0.7074 | 0.4675 | 0.3953 |
| | 1.00 | 0.4049 | 0.5144 | 0.2115 | 0.2182 | 0.7731 | 0.6488 | 0.4801 | 0.3850 |
| | 2.00 | 0.4110 | 0.5602 | 0.1967 | 0.2227 | 0.7410 | 0.6683 | 0.4801 | 0.3714 |
| 5% | 4.00 | 0.4151 | 0.5621 | 0.1930 | 0.2227 | 0.7731 | 0.6031 | 0.4598 | 0.3869 |
| | 5.00 | 0.4212 | 0.5271 | 0.2099 | 0.2394 | 0.7464 | 0.7001 | 0.4613 | 0.3958 |
| | 7.00 | 0.4212 | 0.5285 | 0.1951 | 0.2394 | 0.8113 | 0.6983 | 0.5015 | 0.4464 |
| | 10.00 | 0.4064 | 0.4816 | 0.2025 | 0.2349 | 0.7319 | 0.7763 | 0.4600 | 0.4393 |
| before unlearning | | 0.4433 | 0.4799 | 0.2115 | 0.1843 | 0.8277 | 0.8307 | 0.5302 | 0.3099 |
| | 0.00 | 0.4212 | 0.4935 | 0.2095 | 0.1933 | 0.7577 | 0.6868 | 0.4410 | 0.2884 |
| | 1.00 | 0.4049 | 0.4935 | 0.2039 | 0.1963 | 0.7673 | 0.7560 | 0.4571 | 0.2906 |
| | 2.00 | 0.4212 | 0.4935 | 0.1969 | 0.1933 | 0.7731 | 0.7402 | 0.4865 | 0.3239 |
| 10% | 4.00 | 0.4212 | 0.4935 | 0.2115 | 0.1933 | 0.7731 | 0.7414 | 0.4426 | 0.2674 |
| | 5.00 | 0.4212 | 0.4959 | 0.1967 | 0.1933 | 0.7486 | 0.7688 | 0.4738 | 0.2192 |
| | 7.00 | 0.4212 | 0.4799 | 0.2097 | 0.1933 | 0.7620 | 0.7402 | 0.4784 | 0.2547 |
| | 10.00 | 0.3934 | 0.4799 | 0.1951 | 0.1786 | 0.7394 | 0.7402 | 0.4890 | 0.2547 |

