# OpenReview forum: "Rethinking LLM Unlearning Objectives: A Gradient Perspective and Go Beyond"
_ICLR.cc/2025/Conference — ICLR 2025 Poster_

### Official Review · Reviewer_oKM6 · 2024-11-01

**Soundness:** 3
**Presentation:** 3
**Contribution:** 2
**Rating:** 6
**Confidence:** 3

**Summary:**

This paper focus on analysis of existing unlearning methods in the scope of LLMs. The authors propose a general toolkit for analysis of existing methods, which is named G-effect. G-effect analyzes unlearning methods from both forgetting and retaining. Experiments show the rationality of the proposed G-effect, and the effectiveness of several proposed variants.

**Strengths:**

- The paper is well structured.
- The authors propose G-effect to analyze existing unlearning methods from the perspectives of both forgetting and retaining effects.
- Analysis of G-effect somehow accords with the experimental results. Experiments show the effectiveness of the proposed variants.

**Weaknesses:**

- What is the rationale of designing WGA? The authors did not clearly state the reasons of choosing such format.
- I do not clearly see the challenges of coming up such a general toolkit for analysis of various unlearning methods.
- The claim in Line 081 is somehow too strong. How did the authors conclude that KL is the optimal choice?
- The presentations of Figure 2,3,4 are somehow hard to read. Putting the legends near the figures might be better.

**Questions:**

no

---

> ### Author Response · Authors · 2024-11-21
> **Responses to Reviewer oKM6**
>
> Many thanks for your constructive comments and suggestions! Please see our responses below.
>
> > **Q1**. What is the rationale of designing WGA? The authors did not clearly state the reasons for choosing such a format.
>
> **A1**. We apologize for any confusion that may have arisen. Following Eq.1, the inverse confidence term within the gradient of GA leads to excessive unlearning, which in turn has severe side effects on model retention. It motivates us to propose WGA, as outlined in Eq 2, **to counteract the negative impacts of the inverse confidence**. By comparing the gradients of GA and WGA, it is evident that the side impacts of the inverse confidence can be mitigated through the confidence weighting within WGA. For example, when $\alpha=1$, the inverse confidence term will disappear from Eq. 1, thereby we can fully remove the impacts of the inverse confidence. We will add more discussion in our revised manuscript. Thank you very much for your comments.
>
>
> > **Q2**. I do not clearly see the challenges of coming up such a general toolkit for analysis of various unlearning methods.
>
> **A2**. Currently, numerous studies are proposing new unlearning methods or objectives. **However, to the best of our knowledge, there is no systematic tool available that elucidates their unlearning mechanisms**. Our work is the first one to fill this gap by exploring the gradient dynamics. As demonstrated in Section 4, understanding these underlying mechanisms is essential.
>
> For example, we observe that GA exhibits excessively large G-effects during unlearning, which we have traced back to the inverse confidence term in its formulation. This insight led us to make a slight modification to the original GA, yet notably enhancing its overall unlearning efficacy. Additionally, by analyzing the G-effects, we are able to identify tokens that are especially beneficial for unlearning, which deepens our understanding of why NPO is so effective in practice. We also discover that the NPO weighting mechanism is imperfect, as it sometimes overlooks data points that are crucial for unlearning but have minimal impact on retention. This observation presents us with the open question of how to develop a more effective weighting mechanism to further improve the unlearning efficacy.
>
> We are always happy to further discuss the contributions and the uniqueness of our work, and welcome any new comment or concern. Thank you very much!
>
> > **Q3**. The claim in Line 081 is somehow too strong. How did the authors conclude that KL is the optimal choice?
>
> **A3**. Thank you very much for your comments! Given that the magnitude of unlearn G-effects typically surpasses that of retain G-effects for most of the existing unlearning objective, we can ensure that, with regularization, the overall unlearn G-effects will remain negative. Consequently, our focus primarily shifts to the retain G-effects. Positive values in the retain G-effects are beneficial for maintaining common model functionality, with larger magnitudes indicating stronger retention. Therefore, overall, KL outperforms GD. We will incorporate this discussion into our revised manuscript.
>
> > **Q4**. The presentations of Figure 2,3,4 are somehow hard to read. Putting the legends near the figures might be better.
>
> **A4**. Sincere thanks for your concerns! The x- and y-axes denote the unlearning steps and the values of G-effects, respectively. We will update our figures to make them clearer in our revision.

---

> > ### Comment · Reviewer_oKM6 · 2024-11-25
> >
> > Thanks for the responses. However, unfortunately, none of my concerns are addressed.
> >
> > For Q1, I mean there could be many mathematical expressions to function as *inverse confidence*. For example, $\frac{1}{A}$, $\frac{1}{A^2}$, $\frac{1}{\sqrt{A}}$. Why did you specifically choose the current form. What is the rationale?
> >
> > For Q2, I still find it unclear what the challenging aspects of your work are.
> >
> > For Q3, again, the regularization term could be in many mathematical forms. For example, l2 regularization, kl regularization. How could you just claim that kl regularization is optimal/best?

---

> > > ### Author Response · Authors · 2024-11-25
> > > **Further Responses to Reviewer oKM6**
> > >
> > > We would like to express our sincere gratitude to receive your feedback! We are so sorry to hear that our earlier responses did not address your concerns and we are glad to further clarify them with you.
> > >
> > > > For Q1, I mean there could be many mathematical expressions to function as inverse confidence. For example, $1/A$,$1/A^2$,$1/\sqrt{A}$. Why did you specifically choose the current form. What is the rationale?
> > >
> > > **A1**. In your initial query, you inquired about the rationale behind designing WGA. Here, your questions seem to suggest that you are also interested in the formulation of inverse confidence. We are more than happy to discuss both issues with you!
> > >
> > > The formulation of **inverse confidence** is directly derived from the gradient of the GA risk, which originates from the logarithmic operation involved in calculating the GA risk. The gradient has a closed-form formulation that is infeasible to be generalized to include additional variants of power functions as you mentioned.
> > >
> > > The formulation of **WGA weighting** is driven by the need to mitigate the drawbacks of inverse confidence (i.e., excessive unlearning), which can notably degrade the overall model utility. To counteract its adverse effects, we can adjust the weighting of the GA risk in a token-wise manner, following $p(s^i_u|s^{<i}_u;\theta)$. It directly addresses the issues stemming from inverse confidence.
> > >
> > > To make such a weighting mechanism more flexible, we employ its power function form as outlined in Eq 2, where $\alpha$ modulates the strength used to counteract the inverse confidence term. As you can observe, this formulation encompasses the variants of power functions that you mentioned, providing a more general approach for adjustment.
> > >
> > > We will include additional discussions in our revision, and we sincerely appreciate the further concerns you have raised. Thank you for your feedback!
> > >
> > > > For Q2, I still find it unclear what the challenging aspects of your work are.
> > >
> > > **A2**. In case we misunderstood your meanings, we would like to highlight the uniqueness as well as the limitations of our works.
> > >
> > > - **Uniqueness**. The G-effect tackles previous challenges in systematically analyzing the behaviors and dynamics of various unlearning objectives. To our knowledge, we are the first to explore this aspect by characterizing their gradient dynamics. As detailed in Sec 4, comprehending these underlying mechanisms is essential for advancing the field.
> > >
> > > - **Limitations**. During the development of our tool, we rely on a first-order approximation and the assumption that the (un)learning rate is small. These factors are critical for the success of our G-effect, otherwise we would need to compute the inverse of the Hessian matrix, of which the computational costs can be prohibitively high for LLMs. However, on the other side, it makes the G-effects omit critical information about the unlearning smoothness. Moreover, our derivation is based on the cross-entropy risk as the unlearning metric. It offers many benefits, such as being differentiable and easy to compute. However, it remains an open question whether cross-entropy loss is the optimal choice to characterize the efficacy of unlearning, especially given that model likelihood can sometimes be misleading to characterize knowledge parameterization. The related discussions are detailed in Sec 6 (Drawbacks of G-effects) in our manuscript.
> > >
> > > > For Q3, again, the regularization term could be in many mathematical forms. For example, l2 regularization, kl regularization. How could you just claim that kl regularization is optimal/best?
> > >
> > > **A3**. To begin with, we would like to claim that **a proper regularization term should have high values of the retain G-effects**; the larger these values, the more effective the regularization. Its rationale raises based on two factors of our observations.
> > >
> > > - The unlearn G-effects for unlearning objectives are much larger than that for regularizations, indicating that the side effects of the regularization term on unlearning efficacy are controllable.
> > >
> > > - The retain G-effects for unlearning objectives are typically negative, which necessitates a strong strength of the retain G-effects to restore the common performance.
> > >
> > > In Section 4.4, we compare the retain G-effects between GD and KL. Our findings show that KL outperforms GD, evidenced by the higher values in both the maximum (max) and the summation (sum) of the retain G-effects, further quantifying as follows. Many thanks for your comments and we will add the related discussions in our revision.
> > >
> > > |    | sum | max |
> > > |:--:|:---:|:---:|
> > > | GD | 234 |  57 |
> > > | KL | 337 | 111 |

---

> > > > ### Comment · Reviewer_oKM6 · 2024-11-28
> > > >
> > > > Thanks for the responses. For Question 3, you should avoid claiming that the KL divergence is optimal unless there is theoretical support to substantiate this assertion.

---

> > > > > ### Author Response · Authors · 2024-11-28
> > > > > **Thank you**
> > > > >
> > > > > We agree with your opinion and thank you for your suggestions! We will be very careful about this issue in our revision.

---

> ### Author Response · Authors · 2024-11-24
> **Many Thanks for Your Support and Looking Forward to Further Discussions**
>
> Dear Reviewer oKM6,
>
> We sincerely thank you for your efforts in reviewing our work and for your great support! We also greatly appreciate your insightful questions and hope that our responses have helped to clarify them.
>
> Please let us know if you need any further information or if there are additional points you would like to discuss with us. Thank you very much!
>
> Best regards,
>
> Authors of #1253

---

### Official Review · Reviewer_uXUc · 2024-11-01

**Soundness:** 3
**Presentation:** 3
**Contribution:** 2
**Rating:** 5
**Confidence:** 3

**Summary:**

This paper tackles the concept of unlearning in Large Language Models (LLMs), focusing on the removal of learned knowledge while preserving the overall model integrity. The authors propose a new metric, the G-effect, to quantify the impact of unlearning objectives on model performance.

**Strengths:**

- Investigating unlearning dynamics is an interesting and understudied area.

- Empirical results back up some of the authors claims.

**Weaknesses:**

See questions and comments.

**Questions:**

Legal compliance has been an oft touted reason for unlearning. I have yet to see any compelling argument that unlearning passes any kind of legal bar for data removal. I’m not even sure what the requirements are for meeting e.g. GDPR requirements. I am concerned that the unlearning community may be operating in a vacuum, failing to actively engage with the legal community to determine the practical applications and implications of unlearning.

–

“It is common the cases where shallow layers are more affected than deeper layers during unlearning. It suggests that general knowledge, predominantly encoded in shallow layers (Patil et al., 2023), undergoes substantial alterations“ Why not freeze shallow layers during unlearning using methods like [1]?

[1] Goel, Shashwat, et al. "Towards adversarial evaluations for inexact machine unlearning." arXiv preprint arXiv:2201.06640 (2022).

–

“Unlearning compromises retention. Although conceptually existing (cf., Section 3), current
unlearning objectives all fail to retain the overall model performance when unlearning.“ This is not a new contribution. Many prior works have made this discovery. I also do not understand the difference between this statement and the next contribution: “ Excessive unlearning is harmful. An excessive extent of unlearning has severe impacts such that the deterioration in common model responses can outweigh improvements in unlearning.”

–

I do not believe the contributions highlighted are particularly interesting. The first three listed are already well known phenomena (the first being the motivation for prior unlearning techniques in [1]). The fifth contribution has already been studied. I do think the fourth contribution: “Risk weighting is powerful. Prioritizing certain beneficial points is justified to be effective for unlearning. However, there still exists a large space to further refine risk weighting mechanisms.” is interesting, but I’m not sure this amounts to a significant contribution to the field.

–

The unlearning objectives designed for concept removal assumes that one can pinpoint and incorporate the specific data requiring removal into the unlearning dataset denoted as D_u. I do not believe this is realistic in general.

–

“Removal. The performance on the unlearning dataset Du should significantly deteriorate, i.e., R(Du; θu) ≫ R(Du; θo), revealing effective unlearning on data targeted to be erased.“ Throughout this paper, privacy is highlighted as a use case for unlearning. This is not a good removal metric for anything to do with private information. For example, imagine I want to unlearn “Alice’s phone number is 12345”, and I do this by gradient ascent, up to the point where loss(“Alice’s phone number is 12345”) >> loss(“Alice’s phone number is *any other number*”), then this becomes an oracle, and reconstruction or identification of private information becomes easy. Multiple prior works have discussed this and how to define unlearning for privacy.
Similarly, “We consider the practical objective of erasing targeted knowledge as much as possible (Liu et al., 2024), diverging from the classical definition of machine unlearning (Bourtoule et al., 2021) that seeks to make models behave as if they were trained without the targeted data. Our goal is more suitable for LLM unlearning, driven by the need to eliminate content that poses privacy and copyright concerns, with the understanding that more thorough elimination leads to more favorable behaviors.“ I believe any unlearning definition for privacy that does not try to align with a model that never trained on that data is bad for privacy. It also not clear what kind of privacy we should be concerned about. Reconstruction? Identification of membership?

—

“Sadly, merely comparing performance provides limited insights into understanding the underlying mechanisms.“ What's the definition of “understand” here? Can you formalize it?

—

“Generally speaking, the G-effect compares the gradients of the unlearning objective Lu and the risk metric R. If the gradients of Lu align in similar directions to R, model updating based on Lu
is capable to enhance model performance measured by R, an obvious alternative of R(D; θu) −
R(D; θo) to measure the performance change“ Apologies, I’m a bit confused here. Why not directly optimize R then if it is differentiable?

–

In Figure 1, it’s not clear how the intersection actually maps to successful unlearning. It would be really useful to give some examples (or quantitative results) for various points on the sphere, showing that unlearning is most successful at the intersection.

–

“Due to the high costs in fully computing the G-effects, we focus on experiments based on 5% TOFU fictitious unlearning (Maini et al., 2024) with Llama-2-7B (Touvron et al., 2023a) (cf. Appendix B). All the methods will run for 5 epochs, totaling about 60 steps.
“ This seems like a significant barrier for using G-effect. Also does this mean you only use a dataset of 40 examples? This seems quite small.
–

“The G-Effects across Unlearning Steps” I don’t see what the useful insights are here. I believe we could have instead measured the NLL of examples and gotten an equally useful signal. Overall I found it difficult to assess if the G-Effect is a more useful metric than directly measuring losses over unlearning.

---

> ### Author Response · Authors · 2024-11-21
> **Responses to Reviewer uXUc (part 1/4)**
>
> Many thanks for your constructive comments and suggestions! Please see our responses below.
>
> > **Q1**. Legal compliance has been an oft touted reason for unlearning. I have yet to see any compelling argument that unlearning passes any kind of legal bar for data removal. I’m not even sure what the requirements are for meeting e.g. GDPR requirements. I am concerned that the unlearning community may be operating in a vacuum, failing to actively engage with the legal community to determine the practical applications and implications of unlearning.
>
> **A1**. We are always happy to discuss the role of unlearning in the context of LLMs, which remains an interesting and open question. **From a practical perspective**, as you noted, the GDPR requirement, especially for the so-called “right to be forgotten” (kindly please check https://gdpr-info.eu/chapter-3/ for the details), is one of the key reasons that contribute to the growing interest in LLM unlearning. However, sadly, the technical advancements in unlearning remain notably limited. Several critical challenges persist, including navigating the notorious trade-off between unlearning and retention, controlling the scope of unlearning, reducing dependence on retained data, among many others. Addressing these issues is essential before we can fully rely on unlearning for practical LLM applications. Nevertheless, this does not render the entire field of LLM unlearning meaningless. On the contrary, it indicates that there are still many compelling challenges that must be addressed to unlock its potential for commercial use.
>
> **From a research perspective**, we also recognize that unlearning is not a technique without merit. Similar mechanisms can be observed in the field of LLM alignment, which can be loosely interpreted as simultaneously engaging in unlearning and fine-tuning. Therefore, progress in unlearning—such as investigating why common model functions become corrupted—may also enhance research in alignment. Additionally, many researchers are examining the OOD generalization capabilities of foundation models. Yet, due to data contamination, it is often unclear whether a model has learned from particular data or has a strong ability in generalization. In this context, unlearning can ensure that the candidate OOD data of interest is not directly parameterized into the models, thereby providing more accurate assessments of the generalization capabilities of these systems.
>
> However, we have to acknowledge that there is still a long way to go for LLM unlearning. In addition to the aforementioned challenges, there is also a notable absence of appropriate evaluation frameworks and controllable benchmarks in this field: There is not yet a consensus on the types of metrics that should be used for accurate evaluations of LLM unlearning. Compounding the issue, there is currently no unified definition for the formal goal of LLM unlearning. Moreover, many newly proposed datasets either consist of synthesized data or do not guarantee that the crafted data behaviors can be observed within pre-training data. Nonetheless, we should not view these challenges too negatively. The development of every field requires time, and we can make meaningful contributions and see benefits emerge from this field, both in practical applications and in foundational research.

---

> ### Author Response · Authors · 2024-11-21
> **Responses to Reviewer uXUc (part 2/4)**
>
> > **Q2**. “It is common the cases where shallow layers are more affected than deeper layers during unlearning. It suggests that general knowledge, predominantly encoded in shallow layers (Patil et al., 2023), undergoes substantial alterations“ Why not freeze shallow layers during unlearning using methods like [1]?
>
> **A2**. We sincerely thank you for your wonderful suggestions! Due to the time limit, we would like to first report the results of GA and WGA with shallow layers frozen (layers 1-11), following the same experimental setups as Fig 3. Below, we summarize the results with respect to PS scores. We will add the related discussions in our revision. Thanks again for your suggestions!
>
> |            | PS-exact retain | PS-exact unlearn | PS-perturb retain | PS-perturb unlearn |
> |:----------:|:---------------:|:----------------:|:-----------------:|:------------------:|
> |     GA     |      0.0733     |      0.0000      |       0.0566      |       0.0000       |
> |     WGA    |      0.5005     |      0.0057      |       0.3376      |       0.0041       |
> |  GA frozen |      0.0005     |      0.0010      |      0.0000       |       0.0000       |
> | WGA frozen |      0.5862     |      0.0089      |       0.3858      |       0.0093       |
>
>
> > **Q3 & 4**. “Unlearning compromises retention. Although conceptually existing (cf., Section 3), current unlearning objectives all fail to retain the overall model performance when unlearning.“ This is not a new contribution. Many prior works have made this discovery. I also do not understand the difference between this statement and the next contribution: “ Excessive unlearning is harmful. An excessive extent of unlearning has severe impacts such that the deterioration in common model responses can outweigh improvements in unlearning.”
>
> I do not believe the contributions highlighted are particularly interesting. The first three listed are already well known phenomena (the first being the motivation for prior unlearning techniques in [1]). The fifth contribution has already been studied. I do think the fourth contribution: “Risk weighting is powerful. Prioritizing certain beneficial points is justified to be effective for unlearning. However, there still exists a large space to further refine risk weighting mechanisms.” is interesting, but I’m not sure this amounts to a significant contribution to the field.
>
> **A3 & 4**. To clarify, the main contributions of our research include **the development of a general analysis tool for examining the G-effect and its applications in analyzing various unlearning objectives**. Leveraging this analysis, we have proposed several simple yet effective unlearning objectives, such as WGA and TNPO, which have demonstrated empirical benefits.
>
> In our introduction, we aim to provide an overview of the broader challenges in the field of LLM unlearning for those new to this area, aligning with our empirical findings. We appreciate your feedback suggesting that this part might be misinterpreted as a summary of our principal contributions. To address this, we will relocate the discussion to the "Conclusion and Future Directions" section. Therein, we will emphasize that these issues are ongoing and require further attention, highlighting them as open questions that merit continued research and exploration.
>
> > **Q5**. The unlearning objectives designed for concept removal assumes that one can pinpoint and incorporate the specific data requiring removal into the unlearning dataset denoted as D_u. I do not believe this is realistic in general.
>
> **A5**. Please kindly let us know if we misinterpret your question but it appears that we have not made any claims of this, nor have we mentioned the removal of concepts. We would appreciate your correction if there has been a mistake on our side.

---

> ### Author Response · Authors · 2024-11-21
> **Responses to Reviewer uXUc (part 3/4)**
>
> > **Q6**. “Removal. The performance on the unlearning dataset Du should significantly deteriorate, i.e., R(Du; θu) ≫ R(Du; θo), revealing effective unlearning on data targeted to be erased.“ Throughout this paper, privacy is highlighted as a use case for unlearning. This is not a good removal metric for anything to do with private information. For example, imagine I want to unlearn “Alice’s phone number is 12345”, and I do this by gradient ascent, up to the point where loss(“Alice’s phone number is 12345”) >> loss(“Alice’s phone number is any other number”), then this becomes an oracle, and reconstruction or identification of private information becomes easy. Multiple prior works have discussed this and how to define unlearning for privacy. Similarly, “We consider the practical objective of erasing targeted knowledge as much as possible (Liu et al., 2024), diverging from the classical definition of machine unlearning (Bourtoule et al., 2021) that seeks to make models behave as if they were trained without the targeted data. Our goal is more suitable for LLM unlearning, driven by the need to eliminate content that poses privacy and copyright concerns, with the understanding that more thorough elimination leads to more favorable behaviors.“ I believe any unlearning definition for privacy that does not try to align with a model that never trained on that data is bad for privacy. It is also not clear what kind of privacy we should be concerned about. Reconstruction? Identification of membership?
>
> **A6**. We believe your mentioned classical goals of unlearning in eliminating influence are critical and wide-recognized. However, **both your considered and our goals in unlearning share many facets in common**. For example, both pursue full retention and reduce the likelihood of generating targeted data. Therefore, for many common-existing challenges, such as the notorious trade-off between removal and retention, our manuscript can benefit both goals of unlearning. Therefore, this manuscript can benefit the broad area with different focus.
>
> We also understand that, over our considered goal, eliminating influence necessitates further control over the extent of data removal, ensuring that the unlearned models should behave as if they were pre-trained without privacy data. Generally speaking, it is hard to get such a gold standard for assessment, and more information should be provided for unlearning methods to ensure precise control. Also, current wide-adopted setups typically do not fix new targets/outputs for data required to be unlearned, therefore we do not have the new targets like “Alice’s phone number is any other number”. Expecting models to automatically correct their responses is also not very practical.
>
> Given the challenges of the influence eliminating setups and many unresolved questions that are general within the community, we believe it is pragmatic to pursue a simpler yet still meaningful setup as ours. We also recognize the challenges and changes towards your suggested setups. We will include a discussion on these points in our revised manuscript. Additionally, we will take your mentioned goal of influence elimination as an important direction for our future research.
>
> > **Q7 & 8**. “Sadly, merely comparing performance provides limited insights into understanding the underlying mechanisms.“ What's the definition of “understand” here? Can you formalize it? “Generally speaking, the G-effect compares the gradients of the unlearning objective Lu and the risk metric R. If the gradients of Lu align in similar directions to R, model updating based on Lu is capable to enhance model performance measured by R, an obvious alternative of R(D; θu) − R(D; θo) to measure the performance change“ Apologies, I’m a bit confused here. Why not directly optimize R then if it is differentiable?
>
> **A7 & 8**. Judging solely by performance, we can assess and compare the efficacy of different unlearning methods, but we fail to understand the factors contributing to their efficiencies and deficiencies. The term “understand” here signifies our aim to go beyond mere performance comparisons; we seek to identify factors that influence observed behaviors and determine if there are reliable mechanisms and general techniques to mitigate these drawbacks.
>
> Gradient behaviors often reveal more information than mere changes in performance. The exploration of gradient behaviors,  for example, following Eqs. 1 and 5, has provided deeper insights into why GA leads to excessive unlearning and why NPO significantly outperforms GA. In our revision, we will further emphasize the advantage of analyzing gradient behaviors over simply observing performance changes.
>
> For your concerns about the difference between $R(D;\theta_{u})-R(D;\theta_{o})$ and $R(D;\theta_{u})$, we highlight that they are quite aligned in our setup. The G-effects can either characterize the change of $R(D;\theta_{u})-R(D;\theta_{o})$ or $R(D;\theta_{u})$ directly.

---

> ### Author Response · Authors · 2024-11-21
> **Responses to Reviewer uXUc (part 4/4)**
>
> > **Q9**. In Figure 1, it’s not clear how the intersection actually maps to successful unlearning. It would be really useful to give some examples (or quantitative results) for various points on the sphere, showing that unlearning is most successful at the intersection.
>
> **A9**. We sincerely apologize for any confusion our explanations may have caused. In Fig 1, the gradient behaviors are divided into four distinct regions:
> 1. **Region 1 (Blue Region not intersecting with Red)**: Objectives with gradients in this region excel at retention but are not effective at unlearning.
> 2. **Region 3 (Red Region not intersecting with Blue)**: Objectives in this region demonstrate proficiency in both unlearning and retention.
> 3. **Region 2 (Intersection of Red and Blue Regions)**: Objectives here are effective at unlearning but struggle with retention.
> 4. **Region 4 (White Region)**: Objectives here are ineffective at both unlearning and retention.
>
> In summary, Regions 1 and 3 exhibit a trade-off between unlearning and retention. Region 2 contains ideal objectives for unlearning, whereas Region 4 is unsuitable for unlearning objectives. We will clarify the explanation of Figure 1 in our revision to ensure better understanding.
>
> Our experiments, detailed in Sections 4 and 5, substantiate our claims to be correct. For example, when evaluating the unlearn G-effects of NPO and GA (as shown in Figures 3 and 4), we observe that GA exhibits a greater magnitude compared to NPO. This indicates a stronger capability of GA in removing targeted data, as further evidenced by the results presented in Table 1. Similarly, when comparing the retain G-effects between GA and WGA (as shown in Figure 3), the effect magnitude for WGA is notably smaller than that of GA, demonstrating the superior ability of NPO to maintain original performance, also detailed in Table 1. We will clarify these observations further in the revised version of our manuscript.
>
> > **Q10**. “Due to the high costs in fully computing the G-effects, we focus on experiments based on 5% TOFU fictitious unlearning (Maini et al., 2024) with Llama-2-7B (Touvron et al., 2023a) (cf. Appendix B). All the methods will run for 5 epochs, totaling about 60 steps. “ This seems like a significant barrier for using G-effect. Also does this mean you only use a dataset of 40 examples? This seems quite small.
>
> **A10**. In fact, in the standard 5% TOFU unlearning setup, the requirement is to unlearn 200 examples from the original models, which is actually a relatively large set of unlearning data. We will add more details about the unlearning setups in our revision.
>
> Additionally, the G-effect serves not as a metric but as an analytical tool; therefore, the computational cost associated with it is not a primary concern. Furthermore, the insights gained from the 5% setup using the Llama-2-7B model are applicable to other models and unlearning scenarios, as evidenced in Table 1. We will clarify the role and utility of the G-effect in our revision. Thank you for your comments, and we apologize for any confusion caused.
>
> > **Q11**. “The G-Effects across Unlearning Steps” I don’t see what the useful insights are here. I believe we could have instead measured the NLL of examples and gotten an equally useful signal. Overall I found it difficult to assess if the G-Effect is a more useful metric than directly measuring losses over unlearning.
>
> **A11**. We agree with the reviewer that looking at NLL is more straightforward but complementary to the analysis of G-effect. As discussed in A7 & 8, analyzing gradients offers more information beyond mere performance changes, motivating comprehensive analysis in Section 4 for a series of representative unlearning objectives. The analysis is pivotal as it introduces several new state-of-the-art objectives and helps us to identify new research problems, such as how to enhance the weighting mechanisms within the NPO. We will include the related discussion in our revision, thanks again for your valuable suggestions!

---

> ### Author Response · Authors · 2024-11-24
> **Looking forward to your reply**
>
> Dear Reviewer uXUc,
>
> Thank you for your great efforts in reviewing our work and for your insightful questions. Following your suggestions, we have conducted additional experiments to strengthen our findings. We have also elaborated on our main contributions and clarified numerous technical details for enhanced clarity.
>
> Furthermore, we have included a discussion in A6 on the similarities between the goals of full removal and influence removal. We argue that our focus on full removal provides valuable insights that can further benefit the research in influence removal as well.
>
> We really hope that our answers can help to clarify. Please let us know if there is anything else you need further information on or any additional concerns you might have.
>
> Best regards,
>
> Authors of # 1253

---

> ### Author Response · Authors · 2024-11-25
> **Looking forward to your reply**
>
> Dear Reviewer uXUc,
>
> We sincerely thank you for your efforts in reviewing our work! We believe we have addressed some of your initial concerns. As the deadline for discussion approaches, please let us know if you require any further clarifications. We are eager to continue the discussions with you.
>
> Looking forward to your response!
>
> Best regards,
>
> Authors of #1253

---

> ### Author Response · Authors · 2024-11-26
> **Looking forward to your reply**
>
> Dear Reviewer uXUc,
>
> We are concerned about the possibility of not being able to continue discussions after the deadline. We would greatly appreciate it if you could let us know whether our responses have addressed your concerns. We remain eager to engage in further discussions as needed.
>
> Best regards,
>
> Authors of #1253

---

> > ### Comment · Reviewer_uXUc · 2024-11-28
> > **Reply**
> >
> > Thank you for your detailed response.
> >
> > > In our revision, we will further emphasize the advantage of analyzing gradient behaviors over simply observing performance changes.
> >
> > I would very much like to see this.
> >
> > > Given the challenges of the influence eliminating setups and many unresolved questions that are general within the community, we believe it is pragmatic to pursue a simpler yet still meaningful setup as ours.
> >
> > I disagree. As a field, collectively optimizing towards an objective we know is insufficient to satisfy agreed upon criteria for unlearning is not a good idea. I believe this will delay unlearning as a serious field of study rather than progress it.

---

> > > ### Author Response · Authors · 2024-11-28
> > > **Further discussions**
> > >
> > > Thank you for your feedback! We deeply respect your opinion and believe that disagreement is a vital component in advancing any research field. Your previous reviews clearly reflect your extensive experience and expertise in related areas, and we will certainly consider your insights as we plan our future research directions.
> > >
> > > We also want to express our opinion for the realistic significance of full removal. **First**, the community of differential privacy has developed many intriguing methods in achieving influence removal. However, to our knowledge, many of them still face the challenge of negatively impacting overall performance—a dilemma that is also observed in full removal and extensively explored in our paper. Given the similar challenges encountered in both goals of unlearning, we believe our research offers valuable insights and benefits for full removal.
> > >
> > > **Second**, we believe that pursuing the goal of full removal will inherently facilitate influence removal. Our belief is grounded in a practical connection between the two, where methods crafted for full removal, further incorporating certain early stopping criteria (e.g., those explored in membership inference), can serve as a reasonable approximation for influence removal. To our knowledge, sadly, this type of framework has not yet been explored in the literature. However, given the challenges in achieving full removal directly, we guess that such a simplified framework might prove to be practical and achievable.
> > >
> > > We are curious to know if the discussions above can address some of your concerns, and we are still eager to hear your opinion on these matters, which will be very important for us. Thank you very much!

---

### Official Review · Reviewer_zecJ · 2024-11-04

**Soundness:** 3
**Presentation:** 4
**Contribution:** 4
**Rating:** 8
**Confidence:** 3

**Summary:**

The paper beings with a bird's eye view of unlearning, highlighting its importance and the main drawbacks of the current approach. The paper suggests G-effect, a method to better examine the behavior and properties of unlearning during learning. This is done by taking into account the gradients with respect to unlearning two objectives and computing their inner-product with the gradients taken for unlearning. This method has the potential to serve as an evaluation criteria of existing and new unlearning approaches, while also providing insights to explain the success and failure of unlearning approaches, beyond the simple black-box evaluation.

This method is interesting and powerful, as it was able to generate five observations on unlearning: Unlearning affects shallow layers more, Unlearning compromises retention, Excessive unlearning is harmful, Risk weighting is powerful, Regularization is important.

The paper then explores multiple unlearning approaches; GA, NPO, PO and RMU using the G-effect.

The paper address the current limitation of the approach and suggests new promising directions.

**Strengths:**

The paper is insightful and well written. It provides a lot of context to the reader, explains the main drawbacks of the current approach and the importance of G-effect in addressing and examining unlearning from a scientific point-of-view, as opposed to blackbox trial and error which stems from intuition. The paper provide a very thorough analysis of G-effects on multiple unlearning approaches.

**Weaknesses:**

Not sure

**Questions:**

Is something off with Figure-3 or its caption?

---

> ### Author Response · Authors · 2024-11-21
> **Responses to Reviewer zecJ**
>
> We would like to express our sincere gratitude for your invaluable support in our paper! We believe that our work contributes meaningfully to the field. Our findings include several simple yet potent unlearning objectives that merit further investigation. Additionally, the G-effects we explored could provide benefits beyond the scope of unlearning objectives, offering broader implications for the community.
>
> Moreover, we sincerely apologize for the omission of captions in Figure 3, where the x-axis represents the unlearning step and the y-axis indicates the values of the G-effect. We will clarify this description in our revised manuscript.
>
> Thank you once again for your great support, which means a lot to us! Together, let all of us continue to advance the field of LLM unlearning!

---

> ### Author Response · Authors · 2024-11-24
> **Many Thanks for Your Support and Looking Forward to Further Discussions**
>
> Dear Reviewer zecJ,
>
> We sincerely thank you for your efforts in reviewing our work and for your great support! Please let us know if you need any further information or if there are additional points you would like to discuss with us.
>
> Thank you very much!
>
> Best regards,
>
> Authors of #1253

---

> > ### Comment · Reviewer_zecJ · 2024-11-26
> >
> > No further questions from my side.

---

### Official Review · Reviewer_SBC5 · 2024-11-04

**Soundness:** 2
**Presentation:** 2
**Contribution:** 3
**Rating:** 3
**Confidence:** 4

**Summary:**

Various approaches have been proposed in the literature to perform LLM unlearning. Existing unlearning evaluation metrics compare LLM performance before and after unlearning. To provide more insights into understanding the underlying mechanisms, this paper aims to quantify the impacts of unlearning objectives on model performance from a gradient perspective. To do that, this paper proposes the toolkit of the G-effect:
1. unlearning G-effect: the capability to decrease model performance on unlearning data
2. retaining G-effect: the capability to maintain/enhance performance on other data

G-effect compares the gradient of the unlearning objective and the risk metric that assesses the LLM performance:
 - If the gradients of the unlearning objective align in opposite directions to the risk metric, model updating based on the unlearning objective is capable of decreasing model performance measured by the risk
 - If the gradients of the unlearning objective align in similar directions to the risk metric, model updating based on the unlearning objective is capable of enhancing model performance measured by the risk metric

This paper quantifies the degree of such similarity between unlearning objective gradients and the risk metric gradients using their dot products.

**Strengths:**

1. Gradient-based analysis of the G-effect enables a better understanding of unlearning approaches including
	- examining the dynamics of unlearning procedures
	- explore the impacts of particular layers or data points involved during unlearning
2. Using G-effects to assess Gradient Ascent, Weighted Gradient Ascent, Negative Preference Optimization, Preference Optimization, Representation Misdirection for Unlearning and Regularization. Their study concludes several interesting findings. For example among 3 representative regularization terms, namely gradient difference, KL divergence and representation retention, KL  is superior for retention.

**Weaknesses:**

1. This paper (Section 4) examines G-effects of each unlearning objective independently and in isolation to other learning objectives. Results are also shown and discussed in separate figures and parts of the paper. Studying G-effect of each learning objective in isolation, raises the concern regarding the comparability of G-effect values across various unlearning objectives and approaches.
 	- Why empirical analysis of each unlearning approach is shown and discussed in separate parts of the paper?
	- Are G-effect values comparable across different unlearning approaches? are values comparable and why?
	- Can the proposed G-effect rank unlearning approaches?
 2. Section 5 and its Table 1 provide a comprehensive comparison of various unlearning approaches using TOFU unlearning dataset for the removal of fictitious author profiles from LLMs finetuned on them. However, this comparison uses only existing metrics: forget quality, model utility, and PS-scores, and does not report the proposed G-effects.
 	- Why G-effects are missing in this section?
 	- How do G-effect values correlate with metrics presented in Table 1?
 	- Why are the order and ranking of unlearning objectives different across different removal and retention metrics?

3. G-effects need access to intermediate checkpoints during unlearning, especially given the pattern of values in for example Figure 3 (i.e., a peak and then flat close to zero). How does this limit the applicability of the proposed metric?

4. The G-effect definition uses model checkpoints at different time steps and does not directly take into account the risk and unlearning of the initial model.
	- Why does this make sense?
	- Is this why you need to do accumulative?
	- what does the G-effect at each unlearning step mean?
	- what does accumulation across unlearning steps mean?
	- What does pick mean in Figure 3? Should we stop after that step to have an effective unlearning? what would be the benefit of continuing? is 0 G-effect value the limitation of your method?

5. Some of the claims are not completely supported. For example, the claim "In terms of the unlearning G-effects, it indicates that the unlearning strength of NPO is weaker; however, for the retaining G-effects, it suggests that NPO better preserves the model integrity." As an initial step, I would link it to numbers in Table 1.
6. Membership inference attacks are a common approach in the literature for evaluating the removal capability of unlearning approaches [MUSE]. However, this paper does not report the success of membership inference attacks. How the unlearning G-effect is compared to the success of MIA? Are they aligned?

[MUSE] Weijia Shi, Jaechan Lee, Yangsibo Huang, Sadhika Malladi, Jieyu Zhao, Ari Holtzman, Daogao Liu, Luke Zettlemoyer, Noah A Smith, and Chiyuan Zhang. Muse: Machine unlearning six-way evaluation for language models, 2024.

**Questions:**

I have outlined questions for each weakness above.

---

> ### Author Response · Authors · 2024-11-21
> **Responses to Reviewer SBC5 (part 1/3)**
>
> Many thanks for your constructive comments and suggestions! Please see our responses below.
>
> > **Q1**. Studying the G-effect of each learning objective in isolation, raises the concern regarding the comparability of G-effect values across various unlearning objectives and approaches. 1) Why empirical analysis of each unlearning approach is shown and discussed in separate parts of the paper? 2) Are G-effect values comparable across different unlearning approaches? are values comparable and why? 3) Can the proposed G-effect rank unlearning approaches?
>
> **A1**. We apologize for any confusion that may arise. We would like to first clarify **that the G-effect is not primarily designed for performance assessment**, as merely reporting performance scores does not sufficiently aid our understanding of the mechanism behind different unlearning objectives, as discussed in Sec 3. Instead, the primary purpose of the G-effect is to delve deeper into the unlearning dynamics and mechanisms, aiming to figure out the factors that contribute to the observed model behaviors.
>
> As derived in Appendix A, the G-effect quantifies the performance changes across a series of gradient update steps. However, due to the reliance on first-order approximation, the error may accumulate when applying too many update steps. This potential issue can be mitigated by calculating the G-effect across a series of equal-spaced unlearning steps.
>
> Moreover, **we agree with you that further quantitative analysis of the G-effect will convey more information**. Since the G-effect characterizes the performance changes, we can compute the overall effects of the gradient by summing the G-effects across the sampled steps, thereby allowing for performance comparison and ranking. Below, we list the results computed by the accumulated G-effect (AGE for short), following the same setup as in Fig 3.
>
> |             |    GA   |   NPO   |   WGA   |   TNPO  |  WTNPO  |   PO  |   RMU   |
> |:-----------:|:-------:|:-------:|:-------:|:-------:|:-------:|:-----:|:-------:|
> | Unlearn AGE |  -2.6e5 |  -3.0e3 |  -1.1e3 |  -394.1 |  -5.1e2 |  63.1 |  -8.3e3 |
> |  Retain AGE |  -4.5e5 |  -3.5e3 |  -232.1 |   -28.4 |   -66.4 |  29.9 |  -5.9e3 |
>
> **Since G-effects characterize gradient behaviors, their analysis slightly differs from that for metrics**. Here, we would like to highlight two findings from the results above. Comparing across methods for their retain AGE,  we find that previous methods, such as GA, NPO, and RMU, exhibit large negative impacts on retention. Comparing between unlearn and retain AGEs for each method, it is evident that our suggested methods, e.g., WGA, TNPO, and WTNPO, offer a more favorable trade-off between data removal and unlearning. **All these results quite align with our experimental results in Table 1**. These observations are well aligned with the experimental results presented in Table 1. Additionally, we note that the unlearn AGE for GA is excessively high, potentially leading to a scenario commonly referred to as "excessive unlearning." We plan to include these discussions and further elaborate on these findings in our revision. Thank you for your great comments!
>
> > **Q2**. Section 5 and its Table 1 provide a comprehensive comparison of various unlearning approaches using TOFU unlearning dataset for the removal of fictitious author profiles from LLMs finetuned on them. However, this comparison uses only existing metrics: forget quality, model utility, and PS-scores, and does not report the proposed G-effects. 1) Why are G-effects missing in this section? 2) How do G-effect values correlate with metrics presented in Table 1? 3) Why are the order and ranking of unlearning objectives different across different removal and retention metrics?
>
> **A2**. Thank you for the great question. As mentioned in A1, **the G-effect is not primarily proposed as an evaluation metric** but a tool for analyzing the dynamics of different unlearning procedures. The results in Section 5 are employed to cross-check that our observations and conclusions derived from the G-effect are reliable and correct. In A1, we further reveal **how the accumulation of the G-effects can facilitate our quantitative analysis**, and this leads to some conclusions that are aligned with results in Table 1.
>
> The variation in the order and ranking of unlearning objectives can be **attributed to the use of different metrics, each designed to capture distinct aspects of unlearning efficacy**. For example, for PS-based scores, they are intended to gauge the knowledge parameterization and the success rate of membership inference attacks. On the other hand, MU metric focuses on the probability of generating targeted responses and the degree of similarity between current and original model outputs. This variation among metrics is precisely why we employ a diverse set of metrics in our experiments, rather than limiting ourselves only to those original metrics proposed by TOFU.

---

> ### Author Response · Authors · 2024-11-21
> **Responses to Reviewer SBC5 (part 2/3)**
>
> > **Q3**. G-effects need access to intermediate checkpoints during unlearning, especially given the pattern of values in for example Figure 3 (i.e., a peak and then flat close to zero). How does this limit the applicability of the proposed metric?
>
> **A3**. We are happy to further highlight that the primary goal of the G-effect **is not to serve as a metric** for assessing model performance. Rather, the G-effect goes one step further to explore the dynamics and mechanisms of the unlearning procedures. This analysis helps us identify factors that influence observed model behaviors. Given our focus that involves unlearning dynamics, it is essential that we access intermediate checkpoints. Overall, the intended audience and users for the G-effect are researchers who would like to have a better understanding of unlearning dynamics beyond the overall results from standard unlearning metrics like PS and MU.
>
> **The benefits of the G-effects are substantial**: by analyzing the unlearning dynamics, we identify shortcomings in current methods and propose a series of more advanced ones. Thus, accessing intermediate checkpoints offers significant benefits, not limitations. We apologize for any confusion caused and will clarify our discussion in the revised version of our manuscript.
>
> > **Q4**. The G-effect definition uses model checkpoints at different time steps and does not directly take into account the risk and unlearning of the initial model. 1) Why does this make sense? 2) Is this why you need to accumulate? 3) what does the G-effect at each unlearning step mean? 4) what does accumulation across unlearning steps mean? 5) What does pick mean in Figure 3? Should we stop after that step to have effective unlearning? what would be the benefit of continuing? is 0 G-effect value the limitation of your method?
>
> **A4**. According to Definition 1, the model updates are computed based on the dot products between **the unlearning objective** and the **cross-entropy risk**, relative to the **current model parameters**. Therefore, there is some deviation in your understanding. Moreover, we briefly discuss the rationale behind this mechanism in Section 3 and provide a more detailed derivation in Appendix A. For save your valuable time, we would like to further summarize the key insights as follows.
>
> Overall, **the G-effect quantifies performance changes across a series of future gradient update steps**. However, due to our reliance on first-order approximations, errors associated with higher-order terms in the Taylor expansion may accumulate **when the number of update steps becomes too large, leading to a decrease in the accuracy of the G-effect in reflecting unlearning behaviors**. To mitigate this issue and better capture the unlearning dynamics, we calculate the G-effects at each unlearning step. Additionally, we are also pleased to further highlight that **each unlearning step corresponds to a stochastic gradient update encountered throughout the unlearning process**.
>
> **To elucidate the concept of accumulation and the physical meaning of the peak observed in our analysis**, we take Fig 3(a) as an example. As observed, in the later stage of unlearning, such as at step 60, the strength of unlearning becomes negligible compared to the pronounced effects at the middle stage, such as at step 25. This observation does not imply that the unlearning process allows for the relearning of previously unlearned knowledge. Rather, the strong strength of unlearning in the middle stage has been applied into the models. The diminished additional effects of unlearning in later stages simply reflect that the method has reached its convergence.
>
> Therefore, the G-effects calculated at each unlearning step are aggregated to reflect the overall performance changes of the model throughout the unlearning process. Additionally, we are pleased to further highlight that **the G-effects converging to zero in the later stages of unlearning simply indicate that our methods have reached convergence**, not that they have become ineffective. This convergence demonstrates the stability of the unlearning process and the robustness of our methods.
>
> Moreover, we concur with your excellent suggestions that implementing early stopping after the pick values of the unlearn G-effects can enhance the overall unlearning efficacy, particularly in terms of preserving common performance. To illustrate this, we compared the PS-exact scores at steps 20, 25, and 60, using the same setups as in Figure 3(a).  The results are listed in the post below. As observed, appropriate early stopping, as the reviewer suggests, is shown to be beneficial for retention. Many thanks for insightful suggestions, we will add the related discussion in our revision.

---

> ### Author Response · Authors · 2024-11-21
> **Responses to Reviewer SBC5 (part 3/3)**
>
> | step | PS-exact retain | PS-exact unlearn | PS-perturb retain | PS-perturb unlearn |
> |:----:|:---------------:|:----------------:|:-----------------:|:------------------:|
> |   5  |      0.7868     |      0.7472      |       0.4694      |       0.4016       |
> |  10  |      0.7607     |      0.6253      |       0.4618      |       0.3209       |
> |  15  |      0.3324     |      0.1675      |       0.2595      |       0.1520       |
> |  20  |      0.1288     |      0.0543      |       0.0991      |       0.0547       |
> |  25  |      0.0767     |      0.0139      |       0.0615      |       0.0077       |
> |  30  |      0.0004     |      0.0000      |       0.0016      |       0.0000       |
>
> > **Q5**. Some of the claims are not completely supported. For example, the claim "In terms of the unlearning G-effects, it indicates that the unlearning strength of NPO is weaker; however, for the retaining G-effects, it suggests that NPO better preserves the model integrity." As an initial step, I would link it to numbers in Table 1.
>
> **A5**. We sincerely apologize for any confusion caused! Our claim can be justified from two perspectives.
>
> 1. **G-effects**: For unlearning, colored in orange, we prefer larger magnitudes of G-effects that are negatively valued. We observe that the average values of GA notably exceed those of NPO, leading us to conclude that GA is more effective than NPO in facilitating unlearning. Conversely, for retention colored in blue, we aim for smaller magnitudes to minimize the adverse impacts on retention. Here, the average values of NPO are lower than those of GA, supporting our claim that NPO outperforms GA in terms of retention.
>
> 2. **Metrics**: The results from Table 1 support the same conclusion. For example, in the 5% unlearning setup with Phi-1.5, the PS scores for retention are notably higher for NPO compared to GA, indicating that NPO excels in retention. On the other side, the PS scores for unlearning are much lower for GA than for NPO, indicating GA is markedly more effective at unlearning.
>
> > **Q6**. Membership inference attacks are a common approach in the literature for evaluating the removal capability of unlearning approaches [MUSE]. However, this paper does not report the success of membership inference attacks. How the unlearning G-effect compare to the success of MIA? Are they aligned?
>
> **A6**. Thank you for your suggestion! In fact, we have reported the performance under membership inference attacks. The adopted PS score, initially discussed in [1], serves as a powerful metric for assessing membership. We will emphasize the relationship between PS and MIA in our revision, further underscoring the importance of MIA in LLM unlearning, as per [MUSE]. Thanks again for your wonderful comments!
>
> We apologize for any confusion that we may cause. We will make our discussion clearer in our revised manuscript. We also welcome any further questions or points of confusion you may have. Please do not hesitate to discuss them with us. Thank you very much!
>
> [1] Carlini, et al. Extracting Training data from Large Language Models.

---

> ### Author Response · Authors · 2024-11-22
> **Looking forward to your reply**
>
> Dear Reviewer SBC5,
>
> Thank you for your great efforts in reviewing our work and for your insightful questions. We really hope that our answers can help to clarify. Please let us know if there is anything else you need further information on or any additional concerns you might have.
>
> Best regards,
>
> Authors of # 1253

---

> ### Author Response · Authors · 2024-11-24
> **Looking forward to your reply**
>
> Dear Reviewer SBC5:
>
> Thanks for your great efforts in reviewing and good questions here. Since the discussion due is approaching, please let us know if anything we could further clarify.
>
> Best regards,
>
> Authors of # 1253

---

> ### Author Response · Authors · 2024-11-25
> **Looking forward to your reply**
>
> Dear Reviewer SBC5,
>
> We believe that we have addressed your initial concerns, and we would like to highlight two key points from our responses:
>
> - **The G-effect is not an evaluation metric**.  Our G-effect extends beyond merely assessing performance, as we aim at delving deep into the underlying mechanisms.
>
> - **The G-effect should be computed across steps**. We explain in Section 3 that our aim is to trace gradient dynamics throughout unlearning, along with further justification in Appendix A for its feasibility.
>
> As the discussion due is approaching, please let us know if anything else we could further clarify.
>
> Best regards,
>
> Authors of # 1253

---

> > ### Comment · Reviewer_SBC5 · 2024-11-25
> > **Thank you, read your response.**
> >
> > No further questions from my side.

---

> > > ### Author Response · Authors · 2024-11-25
> > > **Thank you**
> > >
> > > Dear Reviewer SBC5,
> > >
> > > Thank you very much for taking time reading our response. We are glad to hear that our response addressed your questions. If you are satisfied with our rebuttal, we would appreciate it if you could reconsider your score.
> > >
> > > Best regards,
> > >
> > > Authors of # 1253

---

> > > ### Author Response · Authors · 2024-11-28
> > > **Looking forward to your reply**
> > >
> > > Dear Reviewer SBC5,
> > >
> > > We understand that there must be some reasons for your decision to maintain your original score, and we are keen to know your opinions and any concerns you might have, which will be invaluable to us! We deeply respect diverse perspectives and are committed to integrating them to truthfully advance the field.
> > >
> > > We look forward to continuing our discussion at your convenience. Thank you very much for your time and efforts in reviewing our work!
> > >
> > > Best regards,
> > >
> > > Authors of # 1253

---

### Official Review · Reviewer_43oV · 2024-11-04

**Soundness:** 3
**Presentation:** 4
**Contribution:** 4
**Rating:** 6
**Confidence:** 4

**Summary:**

Large language models (LLMs) need thorough audits to uncover risks like copyright or privacy violations. When such risks are identified, prompt updates are essential to filter out inappropriate outputs, ensuring that models remain lawful and safe for use. This concern has fueled new research into LLM unlearning, aimed at selectively erasing specific unwanted knowledge while maintaining the reliability of other, unaffected responses. In this context, the authors of this paper introduce the G-effect toolkit, which measures how unlearning objectives influence model performance through the analysis of gradients.

**Strengths:**

The paper is clearly written and straightforward to understand.

**Weaknesses:**

It is uncertain if the findings discussed in this paper can be applied to scenarios where LLM unlearning aims to eliminate the effects of contaminated data.

**Questions:**

Overall, this paper is quite intriguing. The authors introduce an innovative framework called G-effect, designed to measure the influence of unlearning objectives on model performance through gradient analysis. However, there are some points for the authors to consider:

1) Unlearning is typically applied in two scenarios: first, for privacy reasons, where users seek to remove content related to privacy. This is the scenario explored in this paper. However, unlearning can also be driven by security concerns, such as when an LLM is compromised by data poisoning, and the system needs to mitigate the impact of various poisoning attacks. It remains unclear if the conclusions drawn in this paper can be extended to this second scenario.

2) G-effect appears to share similarities with influence functions and Shapley value methods. The authors should clarify these connections.

3) Research, such as [A], suggests that LLM unlearning may not always fully eliminate the influence of data that users wish to remove. It is not evident whether the proposed G-effect accounts for this limitation.

[A] Machine Unlearning Fails to Remove Data Poisoning Attacks.

---

> ### Author Response · Authors · 2024-11-21
> **Responses to Reviewer 43oV**
>
> > **Q1**. Unlearning is typically applied in two scenarios: first, for privacy reasons, where users seek to remove content related to privacy. This is the scenario explored in this paper. However, unlearning can also be driven by security concerns, such as when an LLM is compromised by data poisoning, and the system needs to mitigate the impact of various poisoning attacks. It remains unclear if the conclusions drawn in this paper can be extended to this second scenario.
>
> **A1**. Sincere thanks for your wonderful comments!  To the best of our knowledge, the security issue aligns more closely with the classical definition of machine unlearning, which focuses on **eliminating the influences of targeted data**. However, unlearning for LLMs with the goal of eliminating the influence of poisoned data is often much more challenging if not infeasible. The LLMs may have to be retrained from scratch to eliminate the data influence. Hence, in related work [1] and this paper, we discern the scope of full removal with the goal of eliminating influence.
>
> Fortunately, as our experiments demonstrated, these two objectives—measured by parameterization strength (eliminating knowledge as the goal) and forget quality (aligning with models trained without targeted data as the goal), respectively—are typically aligned, with only differences observed when a relatively large scale of data is unlearned. We totally agree that eliminating data influence is also a crucial research question, and we plan to explore the use of G-effects to assess this influence elimination in future studies. Additionally, we will incorporate a discussion on the role of unlearning in combating data poisoning following your mentioned paper in our revised manuscript. Many thanks for your wonderful suggestion!
>
> [1] Sijia et al. Rethinking Machine Unlearning for Large Language Models.
>
> > **Q2**. G-effect appears to share similarities with influence functions and Shapley value methods. The authors should clarify these connections.
>
> **A2**. We totally agree with the reviewer that G-effect shares a close connection with influence functions and Shapley value methods. All of them aim to elucidate the factors that affect model output behaviors. **The G-effect closely resembles the formulation of influence functions** when assessing the inherent influence of training data on test data via their respective gradients.
>
> However, **there are also substantial differences between G-effects and these previous toolkits**.
>
> 1. The G-effect primarily explores the roles of objectives, with respect to data of our interest, in shaping performance, whereas influence functions and Shapley value methods focus on the impact of individual data points or features on performance.
>
> 2. The G-effect is derived from the first-order approximation of the SGD dynamics, while influence functions and Shapley values are computed by the linearization of optimal solutions and are based on the average marginal contributions, thus serving completely different purposes. Also, over the Shapley, the G-effect is differentiable and may guide the design of optimization objectives in the future.
>
> We will include a discussion on these points in our revision. Thank you very much for your wonderful suggestions!
>
> > **Q3**. Research, such as [A], suggests that LLM unlearning may not always fully eliminate the influence of data that users wish to remove. It is not evident whether the proposed G-effect accounts for this limitation. > [A] Machine Unlearning Fails to Remove Data Poisoning Attacks.
>
> **A3**. Many thanks for your suggestion! Addressing data poisoning indeed broadens the applicable scenarios of unlearning, and we recognize it as both an important and intriguing future research direction. Although the exact objectives of unlearning may slightly differ from those we have explored, we believe **the G-effect and our studies can significantly contribute to the understanding of the mentioned setups**. For example, the trade-off between unlearning and retention is still a critical aspect that remains at the forefront of data poisoning. However, as mentioned in A1, eliminating the influence of **data poisoning also has its unique challenges**, particularly in controlling the extent of unlearning so that only the influence of the poisoned data is removed.
>
> Overall, we believe our work provides valuable insights for studies that address data poisoning from the unlearning perspective, although **further research is necessary to formally adapt to this promising area**. For example, a particularly interesting question unique for eliminating data poisoning involves verifying the conditions between models before and after unlearning, which can guarantee/indicate that only the influence of specific data points has been removed. We will include this discussion and reference the paper you mentioned in our revision.

---

> ### Author Response · Authors · 2024-11-24
> **Many Thanks for Your Support and Looking Forward to Further Discussions**
>
> Dear Reviewer 43oV,
>
> We sincerely thank you for your efforts in reviewing our work and for your great support! We also greatly appreciate your insightful questions and hope that our responses have helped to clarify them.
>
> Please let us know if you need any further information or if there are additional points you would like to discuss with us. Thank you very much!
>
> Best regards,
>
> Authors of #1253

---

### Official Review · Reviewer_MskC · 2024-11-08

**Soundness:** 4
**Presentation:** 4
**Contribution:** 4
**Rating:** 8
**Confidence:** 2

**Summary:**

This paper discusses the importance of auditing large language models (LLMs) to identify potential risks and the need for timely updates to remove bad responses (unlearning being the solution in consideration here). The authors propose a unified framework to comprehend various unlearning objectives, introducing the concept of G-effect to analyze and compare different unlearning methods. Insights from G-effect analysis prompt focus on a two-step approach to update original LLM parameters to get unlearned one- using a removal and retention step to deteriorate performance on the unlearning dataset while maintaining performance on the rest of the data.

**Strengths:**

-   The paper tackles an important topic in the field of natural language processing, specifically the need for LLM unlearning to remove targeted information without destroying model integrity.
-   The authors propose a novel framework to analyze and compare different unlearning objectives, introducing the concept of G-effect to measure the impact of unlearning objectives on targeted or common data. G effect combines the influence of both goals of a good unlearning objective- removal and retention into a single metric.
-   The paper provides a comprehensive discussion of different unlearning objectives, including gradient ascent, negative preference optimization, PO, and representation misdirection for unlearning. The paper introduces advanced unlearning objectives, such as WGA and WTNPO, which set new state-of-the-art results in unlearning objectives.
- Comparison between objectives Gradient Ascent, NPO, etc. and effect of regularization is thoroughly studied.
The paper introduces advanced unlearning objectives, such as WGA and WTNPO, which set new state-of-the-art results in unlearning objectives.

**Weaknesses:**

- Ablation study on the effect of size of data to be forgotten on the effectiveness of G-effect.
- Effect of 'harder' or 'easier' examples to forget on G-effect.
- Figures plotting the G-effect have their axes not labeled.

**Questions:**

- Is it possible to see the effect of choice parameters like number of samples to be forgotten on the G-effect ?
- Other ablation studies are also appreciated- for instance choice of network architecture, sensitivity to optimizer params, etc. It wasn't clear if these results are averaged over multiple runs.
- I am curious to see how G-effect varies for samples that are harder to forget. This may even help explaining the relationship between influence functions and data values with ability to unlearn.

---

> ### Author Response · Authors · 2024-11-21
> **Responses to Reviewer MskC**
>
> Many thanks for your great support and constructive comments! Please see our responses below.
>
> > **Q1**. Ablation study on the effect of size of data to be forgotten on the effectiveness of G-effect. Other ablation studies are also appreciated- for instance choice of network architecture, sensitivity to optimizer params, etc. It wasn't clear if these results are averaged over multiple runs.
>
> **A1**. Thank you for your suggestions! Due to the high computational costs associated with calculating the  G-effect, we did not report the average values across multiple runs.  We are running experiments with different random seeds to confirm that the results are consistent across runs and we will add the related experiments and discussions in our revision. Regarding the ablation study concerning data size and network architectures, we conduct preliminary experiments in comparing GA and WGA. We observed that **the trends are consistent with our reported findings**. We will include more results and study cases in our revision, thanks again for your comments!
>
> > **Q2**. Figures plotting the G-effect have their axes not labeled.
>
> **A2**. We apologize for any confusion caused. In our figures, **the x-axis represents the unlearning steps, and the y-axis indicates the values of the G-effect**. We will ensure that our figures will be updated in our revised manuscript. Sincere thanks for your comments!
>
> > **Q3**. I am curious to see how G-effect varies for samples that are harder to forget. This may even help explaining the relationship between influence functions and data values with ability to unlearn.
>
> **A3**. Thank you for your insightful suggestion. The G-effect can indeed serve as a powerful tool to explore the hardness of data from a data-centric perspective. **We follow your suggestion and design the experimental setup as follows**.
>
> We define the hardness of data by the point-wise G-effect with respect to either unlearn or retain data. We then rank the data targeted for unlearning based on these point-wise G-effects, selecting the half with the smallest G-effects as "top" data, while the other half, characterized by larger G-effects, is classified as "down" data. Then, **we use either only “top” or “down” data as the filtered data adopted for unlearning**.
>
> Taking WGA as an example, we follow the same experimental setups as in Fig 3, further considering the unlearning of the whole dataset (WGA), top data with retain G-effect (WGA top retain),  down data with retain G-effect (WGA down retain),  top data with unlearn G-effect (WGA top unlearn), and down data with unlearn G-effect (WGA down unlearn). The results with respect to PS scores are summarized as follows.
>
> |                    | PS-exact retain | PS-exact unlearn | PS-perturb retain | PS-perturb unlearn |
> |:------------------:|:---------------:|:----------------:|:-----------------:|:------------------:|
> |         WGA        |      0.5005     |      0.0057      |       0.3376      |       0.0041       |
> |   WGA top unlearn  |     0.5090      |      0.0275      |       0.3462      |       0.0200       |
> | WGA bottom unlearn |      0.4001     |      0.0257      |       0.3012      |       0.0250       |
> |   WGA top retain   |      0.5319     |      0.0132      |       0.3391      |       0.0116       |
> |  WGA bottom retain |      0.3871     |      0.0218      |       0.3574      |       0.0237       |
>
> Surprisingly, sampling with respect to either unlearn or retain G-effects, it is typically easier to remove data with large G-effects, as this tends to better preserve the performance of the original data. However, this observation does not apply to rephrased data, indicating that further exploration is necessary to gain a deeper understanding of these dynamics.

---

> ### Author Response · Authors · 2024-11-24
> **Many Thanks for Your Great Support and Looking Forward to Further Discussions**
>
> Dear Reviewer MskC,
>
> We sincerely thank you for your efforts in reviewing our work and for your great support! We also greatly appreciate your insightful questions and hope that our responses have helped to clarify them.
>
> Please let us know if you need any further information or if there are additional points you would like to discuss with us. Thank you very much!
>
> Best regards,
>
> Authors of #1253

---

> ### Comment · Reviewer_MskC · 2024-11-25
> **Thank you Authors for the clarifications**
>
> I appreciate the comments addressing my questions. I believe that the ablation studies or at least a runtime analysis of the G-effect method would improve the adoption of this work. Adding results averaged over different random seeds is important.  Your results on hard / easy to unlearn examples further warrants exploration between the connection between influence function / shapley based attribution methods and G-effect.
>
> I encourage the authors to actively use and promote the use of G-effect to study the field of LLM unlearning in the future. Better adoption will only benefit and improve on the current G-effect computation method. I have increased my score!

---

> > ### Author Response · Authors · 2024-11-26
> > **Thank you**
> >
> > Dear Reviewer MskC,
> >
> > We sincerely appreciate your great support, which means a great deal to us! In response to your suggestions, we promise to include additional ablation studies, averaged results, as well as explore the connections with influence function and Shapley value methods in our revision.
> >
> > Thank you once again for your time and support!
> >
> > Best regards,
> >
> > Authors of #1253

---

### Author Response · Authors · 2024-11-24
**General Response to All Reviewers**

We would like to express our deep gratitude to all the reviewers for their time and insightful feedback. We appreciate the recognition of our analysis as **comprehensive** and **insightful**  (MskC, SBC5, zecJ, and oKM6). We are also grateful for the comments that our empirical results **support** our claims (MskC, uXUc, oKM6) and that our paper is **clearly written** (43oV, zecJ, oKM6).

Endorsed by most reviewers, our work contributes meaningfully to the field by introducing a general toolkit named the G-effect, which facilitates the in-depth study of unlearning dynamics. Our contributions, leveraging the corresponding analysis, can be broadly summarized in twofold.

- We carefully analyze several popular unlearning objectives, including GA and NPO, revealing their **key unlearning mechanism** as well as their **failure modes**.

- We mitigate the drawbacks of existing unlearning objectives by **designing their improved versions**, supported by extensive experimental results that demonstrate their practical effectiveness.

We also thank the reviewers for their comments to conduct more experiments and improve clarity. Additionally, we are pleased to further clarify two points that may have caused slight confusion for some reviewers (SBC5 and uXUc).

- The G-effect **extends beyond the role of a metric** for performance assessment. It is intended for researchers seeking an in-depth understanding of unlearning dynamics, beyond what standard unlearning metrics like PS and MU provide.

- This paper particularly focuses on the unlearning goal of full removal. However, we acknowledge **its similarity with another representative goal** named influence removal, where our findings could benefit such a broader community. We also recognize the unique challenges posed by influence removal and intend to further adapt our G-effect for it in future work.

We have responded to each reviewer's valuable critiques in our individual responses and look forward to continuing this valuable discussion during the conference discussion period!

---

### Meta-Review · Area_Chair_pdfo · 2024-12-10

**Metareview:**

- This paper introduces G-effect, a gradient-based toolkit for evaluating unlearning methods by analyzing their dual impact: deteriorating performance on targeted undesirable data (unlearning G-effect) and maintaining performance on other data (retaining G-effect). The G-effect enables a detailed examination of various unlearning algorithms and generates five general observations: unlearning affects shallow layers more, unlearning compromises retention, excessive unlearning is harmful, risk weighting is powerful, and regularization is important.

- This paper is well-structured and easy to follow. It conducts insightful analyses, considering both unlearning and retention, and provides a comprehensive exploration of various unlearning objectives (e.g., GA, NPO, PO, RMU). To address the limitations of existing unlearning methods, the authors further propose advanced objectives such as WGA and WTNPO. Experiments demonstrate the effectiveness of the proposed variants.

- However, it is uncertain whether the proposed framework, general observations, and unlearning variants are applicable to other scenarios (e.g., data poisoning) or align with practical legal compliance requirements. The empirical analysis is limited to small datasets (~40 samples) without results averaged over multiple runs. Additionally, there is a lack of clarity regarding the relationship between G-effect and mechanisms like influence functions or Shapley value methods, as well as insufficient analysis of gradient behaviors that could provide insights beyond performance changes.

**Additional Comments On Reviewer Discussion:**

- In the rebuttal, the authors addressed most of the concerns raised by the reviewers.

- In the revision, the authors should address the writing issues, misunderstandings, and clarifications raised by the reviewers. They should include the updated experiments, such as evaluations on hard/easy unlearning examples, discussions about other application scenarios, and alignment with practical legal compliance requirements (e.g., GDPR). According to Reviewer uXUc, the authors should further provide analysis of gradient behaviors to offer insights beyond mere performance changes.

---

### Decision · Program_Chairs · 2025-01-22

Accept (Poster)